# Urban Soils and Road Dust—Civilization Effects and Metal Pollution—A Review

## Manfred Sager

Bioforschung Austria, Esslinger Hauptstrasse 134, A-1220 Vienna, Austria; m.sager@bioforschung.at

**Abstract:** Urban soils have been changed much by human impacts in terms of structure, composition and use. This review paper gives a general introduction into changes from compaction, mixing, water retention, nutrient inputs, sealing, gardening, and pollution. Because pollutions in particular have caused concerns in the past, metal pollutions and platinum group metal inputs have been treated in more detail. Though it is not possible to cover the entire literature done on this field, it has been tried to give examples from all continents, regarding geochemical background levels. Urban metal soil pollution depends on the age of the settlement, current emissions from traffic and industry, and washout. It seems that in regions of high precipitation, pollutants are swept away to the watershed, leaving the soils less polluted than in Europe. Health hazards, however, are caused by ingestion and inhalation, which are higher in 3rd world countries, and not by concentrations met in urban soils as such; these are not treated within this paper in detail. With respect to pollutants, this paper is focused on metals. Contrary to many reviews of the past, which mix all data into one column, like sampling depth, sieved grain sizes, digestion and determination methods, these have been considered, because this might lead to considerable interpretation changes. Because many datasets are not Gaussian distributed, medians and concentration ranges are given, wherever possible. Urban dust contains about two to three fold the hazardous metal concentrations met in urban soils. Some data about metal mobilities obtained from selective and sequential leaching procedures, are also added. Soil compaction, pollution, sealings and run-offs cause stress situations for green plants growing at roadside locations, which is discussed in the Section 5. Environmental protection measures have led to decrease metal pollutions within the last decade in many places.

**Keywords:** urban soils; roadside dusts; trace metals; metal mobilities; platinum group metals

## 1. Introduction

Urban soils might be defined as soils within the administrative boundaries of municipalities or settlings, respective a territory of settlement and natural production, including rests of soils in young cities.

This review should cover all aspects of urban, roadside soils and roadside dusts. Large quantities of pollutants such as trace metals, polycyclic aromatic hydrocarbons, and phthalate esters, are released into the urban environment [1]. Though soil compaction, hydraulic properties, carbon storage and organic contaminations may be of equal importance, due to the experiences of the author, however, it focusses on metal contaminations, particularly the so-called heavy metals and platinum metals, of which much more information is available than about e.g., rare earths and non-metals. If thresholds exist, risk assessment studies necessitate respective investigations and ensure payments for intense analyses, whereas for others, financing is shorter. Because activated surfaces of small and nano-particles have catalytic properties, e.g., Pt, their mere concentration data are just indications of their health relevance, but this is beyond the scope of this article.

Because it is impossible to treat all published data within this review, the reader is recommended to look at other compilations also (e.g., [2–5])

## 2. Soil Function Changes from Human Civilizations

The common approach of calculating accumulation of contaminants down the profile by comparing concentrations of a sequence of layers, cannot be used with urban soils, because they are often mixed, and contain additional extraneous materials. Estimating a city's influence on soil contamination is made by comparing obtained data either with legislative limits, or with background values obtained from natural comparable soils [6].

Long-term changes in soil properties occur where the capacity of soil to retain contaminants is exceeded and discrete metal-mineral phases are formed [7]. The long-term input of metals to the soil can result in decreasing buffering capacity of the soil and to groundwater contamination [8].

The continuing cycle of construction, use, and renewal of urban structures leads to far higher rates of change in urban environments than it is common elsewhere [9].

An overview about urban soil taxonomy and history of urban soil investigations has been given by Lehmann and Stahr [10].

### 2.1. Soil Use

In general, soil functions are infiltration and purification of ground water, water evaporation, location for plant cover, and storage of organic carbon. Considerable compaction reduces soil functions in metropolitan areas. The evaluation of urban soils is important for urban land-use planning, like water infiltration, compaction and contamination [10].

About 6% of the total area of Europa is urban, and 52% of the world's population live in urban or peri-urban areas, but in Europe about 73–80% [11,12]. Human population in urban areas is still expanding. In addition to grow food and feeds, soil in urban areas has recreational and biodiversity preserving functions [2].

Land use, technogenic impact and the disturbance of biological cycles are anthropogenic activities, which discriminate 5 urban landscape regions, i.e., park-recreational, agrotechnogenic, residential, residential-transport, and industrial. In urban soils, variabilities of microbial indices increase, bacterial community structures change, and new predominating populations appear, which are adapted to urban conditions [13].

Human impacts on soils also occur from agricultural activities like long-term flooding, terracing and deep ploughing. Man-influenced soils contain little or no artefacts, but mixing of soil horizons. Man-changed soils contain anthropogenic soil horizons containing ashes, coal, wastes, and particularly after war destruction, bricks and plastics. Man- made soils comprise artefacts solely or mainly from anthropogenic material [10].

The conditions of the urban soils vary from one city to another, depending on the urban structure, the distribution of vehicles, and the type of fuel used [12].

In an urban environment, you may find park soils, garden soils, roadside soils, and fallows. In gardens and parks there may be more biodiversity than in farmland because of neophytes, whereas roadside soils and fallows are extraordinary habitats.

Urban green spaces can contribute to food security, particularly in developing countries, whereas in more developed countries, they are rather for leisure activity or ecological activism to reduce food transportation routes [14].

Fallows are defined as areas where at least natural vegetation can develop for one growth period, like high grasses, bushes, and even trees. Urban fallows develop in areas of past human activities, but are currently left alone, like around ruins, left industrial grounds, former military objects, or along railways. These soils are often full of stones, dry and usually low in nutrients, and wind-borne seeds start growing with intense rootings, but nutrient-rich spots also occur, indicated by e.g., stinging nettles. After the first pioneers, more intense growing develops, until a stable ecosystem develops within

5–10 years, with trees and in their shadows, grasses at the bottom. Then there is much larger biodiversity than in parks, and retreat of animals, like rodents and birds. Because the bottom still contains an urban habitat, natural local woods cannot develop. Many of these fallows finally disappear to yield locations for new houses, parking lots, or parks. Because there is no obvious reason to sample these rather heterogenous fallow areas, soil chemical data are currently not available to the knowledge of the author.

## 2.2. Human Impacts from Constructions

Urbanization means the replacement of the natural ecosystem. Due to excavation, redistribution and mixing of soil horizons, high variabilities in the short range occur, as well as rapid changes of land use [2].

Urban soils are more alkaline from construction residues, higher in carbon, contaminants, density and soil temperature, but lower in soil moisture because of sealing and drainage [10]. Spatial and vertical distribution of soil parent rocks and a high share of technogenic materials strongly influence the state of urban soils. The presence of technogenic materials, like construction debris, slag dust, coal and sludge, also strongly influences the migration of water and solutes in the soil profile and may act as parent rocks. Construction works change the order of the soil horizons or mix topsoils with subsoils. Soil chemical properties vary with urban age, because of the change of construction techniques and law regulations [11].

Urban soils are prone to direct littering. Before the time of organized waste collection, garbage was burnt in the city streets to reduce waste volume, thus increasing organic matter and ashes in the topsoil. Industrialization intensified the anthropogenic influence, increasing soil excavations and introducing artefacts like bricks, glass, metals and plastics, but also inputs from atmospheric deposition [10]. Plastic waste, ceramics, and metal items do not degrade as easily as biowaste and may be also contained in compost as small grains. In Austria, high quality compost used for landscape design and conservation, should not contain more than 1% of ballast dumpings, including 0.4% plastics > 2 mm, 0.04% microplastics > 0.02 mm, and 0.2% glass [15]. Anthropogenically deposited metals may remain in urban soils for centuries. In Vienna at the time of the Roman settlement about 1700 years ago, streets levels were about 8 m deeper [16], and cultural layers down to 15 m depth have been reported from Moscow [10]. This effect is occurring at almost every site of long civilization as an effect of the input of construction materials, waste and soil compaction.

## 2.3. Human Impacts from Roads

Road construction means soil decapitation, sand application and final sealing the surface. A large surface deposition of carbonates from the atmosphere was noted in industrial areas, but also from coal heating and waste incineration. Thus, the pH of technosols was 0.8 pH units higher than for the soils met in parks and forests, and salinity significantly increases at roadside soils [11].

In the roadside environment, the area 0–2 m is dominated by runoff water, 2–10 m by splash water, and 10–50 mm by airborne pollutant transport. Roadside soils often contain up to 30% of technogenic materials and stones, as well as abrasion from road and brake linings, which cause alkalinization. With respect to soil profiles, at the distance of 5–10 m, the soils are compacted with little vegetation, and after 10–15 m predominantly original soil profiles occur. Lysimeter studies showed that the emitted metal compounds get preferably immobilized by the first 10 cm of roadside soils, particularly Pb, whereas Cd and Zn can penetrate to deeper soil layers. Elevated pH enhances metal retention. After intense rainfall events, solutes and particles get transported aside via preferential flow. Urban soils receive NaCl and $CaCl_2$ for de-icing in winter, as well as Ca-rich irrigation water during dry and hot summers, which results in an increase of pH. Throughout the world, annually about 35 million tons of salt are applied to roads for maintenance in winter. The runoff of these dissolved salts infiltrates the soil and salinifies the groundwater. Na induces ion exchange versus Ca and Mg and other metals, and Cl concentrations in limnic waterbodies and wetlands

increase. In addition, increased temperatures compared to their surrounding leads to increased evaporation [17,18] (Jamshidi et al., 2020). De-icing agents increase Cd, Cu, Pb and Zn in the soil solution, possibly by ion-exchange. Enrichment in topsoils is also due to physical retardation of particulate trace elements. Vertical transport in roadside soils within 0–5 m distance was highest for Cd, and in some cases also for Cu, but low for Cr, Ni, Pb and Zn. This depends on soil type, road operation time, vegetation, rain intensities, soil pH etc. In 5–15 cm depth, Cd may even increase with respect to top layer contents [3].

In the city of Hamedan (Iran), 110 days with minimum temperatures below the freezing point, and snow cover up to 4 months are expected, because of its position at high altitude (1850 m, close to mountains of 3500 m). In Malayer (Iran), at 1700 m above sea level, 65 days below the freezing point occur on average. At both sites, 10 wells were sampled within the period of 2004–2018. Whereas chloride concentrations in Malayer had only little changes over this 15 year period, chloride levels in Hamadan wells were about 5 times higher, and increased 2.5% annually [18].

*2.4. Carbon and Phosphorus*

In urban soils, soil functions and soil organic carbon dynamics are different to natural or agricultural soils. Sealed soils might be undisturbed covered soils, or anthropogenically altered with additions of charcoal, building materials and waste of all kinds. Because buildings and streets store solar energy much more than fields, meadows and pastures or forests, the average T is 2 °C higher (city of Vienna) than in its vegetated surroundings.

As a result of a greater compaction, microbial decomposing organisms are generally lacking from urban soils owing to the lack of oxygen in the soil pores, which results in low amounts of humic substances and thus low adsorption capacities [17].

In cities like Manchester, greenspace soils are enriched in organic carbon, but highly variable. In undisturbed sealed soil, showing rather horizontal layers of soil horizons, about 4 kg organic C per m$^2$ had been found, particularly bound to the clay-size fraction, which is, however, regarded as less available. There has been long-term input from waste and ashes, but not from green plants. Particular organic matter of >20 μm has been assumed to be more plant available [19].

In New York City parks, study soils were coarse textured, the >2 mm fraction ranged from traces to 21%, but clay contents were low. Soil pH ranged from 3.9–7.4, but most values were <5.0. High levels of asphalt and slag fragments significantly contributed to total carbon [20].

In urban soils of St. Petersburg (Russia), the radiocarbon age of organic matter varied between 500 and 2700 years, thus the humic layers are older than urban life. Urbanization caused alkalization, compaction till 44% of density and thus lowering of porosity and gas exchange. Humus horizons of the lawns along highways were top contaminated with Pb, Zn, and Cu, whereas the major nutrients NPK were irregularly distributed [21].

In summer 2009/2010, 18 sites were sampled in six central city districts in Russian cities, situated in different climatic zones, at a distance of 1–1.5 m from highways, and a depth of 3–10 cm. In the northern cities, organic carbon, pH and electrical conductivity increased at urban sites versus control due to addition of peat compost mixtures and carbon-containing pollutants, like bitumen, soot and petroleum products. In southern regions, both rising and falling trends of organic carbon were noted, e.g., from removal of fallen leaves. In particular in parks and public gardens, there was a trend of soil enrichment with organic carbon in the northern cities, and depletion in the southern cities. pH increase was due to Ca release from construction waste, application of de-icing agents, and liming of green plantations. In the southern cities, pH remained either the same or even decreased because of downward migration of carbonates and increasing anthropogenic load. Electrical conductivity increased from north to south two to six times due to increased electrolyte contents, but less pronounced in areas of high precipitation level and watered tree plantations [13].

Prokaryote microbial communities operate as destroyers of anthropogenic pollutants, regulate the composition of air fluxes, and the availability of nutrients for plants. In Russia, their amount

increased in the suburb soils from north to south due to rising temperature and precipitation, additional energy substrates, and new ecological niches. The amount of microorganisms increased upon low anthropogenic load, and decreased upon high load. Compaction of surface horizons reduced the abundance in courtyards. Under urban conditions, actinomycetes, rhodococci and halotolerant bacteria number grew. A part of them may acquire tolerance to heavy metals [13]. The intensity of $CO_2$ and $CH_4$ emission as well as nitrogen fixation activity in soils in the suburbs rose from north to south. Contamination with easily soluble salts, carbon containing compounds and heavy metals decreased microbial respiration, particularly $CO_2$ emission. Methanogenic activities depend on the activity of archaebacteria and correlated with the contents of easily available organic substances, and also clay content in the soil. Methanogenic microorganisms develop under anaerobic conditions, strong compaction, and neutral to weakly alkaline pH. Maximal methane formation was registered in high humus and heavy loamy soils at the steppe regions, and minimal at acid and light textured soils. The activity of molecular nitrogen fixation increased from north to south [13].

Phosphorus is usually enriched in urban areas and may cause aquatic eutrophication, originating from household ash, anthropogenic waste deposits and wastewater, as well as from fertilization of parks. In Nanchang, Southern China, total phosphorus and mobile fractions were analyzed in park soils at 0–5 cm, 5–15 cm, and 15–30 cm depth, along an urban—rural gradient. The original soils were ferralsoils, acid read clays with kaolinite. Total phosphorus in urban soils was more than twice than that of the rural soil at all depths. Sequential leaching was done by extraction with $H_2O$, 1 M KCl, 0.1 M NaOH and 0.5 M HCl, and final digestion with $HClO_4$. In the NaOH extract, the difference between soluble reactive P and total P was estimated to occur as organically bound P. Main fraction was the inorganic NaOH-extractable P, which correlated strongly positive with exchangeable P. Increase of soil pH, Ca and Mg in urban environments due to the dissolution of calcareous materials in cement from built environments caused a rise of the proportion of HCl-extractable P, in parallel with acid mobile Ca and Mg, as well as a decrease of oxalate-extractable Al. The inorganic NaOH-extractable P was significantly correlated with oxalate extractable Fe. Soil pH and HCl-extractable P could be fitted versus log Ca by a parabolic equation. P in the groundwater was significantly correlated with $NaHCO_3$-extractable P of the soils [22].

### 2.5. Methods to Detect the Changes

Contaminations can be caused by metals, organics, radionuclides, and pathogens. Industry, traffic, fuel combustion and waste disposal lead to erosion, salinization, compaction, sealing and contamination. Knowledge of environmental pollution is important for planning strategies to achieve better urban environmental quality and risk control [2,23].

Sampling is usually done in parks and roadsides, but sampling depths vary widely, from mere road dust to mixed samples of down to 30 cm, which means different amount of dilutions of deposited loads with the original soil. In this review, road dust has been defined by mere brushing, whereas road soils and urban soils have been sampled at a given depth at unpaved sites. Some papers use a sampling grid including presumably non-contaminated suburban areas, whereas others sample preferably at hotspots. Standardization is still lacking to ensure comparability of datasets.

High density surveys are more commonly used in urban areas to map the relationships between soil contaminants and land used. In urban soils, short-range vertical and spatial variability are commonly observed, resulting from excavation and backfilling. In cities, many playgrounds, parks and home gardens are located adjacent to major highways, landfills, brownfields, and human made lands [20].

In soil analysis, dry sieving down to 2 mm grain size is usual, but some papers refer to lower grain sizes (details given). Unlike in most sediments, smaller grain sizes do not mean higher metal levels, because abrasion of e.g., paintings and tires might be found in the coarse fractions as well, whereas inputs from combustion processes prefer fine fractions.

Urban ring trees can also be used to detect changes in elemental concentrations of environmental materials [24].

In the city of **Ostrava** (ČR), changes of the composition of street dust between identical sampling strategic points in 2008 and 2018 have been monitored by XRF chemical analysis, magnetic susceptibility and scanning electron microscopy, and compared with 144 topsoil samples taken 0–10 cm. Most of the data were not normally distributed, and thus logarithmically transformed. Changes of industrial emission occurred from desulphurization of energy sources and technological modifications of the sintering process in the steel work [25].

## 3. Metal Contaminations

### 3.1. Sampling and Determination Methods

For digestion, aqua regia, admixtures with HF, nitric acid plus hydrogen-peroxide, as well as sieving to various grain sizes, have been used, which should be considered in comparison and interpretation of datasets. Because the amount of Cr and V released by aqua regia can be down to 1/3 of total but may be much higher in substrates of high organic substance contents, comparison of data obtained from different digestion methods should be done with caution. Limits and risk assessment for agricultural soils refer to standard aqua regia digests, whereas geological maps refer to total contents [26,27].

If data refer to digests in aqua regia or reverse aqua regia, comparability with total contents data is rather qualitative. Whereas carbonic, oxidic, sulfidic and organic phases are usually recovered completely, it is well known that much of the silicate matrix remains in the residue filtered off, and standardized conditions (boiling time and temperature) are needed to get reproducible results for the partially weathered silicates. Therefore, data compilations without giving digestion method and grain size, should be handled with care!

Digestion with aqua regia was found comparable to digestion with $HNO_3$ for As, Cd, Cu and Pb, whereas it dissolved more Fe, Mn, Ni and Zn [8].

### 3.2. Metal Sources

Soil serves both as a sink and a source for trace metal contaminants in the terrestrial environment. Anthropogenically deposited metals may remain in urban soils for centuries. Main anthropogenic sources of urban dust are combustion processes (heating in winter, cooking), industrial emissions, weathering of building structures and streets, waste disposal, and traffic. Some metals can form volatile oxides and halogenides during combustion [28], leading to condensation aerosols till deposition and adsorption at dust particles. Filter systems reduced hazardous emissions from coal—and oil-fired power plants, gasworks, metallurgical, chemical and electronic plants, and motor vehicles [29].

Metal emission patterns close to roads are a function of driving activities, like deceleration and acceleration, speed, traffic density, traffic lights and roundabouts. Roadside soils are constantly loaded with both organic matter and metals [30].

In Palermo, fertilizers enhanced P-levels about four to five fold with respect to natural background soils, and additional input of Hg-Pb-Cu-Zn was indicated by significant correlations with P [31].

Urban soils have been found enriched versus mean upper crust values for Ag, Cd, Hg, Pb, Sb, Sn and Zn, particularly at historic sites, and also occasionally As, B, Co, Cu, Mo and Ni. Particularly in case of Cr, but also for V, Mn, and Ba, the data depend on the digestion method, and should not be taken to a table without this additional information. In addition, sieving to finer fractions leads to higher concentrations of pollutant metal inputs, at least for stream sediments, or as condensation aerosols, whereas coarser particles often can be traced to rock abrasions [32].

Median and range seem to be more indicative to compare occasional pollutions between different datasets than means and standard deviations, because data are not normally distributed in many cases, and top values are better recognized (see also Nemerov pollution index). Some compilations

about urban soil and roadside dust data can be found in the papers of [2,3,33]. However, apart from mean and median values, range and top pollutions, as well as the analytical method, grain size and sampling period have to be considered. Nevertheless, the location and the selection of samples sites seems dominant.

In order to visualize interelement relationships within multi-element datasets, principal component analysis has been performed in many cases. Frequencies of joint occurrences or pollutions for individual metals are added in the text below.

### 3.2.1. Brakes

The abrasion of brakes produces particles of high Cu, Ba, Fe, and Zn, as well as of K, Sb and Ti. $K_2O.nTiO_2$ is sometimes used to improve the terminal resistance and to lower the brakes wear [34]. Sulfides ($Sb_2S_3$, SnS, $MoS_2$) and barite are present at percentage levels within the under-layer brake pads, or added as lubricants [12,35]

Micro-analysis of single dust grains sampled in Prague city revealed that the abrasion of brakes produces coarse aerosol particles of high Cu-Ba-Fe-Zn, resp. Sb-K-Ti. $K_2O.nTiO_2$ is sometimes used to improve the thermal resistance and to lower the brakes wear [34].

### 3.2.2. Tires

As a result of tire wear, roadside samples get enriched with Zn, and sometimes also with Cd and Cu (e.g., in Sydney), tires release ZnO particles, and Cd is probably a contaminant of technical grade Zn [34,36–39].

### 3.2.3. Metal corrosions

Corrosion of vehicle chassis and tire wear contaminates roadside dusts and soils with Zn, Cu, and Cd. In Prague city, Fe-Cu-Mn-Zn had been assigned to the abrasion of different vehicle parts, occurring at a size of about 2.5 μm, but ultrafine particles carried also much Zn, indicating combustion sources. Si-Al-Ca-K were centered in particles at about 5 μm, representing the elements typical from earth crust erosion [34,39].

Metal corrosion increase Zn, Cd, Cd, Ni, Cu and Mo [38].

### 3.2.4. Antimony

For the Upper Continental Crust, an As/Sb proportion of about 6 can be expected, which is widely met in soils and sediments also. While arsenic contaminations have recently decreased thanks to a massive decline of arsenic emissions, antimony contaminations indicate a dangerous trend due to growing automotive traffic. Enrichment of antimony in dusts has been observed in many countries, even in Arctic ice [40].

$Sb_2O_3$ and $Sb_2S_3$ are required in the production of polyethylene terephtalate, $Sb_2O_3$ as a flame retardant in rubber, Sb-dialkyl-dithiocarbamates are used as additives in greases and some oils, and as metal as hardener for Pb bullets and batteries [35,41]. $Sb_2S_3$ has been introduced in brake pads as a replacement for asbestos [37]. Additional point sources for Sb, e.g., in Palermo (Italy), were fireworks [31].

Due to health risks, asbestos brake parts have been generally replaced by non-asbestos organic type pads. Within a model testing series, each deceleration event caused emissions of 34 μg Sb, assuming a car of 2000 kg running at 40–60 km/h. Higher disk T caused emission of coarser particles, i.e., 0.8 μm at 100° and 2.0 μm at 400°. Abrasion dusts from brake pads contained 1.5% antimony [42], and also 3.1% Fe, 15% Cu, 1% Zn, and 12% Ba. Sb is a component of brake linings and a flame retardant in vulcanization of rubber [35]. The samples of wheel rims, road dust and atmospheric particulate matter contain Sb in an admixture of $Sb_2S_3$, $Sb_2O_3$ and $Sb_2O_4$, the latter being a mixed oxide of Sb(III) and Sb(V). Brake pads may reach a temperature high enough to induce oxidation of Sb. Road dust of Palermo contained 104 ± 5 mg/kg Sb, and brake pads 1–2% Sb as $Sb_2S_3$, together with Fe and Cu [43].

In the environment, Sb is present as the octahedrally coordinated, pentavalent antimonate anion $Sb(OH)_6^-$ over a wide redox potential range. Compared to arsenate, Sb has a considerably higher mobility under oxidizing conditions due to a larger ionic radius and a lower charge density [41].

### 3.2.5. Arsenic

Some coals contain high amounts of As, which gets preferably lost to the fly ash [28]. Fossil fuels, in particular coal burning, are main sources for As in dust, but there are also some emissions from metal processing [2]. At least in the US, Pb from wheel weights contains 5% of As [44].

In Europe, contamination from atmospheric As has sharply decreased thanks to the pan-European regulation of As-emissions, about 8 times compared with 1985 [41].

### 3.2.6. Barium

Ba is a constituent of paints, ceramics, glass or plastics, but also contained in tires and brakes [12]. Ba dispersions are used as corrosion inhibitors in lubricating oil and smoke suppressants in fuels [38].

For soils, total digestion might yield about 4 times higher Ba concentration than aqua regia, thus reference data should be read with care [27]. Nevertheless, Ba met in some street dust (e.g., Seoul) was higher than expected for agricultural soils (in aqua regia), but there are not many data available. In Japan, abrasion dusts from brake pads contained 12% barium [42].

In principal component analysis, As appeared within the same component as Hg and Sb, but also surprisingly also together with Cr, Ni, and V in some cases, but never with Ba.

### 3.2.7. Cadmium

Cd is used for Ni-Cd batteries, pigments, as well as in plastics, electronics and tires (as a contaminant of ZnO). Almost all urban soils have been found to be enriched versus mean soil levels. Even more than Zn, Cd emissions can be due to combustion sources and smelters.

In urban areas, main cadmium sources are Ni-Cd batteries, electronics, photography, pigments, and plastics [2]. Phosphate minerals and emissions from smelters might be less important. Cd tends to accumulate in roadside dusts and soils, plants and trees [45–47].

Cd concentrations often paralleled those of Pb and Zn, but also with Cu in half of the cases found.

### 3.2.8. Chromium

Total digestion yields about 3 times more Cr than aqua regia, but the recovery from organic soils and composts can be higher. Chromium is used in metallurgic and galvanic industry, and is a likely component of engine metal alloys [44]. Motor vehicle exhaust and waste incineration, as well as metallurgic and galvanic industries have been regarded as the main sources of Cr in urban dust, but erosions of ultramafic rocks substrates may also contribute significantly. Cr is preferably bound to the residual fraction [2].

In principal component analysis, Cr was most frequently associated with Ni, and slightly less frequent with Co and As, but never with Hg.

### 3.2.9. Copper

Copper is likely recovered from any digest, and found remarkably enriched in the samples investigated. Electronic waste and electrical wearing [2] are well known for copper, but also abrasion dusts from brake pads contained 15% copper [42].

Cu is above all used for electrical and electronic equipment, leading to 2 kg of Cu-waste per capita in Europe. Contaminations have been detected above all in Europe and China, but hardly in Africa. Selective leaching confirms the affinity to the organic (=oxidizable) fraction in many cases, similar to agricultural and forest soils [2].

Some hotspots of Cu in urban soils can be caused by the corrosion of Cu-pipes [39].

In urban gardens and parks, Cu-foliar sprays containing about 29% Cu, may be still in use as fungicides. In organic farming, annual application had been limited to 6 kg/ha.a in the EU, but for non-food plants, application could be much higher [48].

Cu pollutions preferably occurred together with Zn and Pb, and less frequent with Cd, but never in the same component together with Al, Fe, and Mn.

### 3.2.10. Lead

Pb-sources are mainly batteries, glass, radiation shields, and electronic products. Pb-paints have been banned in 1978 (in Italy), but still corrode to yield atmospherically transported particulates. In road dust and urban soils, Pb-levels in African and Indian cities seem rather low, contrary to cities in Europe, China, Arabian countries and the US. Besides accumulation in the fine fraction, also discrete metal particles have been found [2]. In Australia, Pb-based paints were banned in 1970, but still particles from their deterioration get emitted [49].

Pb gasoline used was banned in South Korea in 1993, in Austria since 1 November 1993, in Hungary since 1999, in China since 2001, in Australia since 2002, and in Russia since 2003. Restriction of leaded gasoline use gradually leads to a decrease of Pb contaminations, but as Pb is largely immobile, levels remain still high. After cease of using leaded gasoline, main sources are brake wear and the loss of Pb wheel weights [35,44].

### 3.2.11. Manganese

Because of high ambient levels, in urban areas, main sources of manganese are fertilized agricultural organic soils around the cities for horticultural and crop production. Mn tends to accumulate in roadside dust soils, plants and trees [50].

Mn as an alloy component increases corrosion resistance [35]. Mn is a significant component for steel used in rails for trams and railways.

Methylcyclopentadienyl-Mn-tricarbonyl is used as an antiknock agent in fuel addition as a replacement for Pb [35,37].

### 3.2.12. Mercury

Sources of Hg are smelting, coal and oil combustion and waste incineration, but also chlorine electrolytical production, dental amalgams and lamps [2,51]. Additional point sources for Hg were laboratories e.g., in Palermo (Italy) [31].

In Berlin, the metal working industry, manufacturer of paints and other coatings, chlorine, asphalt, photochemicals and electric compounds were typically associated with Hg contaminations. Hg was often bound to organic matter, but also strongly adsorbed by Fe/Al-hydroxides and clay minerals [29].

An additional source of Hg might be a crematorium, when dental amalgams get volatilized, or dental clinics, unless appropriate filters are installed [8].

Among the components of urban soils and dusts investigated, Hg occurred preferably combined with As, and never with Al and Ba.

### 3.2.13. Molybdenum

Molybdenum is enriched in green plants and coals, and remains in the bottom ash. It is also frequently added to mineral fertilizers. Molybdenum has been used in lubricants and as a component of brake linings [35].

### 3.2.14. Nickel and Vanadium

Ni levels in urban areas, but also mean crust values, have been found variable [52]. Nickel is a component of stainless steel, and thus present in engine metal alloys, and appreciable amounts are also found in Ni-Cd batteries [2,35,44].

But Ni is also emitted by oil combustion, which is likely a main source of nickel pollution in street dust, and after oil spills. In oil, in particular vanadium, but also nickel, have got enriched, they are therefore regarded as markers from oil combustion. Medium and heavy crude oils as well as high boiling fractions contain high amounts of sulfur and metals, like 10–160 mg/kg Ni, bound to porphyrines. During raffination, Ni and V remain largely in the residue (asphalt), which contains an average of 336 mg/kg (range 250–500 mg/kg) Ni and >2000 mg/kg V (range 500–5000 mg/kg) [53].

Aqua regia does not release total vanadium, but also it may not release total nickel in all cases, particularly at hotspots. Thus, median nickel for arable topsoils in France has been determined as 21.0 mg/kg in total (4363 samples), and as 19.1 mg/kg (5184 samples) in aqua regia [54]. In the UK, a median of total Ni was found at 15.8 mg/kg in arable soils and at 22.0 mg/kg in urban soils [55].

Positive correlations of Ni were noticed preferably with Cr and Mn, but also with others in some cases, but not with Hg. V occurrence often paralleled Co, Cr and Ni, but never Ba, Cu and Zn

### 3.2.15. Sodium

Mean Na concentration in urban soils and dusts seems to decline with increasing distance to the seaside, but levels are also prone to the use of de-icing salts and washout.

Topography, ground water flow and hydrodynamic dispersion influences transport and fate of de-icers. Na are retained through adsorption to the clay particles, resulting in decreased soil permeability, whereas Cl is not retained during pore water flow. Model simulations revealed that even after the stop of de-icing, high salt concentrations will be still found near the roads and urban areas due to the slow release of halite compounds from the subsurface soils and groundwater storage). Agricultural activities can be negatively affected by salinization. Salt water intrusions can be also easily monitored by electrical conductivity measurements [18].

In wells in western Iran, the spatial distribution of As was consistent with the observed salt contamination, probably due to the dissolution of arsenopyrite, which can take place within long time spaces of interaction [18].

### 3.2.16. Zinc

Zn has been highly enriched in urban areas since the bronze age, and can come from many sources. In urban soils, runoff from roofs, gutters and windowsills contributes much to Zn enrichment. Zn is present in tire rubber within 0.5–2.0% ZnO, it is widely used in the metallurgic and galvanic industry, as well as for tires, batteries, electronic equipment, and alloys. It gets enriched in flue ashes, sludges and waste, and in some cases nearby smelters significantly contribute to enhanced Zn emissions, thus it tends to accumulate in the fine fraction [2].

Traffic related sources in particular are tire rubber, brake pads, safety fences, and oil additives for wear protection [35]. Abrasion dusts from brake pads contained 1% zinc [42]. In the US, Zn-dialkyldithiophosphate is used in oil additives and is a major component of galvanized metals and many paints [35,44]. In dust grains sampled in Prague, the broad size distribution of Zn having also an ultrafine mode reflects various other sources than from tires, like additives of engine oils [34]. Zn is usually completely recovered in aqua regia.

In principal component analysis, Zn often joined Pb, Cu, Cd, and possibly Sb (which was rarely investigated), and in fewer cases with others, but not together with main components Al and Fe.

### 3.3. Cumulative Indices for Characterization of Metal Contaminations

In order to get an overview about contamination levels and hazards from multi-element analysis, indices have been developed to get a rapid overview and to compare data worldwide.

### 3.3.1. Geo-Accumulation Index

The geo-accumulation Index has been defined by G. Müller [56,57] as the logarithm (based on 2) of the measured concentrations over the geochemical background times 1.5. This has to be calculated

for every element analyzed and shows the enrichment versus the original geogenic level met on site. The geo-accumulation Index classifies metal enrichment like it is usual for the microbial saprobial status (class 1–5); the concentration space of each class doubles with increasing class number, which approximates the effect of environmental hazard [56,57].

A geo-accumulation Index around zero means uncontaminated, around 1 means moderately contaminated, around 3 means heavily contaminated, and larger than 5 means extremely contaminated. For the calculation of the geo-accumulation Index, baseline values similar to the contamination index can be used. From this, e.g., Seoul road dust is strongly enriched in almost everything. Road dust sampled outside Budapest still contains significantly more Cu, Zn and Pb than the background [58,59].

Some problems might arise, if the background data refer to deeper sampling depths, or to aqua regia, and the urban soils or dusts refer to total contents, other grain size fractions and other sampling depths.

### 3.3.2. The Contamination Index

Like the geo-accumulation index, the contamination index refers to the background values of the soils in non-urbanized surroundings, which are slightly different between various sites. It summarizes the multi-element-approach to one number, thus smoothing individual outliers of a few contaminants only. As examples, calculations for Seoul and Budapest are given. Based on data from [60], the contamination index was calculated for Seoul as:

$$=\{(As/6.83) + (Cd/0.29) + (Cr/25.4) + (Cu/15.3) + (Ni/17.7) + (Pb/18.4) + (Zn/54.3)\}/7$$

Considering the background values of soils from the Hungarian Planes [61], and thus for Budapest, the contamination index was calculated as:

$$=\{(As/7.3) + (Ba/95) + (Cd/0.5) + (Co/9) + (Cr/21) + (Cu/19) \\ + (Hg/0.1) + (Ni/22) + (Pb/17) + (Zn/65)\}/10$$

### 3.3.3. The Pollution Index

The pollution index refers to the tolerable levels, which have been set by legal advices, based on health impacts. The tolerable levels refer to a defined grain size and digestion method. Because thresholds might differ, the pollution index has to be calculated for each country separately.

Legal limits for soil use and health hazards of metals and other pollutants change from country to country and from time to time, depending on the results of occupational studies and analytical possibilities. It is not clear in any case, if the thresholds refer to total or aqua regia soluble contents, which makes differences particularly for Cr. Some countries may have strong regulations, but administration is more or less tolerant. Others have no regulations and keep to the data of the main industrialized countries. In the country of the author, some regulations differ from province to province. Therefore, within this review, legal limits are out of scope, and the reader should rely on the specific national legislation.

Taking the aqua-regia based thresholds from the Austrian Province of Styria (plants grown at substrates above these levels need to be analyzed), this would be (Steiermärkisches landwirtschaftliches Bodenschutzgesetz) [62], if aqua regia data are taken:

$$=\{(As/20) + (Cd/2) + (Co/50) + (Cr/100) + (Cu/100) + (Hg/1) \\ + (Mo/10) + (Ni/60) + (Pb/100) + (Zn/300)\}/10$$

### 3.3.4. The Nemerov Pollution Index

The Nemerov pollution index uses both the average and the maximum value for each metal and is obtained from the root of half of the sum of respective squares. (Readers may calculate it



from median/mean and max given in the appended tables). It has been classified as "clean" for <0.7, "warning limit" between 0.7 and 1.0, "slight pollution" between 1 and 2, "moderate pollution" between 2 and 3, and "heavy pollution" for values >3 Thus it highlights the occurrence of a few heavily polluted sites more distinctly [63].

*3.4. Levels Met in Urban and Roadside Soils*

Urban soil geochemistry depends on geological, geomorphological, and landscape factors, in addition to the city's historical, economic and social conditions [64]. In this part, examples of special investigations made upon urban and roadside soils are given, arranged from North to South.

High annual rainfall and occasional flooding lead to rather low metal levels in South and South-East Asia because of washout, which spoils the groundwater instead. And last not least, ingested resp. amounts transferred to green plants are hazardous, and not the deposited amounts as such.

3.4.1. Europe

**Aberdeen** is a town at the Scottish east coast with no heavy industry, but busy with oil prospection, surrounded by soils that had developed on a glacial till derived from granitic rocks. Soils 0–10 cm were sampled at 50 roadside locations and 30 sites within public parks, sieved <2 mm, digested with aqua regia, and metal concentration levels compared with cultivated podzols within 50 km distance to the city. The lithogenic composition was similar throughout, which was concluded from marked similarities of Mg, K, Ti and Fe abundance, and confirmed by X-ray diffraction data for the soil minerals. The pH values for the roadside soils were about one unit higher than those from the parkland sites. Principal component analysis grouped data from Ca-Na-Cu-Zn-Pb-Sr-Ba into one component, which was enhanced in roadside soils, and P-Mn-Ni-Cr-Co into another one. With respect to rural soils, particularly Cu, Zn, Pb, Ba, and Ti were found at higher levels. With respect to parkland soils, in roadside soils some hotspots of Pb, Zn and Ba were observed, as well as more Na and Ca from de-icing salts and building material [65].

**Trondheim** is largely situated upon marine clay, and upon fluvial sediments along the Nidelva river. Anthropogenic cover makes an average of 2 m. Main rocks around are tholeitic basalt and greenschists. Soil samples were taken at 0–2 cm both in 1994 and 2004 at sites as identical as possible. Digestion with aqua regia in 1994 and 1:1 $HNO_3$ in an autoclave yielded similar results except for Na, Mo, Ag and B. All investigated elements followed a log-normal distribution. In 2004, concentrations tended to be lower except for As, and similar geographical patterns for Cr, Cu, and Ni were recognized in 1994 and in 2004. The oldest city parts showed higher levels of Cd, Hg, Pb, and Zn. In the area of the main crematorium and the hospital waste incinerator, a clear maximum of Hg, and enrichments of Pb, Zn and Cd were found. However, the effect of the waste incinerator, which was equipped with flue gas electrofilters and fluegas washing, was negligible. At roads, Pb and Hg decreased from 1994 to 2004, whereas Zn and Cd increased, due to increase tire wear from increasing traffic but use of unleaded gasoline [66].

In **Sweden**, roadside soils aside of two highways close to cities of Kalmar and Varberg were sampled at various distances as well as various depths, sieved <2 mm and digested with $HNO_3/HCl = 1 + 1$, as well as leached to obtain mobile fractions. At one site, Cd, Cr, Cu, Ni and Sb reached regional background levels after 5 m, and Pb and Zn after 10 m, whereas at another site, background levels were already reached after 2.5 m. During 20 years after construction and use of the road, Cd-Cr-Ni at roadside topsoils increased 3 times, Cu-Zn 6 times, and Sb 20 times versus background levels. High metal concentrations were found down to 10 cm. Principal component analysis grouped Cu-Sb-Zn and Cr-Ni together [30].

Soils of **Uppsala**, a town of about 140,000 inhabitants in central Sweden (in 2019: 231,000), are based on various igneous rock types of the Fenno-Scandian shield, either till, a sandy glacio-fluvial material with low clay content of 1–4%, or glacial clay containing 45–80% clay and 15–30% carbonate can be found, and occasionally gyttja clay with 40% clay and 2–30% organic carbon. This geochemical

background of Uppsala leads to naturally elevated contents of As, Cd, Cu, Co, Cr, Mo, Ni and V, versus low Hg, Pb and Se. Brickworks and milling industries, and later on, metal, textile, food and chemical industries were established, but moved away during the second half of the 20th century (decades before actual sampling), leaving a mainly academic city. Soils were sampled a 0–5 cm, 5–10 cm, and 10–20 cm depth, sieved <2 mm, digested with aqua regia, and categorized as industrial, roadside, central, natural and past industrial areas. Main anthropogenic sources were heating and energy plants, emitting Hg, Pb, Cr, Cu, Ni, Zn, and As. An additional source of Hg had been the crematorium, where installation of appropriate filters in 2000 reduced the annual Hg release from 8.5 kg to 0.5 kg [3]. Organic matter, pH, grain size, Al, Fe, Mn and Cr followed normal distributions, others had outliers. Similarly, throughout the profile, Al, Fe, Mn, Cr and Zn data showed normal distributions. Hotspots of Hg, Pb and W were found, but no correlations with clay contents. Leaded petrol was banned in Sweden since 1995, about 10 years before sampling in Uppsala. The layers 10–20 cm contained slightly more As, Al, Cr, Fe, Mn, and Ni, whereas no differences versus sampling depth were found for Cd, Cu, Hg, Pb, W and Zn. The cluster of As, Al, Cr, Fe, Mn, and Ni is most likely to have a common natural origin, while Cd, Cu, Hg, Pb and Zn were influenced by anthropogenic inputs. High sand contents observed in top layers of most sites indicate that this material originated from elsewhere. Higher percentage of fine particles in deeper layers is due to downward washing from rain [8].

**Gothenburg** is a center for trade and shipping at the Swedish West Coast, housing half a million inhabitants. Most trace metals occurred at a higher median levels in the soils of the older but smaller town Uppsala than in Gothenburg, particularly Cr and Ni 4-fold, and W 7-fold, whereas Cd levels were similar, and Hg and Pb were higher in Gothenburg. In Gothenburg, Cd, Cu, Hg, Ni and Pb were concluded to be of anthropogenic origin [8].

In **Tallinn,** the capital of Estonia (430,000 inhabitants), sandy and clayey soils occur in the northern part, sandy shallow soils in the northwestern part, and swamp soils on the peninsulas and near the lakes. Bedrocks are sandstones, siltsones, clays and carbonates. About 30% of the total content of Cu, Mn and Pb were exchangeable in weakly acid solution, but only 10% of Cr, Mn and Ni. Among 27 elements determined, principal components of K-Al-Fe-Ti-Ga-V can be assigned to terrigenous bedrocks, and Ca-Mg-Mn-Fe to carbonates. Components containing Cr-Mn-Ni-Ba and Pb-Zn-As are due to anthropogenic loading by industrial pollution. Summing up individual data by mapping the pollution index showed hotspots in the northern region, where it reached about 3.0, particularly close to a former but now exploited phosphorite deposit and at an industrial site of long use. Contaminations of soils by Cr, Cu, Mo, Ni, Pb, Yb, and Zn were connected with metal working, contaminations by Ag, Co, Cr, Cu, S, Sn and Zn with plastic article and trade equipment manufacturing, contaminations by Cr, Mo, Pb, Sn and Ti with radio engineering, and Pb and V with railways and roads [67].

In **Lithuania**, 3179 urban soil samples 0–10 cm were taken in 4 towns (Vilnius, Šiauliai, Mažeikiai, Joniškis), sieved <1 mm, ashed at 450 °C, and analyzed by DC-arc emission spectrometry, and finally compared with regional backgrounds. In all towns, Zn exceeded 300 mg/kg at most of the sites, and Ba exceeded 600 mg/kg in Šiauliai and Mažeikiai. Metal accumulations were greatest in Vilnius, especially in its central districts, which is the oldest (founded in 1323), largest and most densely populated town investigated, housing about half a million inhabitants. With respect to peripheral districts, accumulation in central Vilnius of Ag was found 2.63 fold, Pb 2.09 fold, Sn 1.53 fold, Zn 1.37 fold, and others less. Pb, Cu, Sn and Ag are indicators of transport and household pollution. A high percentage of >100 mg/kg Cr could be explained by tanneries and galvanic shops of the bicycle production plant. Zn can be related to transport or to corrosion of roofs. In Mažeikiai, accumulation of V and Cr reflect the impacts of the oil refinery and the power plant, whereas at this site, Pb, Ag and Sn were lower, due to the short history of the site starting in 1869 [68].

In **Warsaw** (Poland), about 1.7 million people live on 495 km$^2$. Soils 0–20 cm were sampled along 3 transects, sieved <1 mm, muffled at 450 °C, and boiled in 1 + 1 HCl. Soil pH (3.8–7.6; mean 6.6) and total organic carbon (0.7–8.5; mean 4.9%) covered a wide range. Sources of Cd

contaminations were atmospheric deposition of coal burning as well as metallurgical and electronic industries. Similar spatial distributions of Cu, Pb and Zn were observed. With respect to sampling at the same sites 20 years beforehand and analyzed by the same method, Cr and Mn accumulated, Pb increased at 45%, particularly in the city outskirts, but declined at 22% of the sites in the city center, just a few years after the introduction of leaded fuel in the early 1990ies [7].

**Zielona Gora** (Poland) is situated upon a moraine belt between two main ice-age valleys, upon sandy soils of glacial and alluvial origin. Deposits connected with human activities are mainly related to house and road building, in particular slag as a terrain filling material. In the close surroundings, pine monocultures have replaced most of the arable lands, which form Brunic Arenosols, and subsequently host new urban areas. 562 soil samples from 105 soil profiles to a depth of 150 cm were taken in 2012, sieved <2 mm, and extracted with water, 0.1 M HCl or aqua regia. The skeleton content of the soils was mainly building rubble, increasing pH, carbonate and salinity due to the solubility of lime and gypsum. Highest Cu was found in former vineyards, and Zn and Pb from technogenic materials of reconstruction. Hotspots of Cu, Pb, and Zn occurred at roadside areas and parks, of Zn and Pb also at squares and barren land, and of Cu and Zn also in sealed areas [11].

**Kielce** in central Poland is situated upon sedimentary rocks, containing fissure intrusive deposits of Pb, Zn, Cu, and Fe, which have been exploited since the 14th century, and which led to intense mining and processing. Nowadays, about 200,000 people live there upon 109 km$^2$. Soils had been sampled in residential areas, roads, urban greenery, allotment gardens, and agricultural areas at 60 sites for 0–20 cm, and sieved <2 mm, at the same places than 16 years before, using identical techniques. Implementation of efficient filters in the metallurgical plant as well as substitution of hard-coal in the thermal power plant significantly reduced dust emissions, but car-traffic increased. Highest Pb contents were still found along transportation routes, but threefold lower than 16 years ago, due to the used of unloaded petrol for at least 15 years. In urban greeneries and residential areas, however, Pb decreased only at about 35%. In addition, Zn decreased almost twofold within transportation areas, and almost remained at the same level in residential areas and allotment gardens. Additional sources of Zn in residential areas are the operation of motor vehicles, like abrasion of brake linings and coaches. Zn inputs to allotment gardens result from fertilizers, pesticides and composts. In soils of allotment gardens, also high As, Cd, Ba, Co, Cr and Ni levels were found, but still within ambient levels, which might be an effect of retainment within increased humus layers. Mn was generally high due to local geology. Largest positive correlations appeared between As and Co, Cr and Ni, and Co and Ni, as well in case of samples taken near the roads also between Ni and Zn. Principal component analysis resulted in high factor loads of all investigated elements in the first component representing the high impact of the geological substrate. The second component contains mainly Zn, the third mainly Ba, and the fourth mainly Pb (Ciupa et al., 2020) [69].

In and around **Berlin**, 4000 soil samples were taken 0–20 cm, sieved <2 mm and analyzed by mainly XRF. The inner city and surrounding area of Berlin showed enriched concentrations of Cd, Cr, Cu, Hg, Mo, Ni, Pb, S, Sr, and Zn with respect to the regional geochemical background. In Berlin topsoils 0–20 cm, anthropogenic loads of Cd, Cu, Hg, Ni, Pb, Sn and Zn exceeded the geogenic content more than double, whereas the input of As and Cr was found to be less. Al, K, Si, Na, Rb, Zr, Nb, Co, Sc and Ti in topsoils were mainly of natural origin. Old industrial sites have pollution peaks of Cu, Hg, and Cd [29]. Hg contaminations were typically associated with metal working industries, manufacturing of paints and other coatings, chlorine, asphalt, photochemicals and electric compounds. Hg was often bound to organic matter, but also strongly adsorbed by Fe/Al-hydroxides and clay minerals [29]. Factor analysis considerably reduced the number of parameters to be compared and could explain up to 90% of the total variance in the suburban industrial area of Schoeneweide. Factors of anthropogenic soil contaminations were Sr-F-La-Y-Ca from building materials, Cr-Mg-Ni-Mn-Co from industrial wastes, As-Hg from old anthropogenic loads, and Ag-Cd-P- from compost materials. Factors of natural origin contained Na-K-Rb-Al, and Zr-Nb-Zi [29].

At its western margin, the city of **Vienna** covers parts of the flysch zone which consists of sandstone, marlstone and shale. The Vienna basin is filled with marine to lacustrine sediments of Neogene ages, overlain by quaternary sand and gravel terraces deposited by the river Danube and its tributaries. Within the territory of **Vienna** (Austria), 286 soil samples of 0–10 cm depth were taken for screening purposes, submitted to aqua regia standard digestion and measured by ICP-MS. The points were selected along a sampling grid, or at the closest non-paved soil sites, and the sampling density intensified in areas of high population density and traffic intensity. From sampling 0–10 cm in 2003, Cd and Pb concentrations were significantly higher in traffic and industrial areas (sampling in 2003) as compared to parks and green spaces, but for other trace elements (As, Co, Cr, Cu, Hg, Mo, Ni, V, Zn) no statistically significant differences existed. In a few cases, Pb, Zn and Cu enrichment in soil were caused from washout of roofs and facades by spot infiltration. Maximum Cd, Cr, Mo and Zn was found along principal routes where is a lot of stop and go, and some park close by, whereas along the highways and smaller roads, levels were about equal. Cd, Pb and Zn were 2–3 times above the levels found for the same soil types in agricultural areas from the surrounding province. Some hotspots of Hg also occurred in parks, particularly in inner districts. The contents of As, Co, Mo, Ni, and V were within the range of geogenic variation. Soils sampled in Vienna city contained Ni just 30.3 ± 6.3 mg/kg. With respect to surrounding recreational areas still belonging to Vienna, Cr and Co were lower, and Hg higher in the Danube wetlands than in the Western forest. From preceding investigations done within 1992, 1994 and 1997 at the same sites, average Pb at the most polluted routes dropped from 250 mg/kg to 150 mg/kg, due to the prohibition of leaded gasoline in Austria at 1 November 1993. Zn/Cd was lower than in restrictive arable soils, indicating a preferred input of Cd over Zn. Principal component analysis grouped Cr-Cu-Ni, Co-V, As-Hg, and Cd together. The Co-V component revealed significant differences between soils overlying flysch and other geological formations. Transects across the city showed a 12 km wide central area of higher trace element levels. At levels of the Roman settlement, about 8 m deeper, Cd-Cu-Zn was found 3 times lower, Pb 6 times lower and Hg 17 times lower than recent. Inputs of heavy metals through abrasion from brake pads and tires were proven by mineralogical investigations [16,70,71]. Surprisingly, median levels of investigated metals in roadside sides and parks were not different, but top values of Cu, Mo, Pb and Zn were met at roadsides. From samplings 1997 to 2003, Cu-Hg-Pb-Zn declined, whereas As-Cd-Co-Cr-Mo-Ni-V remained at the same ambient levels [16,70].

In **Sopron** (Hungary), the original slightly acidic clay soil (pH 5.9) had been changed by introducing building waste and artifacts to raise the soil pH to 6.2–7.8, and to 6–14% $CaCO_3$. The alkaline pH is favorable for metal accumulation. Corrosion of vehicle chassis and tire wear contaminated roadside dusts and soils with Zn, Cu, and Cd. Whereas Co and Ni were found at about ambient levels, some hotspots of Cu were caused by the corrosion of Cu-pipes. Despite the disappearance of leaded fuels, Pb levels remained still high, because Pb is largely immobile in humus layers of green areas and within alkaline soils [39].

**Szeged** is a major city in Southern Hungary with 160,000 inhabitants, continental climate, receiving an annual precipitation of 520 mm at a mean temperature of 10.5 °C. Soils had developed on fluvial deposits of Pleistocene and Holocene, in particular phaeozems upon a loessy parent material, fluvisols upon alluvial parent material, as well as gleysoils and solonetz soils. However, after a great flood in 1879, the original surface was in parts elevated by significant infillings of the low-lying areas. Infilling and mixing transformations led to large modifications of horizontal and vertical soil parameters. High degrees of surface cover and high artefact contents resulted in poor microbial activities. Artificial landfills were dominated by sand, sandy loam and clay, whereas original soils were clay soils. 88 samples 0–10 cm and 15 profiles were taken together with local grasses and herbs, and digested with aqua regia. The average humus content of the profiles ranged between 0.6 and 1.9%, and a gradual downward decrease was interrupted in disturbed profiles. In addition, some profiles were highly carbonaceous from loessy parent material and artefacts from building materials, with gradual downward increase of carbonates. Some profiles were slightly saline (0.05–0.15%), and $pH_{KCl}$ averaged

between 7.3 and 8.1. Total N correlated with the humus content. Cu, Ni, Pb and Zn were enriched in the <2 mm component (versus > 2 mm), reflecting a human origin, contrary to Co and Cr, which were assumed to be lithogenic. The vertical distribution of metals in the profiles was largely due to mixing and embedding of other materials. Metal transfers into grasses and herbs were low [17].

**Novi Sad** (Vojvodina, Serbia) has been built on sandy alluvial fluvisols originating from the River Danube, of slightly alkaline pH (7.5 ± 0.1) and some carbonate (11.6 ± 3.5%). 121 samples were taken 0–10 cm, milled and digested with $HNO_3/H_2O_2$. The group of Co, Cr, Mn, Ni and As was found within a range close to background levels, whereas in case of Cu, Pb and Zn concentrations some hotspots occurred, which were highly intercorrelated also. Mapping of Cu Pb and Zn yielded patterns similar to those of the vicinity of major roads. Pb within the range of 200–300 mg/kg was traceable to the use of leaded petrol, which was still in use in the time of sampling in 2010. A Pb hotspot from an accumulator plant could be traced, whereas the origin of 3 Cu- hotspots remained unclear. In spite of an oil refinery and a thermal power plant just 3.5 km away from the city center, Ni levels remained low (V was not measured) [72].

The city of **Zadar** (Croatia) is situated within an area of limestone of low Mg content. Topsoil samples 0–10 cm were sampled parallel to the coast line near the shore of the city, sieved <2 mm, and digested with reverse aqua regia in presence of bromine, in order to dissolve Pt-metals also. The soils were of terra rossa type and differed largely from the marine sediments close by. With respect to mean soil reference data and agricultural soil, urban soils from Zadar were higher in Be, Cd, Ni, Sn and Tl, and contamination hotspots of Cu, Pb and Zn occurred. Whereas in urban soils enriched TOC, Ca and Pb formed a cluster, agricultural soils in suburban regions contained a cluster of Cu, Zn, and P. Pt, Ir, Au, Tl and Bi in the topsoils were found at ambient levels and showed symmetrical frequency distributions [73].

Within a scientific collaborative project, urban soil of **Rome** (Italy) and **Novi Sad** (Serbia) were compared. Pb in Rome was 8 times higher in Rome than in Novi Sad, and in Rome also some Pt (3–15 μg/kg) and Pd (4–16 μg/kg) was found, due to the introduction of catalytic converters some years before sampling, whereas this was negligible in Novi Sad. Samples from Novi Sad, however, despite their lower total metal concentrations, exhibited higher mobilities, presumably due to different soil parent mineral phases. Rome urban soils originate from volcanic rocks characterized by high natural concentrations, whereas Novi Sad soils mainly derive from Danube River sediments and loess. Main additional metal input in Rome is from vehicular traffic, but in Novi Sad from previous industrial activities [74].

**Torino** is an Italian city of about 1 million inhabitants, situated at alluvial plains formed by 4 rivers originating from a mixture of rocks, including ultramafic rocks with serpentinites. 70 sites were sampled 0–20 cm among parks and roadsides, as well as 92 sites from surrounding rural locations, sieved <2 mm and digested with aqua regia. Whereas the ranges of soil pH seemed about similar between urban (pH 4.7–7.8) and rural (pH 3.7–8.0) locations, urban influence increased the median pH from 5.3 outside to 7.4 in the city. However, range and median organic carbon and clay size were about equal, and rather low. Versus rural surroundings, the pollution index in the city for Pb averaged 7.5, for Zn 2.9, for Ni 2.8, and for Cr 2.0. Pb, Cu, and Zn seemed to be mainly associated with traffic, from leaded fuel as well as from brakes and tires. The utmost enrichments were observed in case of Pb, the urban data of which were strongly positively skewed. High levels of Cr and Ni derive from a relevant natural contribution, as well as from metallurgy and car factories. Binary correlations in the urban samples group Cu-Pb-Zn and Cr-Ni together, but in rural soils only Cr-Ni [6].

The Italian city of **Salerno** houses 138,000 inhabitants upon 59 km$^2$, at the coast of the Mediterranean Sea. Beneath an important seaport, the economy of the city is based on services, tourism, pottery and food production. Alluvial soils occur along the coastline, inland hills are covered by soils developed on calcareous conglomerates, and volcanic soils prevail in the northwest area. Surrounding mountains are mainly from carbonate rocks. 151 topsoils were sieved <2 mm, digested with aqua regia, and submitted to multi-element analysis. Al, K, Fe, and Ti mostly reflect the underlying geology, showing the

highest values in the area of pedogenized tuffs, and Ca was found preferably in soils upon limestones and conglomerates. Highest Na and P were found in alluvial sands and recent flood deposits. A lot of elements, like B, Ba, Bi, Ga, La, Mn, Mo, Sc, Te, Th, U, and W showed spatial distribution patterns consistent with local lithology. Top Th, La and Ga were found in tuff and pyroclastic covered sites. Tl, Be and Sn levels were rather high, but no anthropogenic source could be identified. High Sn was found in the city center, and top Cu in the north-east and the most urbanized areas. Hotspots of Pb, Sb and Zn occurred in urban and industrial areas and sometimes close to main roads. The distribution of Hg, Ag, and Au was similar to traffic related elements Pb, Sb, and Zn. Factor analysis combined Sn-Pb-Sb-Cu-Zn-Cd-Hg as factor 1, which was highest in the most populated areas and in correspondence with the road network. Factor 2 contained Co-Mn-Fe-Ni-Cr-As-V-Se, representing an input by natural processes. Factor 3 focused on Mg-U-V-Ca, and was assigned to the input of volcanic material, whereas factor 4 with K-Na versus La-Be was explained by the marine influence [75].

Topsoil samples 0–10 cm from **Palermo** city (Sicily, Italy) were collected from green areas and public parks, where apparently no pesticides or sewage sludges had been used during the last 10 years, and compared with the results of screening of the "natural"soils of Sicily. Local soils have been developed on Pleistocene calcarenites or recent alluvial sediments derived from the erosion of calcareous mountains. The topsoil mineralogy is dominated by calcite, dolomite, clay minerals, feldspars, and quartz. Samples were sieved <2 mm, main elements were determined by XRF, but trace elements after aqua regia digestion. Topsoil pH ranged within 7.2–8.3, due to 7–69% $CaCO_3$ (average 32%), and organic matter within 3–25%. Concentrations of Hg, Pb, Zn, Cu, and Sb were 5–10 times higher than met in natural soils of Sicily, which ranged within other smaller European cities. Co, Cr, Mn and Ni, however, were at low levels, and correlated well with Si, Al and Fe, indicating alumino-silicate origin. In addition, clustering discriminated V-Ni-Mn-Co-Cr-Cd (alumino-silicate phase) from Cu-Zn-Sb-Pb-Hg (anthropogenic input), which was additionally verified by principal component analysis. Cd might have partially derived from natural enrichments in Fe-Mn nodules present in the Terra Rossa soil, which is common at those locations. Spatial distributions of Pb, Zn, Sb and Hg showed particular enrichments at the historical city center and close to the crossings of major urban traffic axes. Additional point sources were laboratories for Hg, and fire works for Sb. Fertilizers enhanced P-levels about 4–5 fold with respect to natural background soils, and additional input of Hg-Pb-Cu-Zn was indicated by significant correlations with P [31].

The city of **Vigo** in northwestern Spain has about 300,000 inhabitants, an average temperature of 14.5 °C, and 1450 mm of annual precipitation, surrounded by granite and gneiss rocks. It houses a car manufacturing plant and dockyards at the sea. 36 soils were sampled 0–20 cm, at sites of no apparent changes for the last 10 years, sieved <2 mm, and analyzed by XRF. The data were compared with background values obtained from the Galician soil survey, to obtain appropriate enrichment factors. The $pH_{H2O}$ ranged between 4.30 and 7.79 (average 5.87) due to acid rocks, higher values were found close to the beach because of calcareous mollusk shells. Garden-tending processes raised pH, organic matter, N, P and cation exchange capacity in urban soils compared to the rural environment, because of intense addition of fertilizers, pruning waste, organic residues, construction waste, and vehicle emissions. The average enrichment factors were 2.64 for Ba, 2.43 for Cr, 1.91 for Cu, 2.07 for Pb, and 1.73 for Zn. Mn and Ni were found at lower levels than the regional background. Mapping of spatial distributions showed Ba and Pb mainly associated to transport routes and to industrial areas. Local Cr seemed traceable to wooden fences treated with chromate as a preservative. Mn, Ni and Zn had no hotspots. Clustering resulted in a lithogenic group with main elements Na and Si, a mixed group with Ca and Sr, a second lithogenic group Mg-Cr-Fe-Mn-Ni, and an urban group Ba-Cu-Pb-Zn. The metal levels in Vigo were lower than obtained from other Spanish cities, which had been sampled 10 years earlier, however [12].

3.4.2. Asia

**Tyumen** is a city within an intensively developing urbanized area in Western Siberia on the border of the sub taiga and forest steppe zones, doubling its population during the last 30 years, where now 770,000 people live at about 700 km$^2$. Industrialization started during World War II with metal working, chemical, consumer and food industries, and later turned to industries related to developing the oil and gas fields of Western Siberia. 241 samples were taken at 0–10 cm depth, sieved to <1 mm, and determined by XRF. Soil pH varied greatly within 4.8–8.9, and increased towards the city center because of alkalization from atmospheric dust inputs and construction debris enriched with carbonates. In addition, organic carbon varied widely (0.1–22.8%) due to widespread peat soils in the northern marshy part. Maximum silt and clay were observed within agricultural areas, and maximum sand in floodplain soils. Fine particles contained Fe and Al hydroxide coatings, whereas Si was predominant in the sand fraction. Zn, Sr, and Pb data were not normally distributed, some samples contained higher levels. To the contrary, the distributions of V, Co, and Cr were close to normal, and Cr had some natural enrichment. Moderate pollutions were indicated for Cu and Co at some hotspots. Rather low levels of Zn and Pb could be explained by the absence of heavy industries and rather late automobilization. The spatial distributions of Ni, Cu, and Zn were generally similar. In the low-populated and mainly agricultural northern part of the city, low levels of V, Cr, Co, Ni, Cu, and Zn were observed. Arsenic and Pb hotspots were found near the automobile starter enterprise sites. With respect to soil main component, Al and Fe correlated strongly positive with V, Cr, Ni, and Cu, and moderately with Co, Zn, As, and Pb. V and Co rose with increasing clay content, and Ni, Cu, and Zn with increasing organic carbon. From principal component analysis, component 1 contained high loads of V-Cr-Co-Ni-Cu, component 2 of As-Pb, and component 3 of Sr, whereas Zn was intermediate between 1 and 2. Particularly Zn and As were at maximum at the old historic center. High concentrations of Co, Ni, Cr, Cu, and Pb were typical for areas formed in about 1949–1960. A sharp decrease of Pb content was associated to the use of unleaded gasoline [64].

Around **Seoul** (Korea), soils of two rather newly built representative satellite cities had been investigated in detail. The original soils are of alluvial origin and well oxidized, with no specific geogenic metal enrichments. Topsoils of 0–15 cm were sampled from roadsides, paddy sites, dry sites, and adjacent forest soils. The climate of the East-Pacific regions is characterized by humid summers, receiving half of the annual rainfall within June-July-August. Average pH values in forest, paddy, dry-field and roadside soils were 5.2, 5.7, 6.0 and 7.6 respectively. Mapping showed rather uniform contamination for Pb and Cd, whereas industrial areas caused hotspots for Cu and Zn. Urban dusts were significantly higher in Cu, Pb and Zn than levels encountered in soils, thus making an important source of input. After the rainy season, metal levels in dusts decreased, but there was no significant variation in soils. In order to elucidate speciation and mobility, a sequential leaching sequence was also applied, see below [58,76].

**Shenyang** had a population of over 7.2 million (at the time of sampling in about 2010), and is an emphatically constructed city in China, housing a complete industrial system with all necessary departments. 36 surface soils 0–5 cm were collected from different main functional sections, sieved <1 mm, and digested with HNO$_3$/HClO$_4$. Concentrations of Cd, Cu, Pb and Zn varied widely, about 100-fold, but the average was only 2–3.6-fold above local background. The dataset was extremely asymmetric containing many moderately polluted (pollution index 2–3), and a few heavily polluted sites. Maximum pollution index was reached for Cd at 13, for Cu at 11, at 24 for Pb, and at 6.8 for Zn. Partition into districts facilitated data interpretation. Similar spatial distributions of Cd, Pb and Zn were found. The most serious contaminations occurred in the Tiexi District, which is the oldest and largest industrial zone in Northeast China, housing 1200 factories of all kinds. Industrial activities, coal combustion and refuse incineration were the main sources of metal emissions. Cd, Cu, Pb and Zn were strongly intercorrelated, indicating related sources. The lowest levels of Cd, Cu, Pb and Zn were found in residential areas, public squares, suburbs and parks [77].

**Beijing** has a monsoon-influenced climate, characterized by hot humid summers and cold dry winters, with an annual precipitation of 500 mm. Beijing is located in an alluvial plain with effective clay and carbonate contents. In order to investigate the impact of main roads, 80 top soils 0–20 cm were collected 1 m away from 10 main roads, and the fine soil part digested with $HNO_3/HF/HClO_4$. The soil pH ranged from 7.79 to 8.80, with a mean of 8.39. Clustering parted the investigated elements into 2 groups, Cd-Cu-Pb-Zn, and As-Cr-Ni, of which the first one was affected by traffic volume. Soil concentrations of Cd, Cu, Pb and Zn were higher than corresponding background values, showed a decreasing trend with distance from the road, and were positively skewed because of some hotspots, whereas As, Cr and Ni followed normal distributions. Black carbon ranged within 0.03–0.75% (mean 0.25%), and correlated significantly with Cd, Cu, Pb and Zn. As, Cr and Ni correlated positively versus grain sizes at <0.001 mm, but negatively at grain sizes 1–0.01 mm. Compared with other cities, Cu-Zn-Ni ranged middle, and Cd and Pb rather low. Rather high Cr might be due to HF digestion [78].

In Beijing, 21 million inhabitants can use 381 urban public parks as recreational areas, some of them had been former factory sites. 121 park soils were samples 0–5 cm, sieved <0.125 mm and digested with $HNO_3$:HCl= 1 + 1 for trace elements determinations. Hg and Cd were all, and Cu-Pb-Zn-As were largely above background values. Pb-Zn-Cu-Hg were strongly intercorrelated, whereas Cd and As were rather independent, and Cr and Ni were found evenly distributed at background levels, except for some Cr-hotspots in the northwest due to some former metal processing plants. Areas with high Pb and Hg were located in the city center and had hotspots at different locations. Zn and Cu hotspots occurred in the center and northeast. In the northeast, some areas contained high As, whereas Cd was mainly accumulated in the southwest. The hazard index followed the order As > Cr > Pb > Hg > Ni > Cu > Cd > Zn, but was less than 1 in all cases. Pb and Cu originate mainly from exhaust emissions, and Zn from tires. Hg in soil was closely related to the duration of urbanization and increased with the age of the park. Hg gets emitted from fuel combustion (coal), from Cu-Zn smelting as well as from waste incineration, whereas main source of Pb pollution is surely traffic. Zn and Cu decreased with increasing distance from the city center. Sources of As are coal combustion as well as use of As-containing pesticides, herbicides and As-containing phosphate fertilizers in former farmland [79].

**Xi'an** is a city of 7.14 million inhabitants in northwest China, receiving 500–700 mm annual precipitation and 13 °C of temperature. Beneath tourism, the city lives on aerospace products, petrochemical industry and high-tech manufacture. 62 surface soils 0–10 cm were sampled in industrial areas, mixed commercial and traffic areas, residential areas, educational areas and parks, sieved minor 0.075 mm, and analyzed by XRF. With respect to the element background values of Shaanxi Province soils, all analyzed trace metals exceeded their corresponding background values. Elevated levels of Co and Ba were found in industrial areas, Cu, Zn and Pb levels were rather high in mixed commercial and traffic areas, but largest Cr and Mn were observed in a residential area. The pollution index for Co reflected a moderate pollution level, and the others slight pollutions, but hotspots of Pb, Zn, Cu, and Co implied heavy local pollution. Principal component analysis grouped Cu-Zn-Pb, Ni-V-As, Cr-Mn, and Co-Ba together [1].

**Xuzhou** in Eastern China is a center of historic and modern-day industrial activities based on coal. Low wind velocities often occur, which enhance the deposition of emitted particulates in the urban area. 21 topsoils of 0–10 cm were collected mainly on bare urban soil, sieved <2 mm, and separated into various grain sizes, and analyzed by XRF, resp. after HF-HCl-$HNO_3$ digests, for 28 elements. The soil pH ranges within 7.51–8.49, organic matter within 2.5–11.2% (median 7.45%), and clay content within 2.3–12.7% (median 7.0%, and silt was the main grain size fraction. Al, Ti and Co correlated positively both with pH, organic carbon, and percentage of clay. The soil pH correlated positively further with Ba, Mn, Pb, Fe, Li, V, Cr, and Be, the organic carbon further with V, As, Hg, Sb, and Sn, and finally the percentage of clay also with Fe, Mn, V, Ga, Be, and Li. Most of the total element data followed a normal distribution, apart from hotspots of As-Sb-Hg-Cr, and others of Ni-Au. Trace metals including Pt and Pd were found at a rather modest level, except for some Cd, Zn and Sn. The element group of As, Hg, Bi, Cr, and Sb showed similar spatial distributions and strong intercorrelations, indicating a common

source from a polluted zone at the industrial center of the city. Zn, Cu and Pb spatial distributions coincided with the main roads, probably due to vehicle emissions. Al, Ti, Ga, V, Co and Mn were evenly distributed and reflect the composition of soil parent materials [80].

**Hangzhou** in Eastern China has a subtropical climate of 16.5 °C average temperature, 1455 mm annual precipitation, at an altitude of 3–6 m above sea level. The soils had developed from marine sediments with low concentrations of heavy metals. 182 soil cores were taken 0–10 cm, sieved <2 mm and digested with $HF/HNO_3/HClO_4$. Wide concentration ranges were found. The soil pH increased from parks to residential areas to roadside soils to industrial sites. Roadside locations tended to contain more organic carbon. Industrial soils contained 40% more Fe on the average, with respect to the level found in parks. Mean Cu, Zn and Pb exceeded the background threefold and Cd even 7.4-fold, and additional hotspots occurred, particularly for Zn. The data were not normally distributed, there was some tailing towards lower values. The levels in Hangzhou were lower than in Shenyang and Luoyang, but about the same level as found in other cities in Eastern China. The metal concentrations decreased from industrial areas > roadside > residentials > parks. Cd, Cu, Pb and Zn were closely correlated indicating similar pollution levels and sources. The pollution index increased from Cu < Zn < Pb < Cd, but at different amounts at industrial and residential areas [23].

Within a series of urban soil monitoring in Hangzhou 10 years later, and sampling 0–30 cm, Mn, which was comparatively high, and total heavy metal content showed similar spatial distribution trends. Hg and Ni were enriched in the east, indicating the petrochemical industry as the same source. Co and Cr distributions were relatively uniform, they might have come from a natural source. Pb and Zn were associated with busy local traffic and the airport. Enrichments of Cd and Cu were traceable to local heavy equipment industry. Compared with sampling at 10 years before, particularly Cd, but also Cu, Pb and Zn were much lower, which can be explained by less pollution or increased sampling depth; others had not been investigated earlier [63].

**Shanghai** is the largest and densest populated city in China, where more than 17 million inhabitants lived at the East coast at the time of sampling, in a subtropical monsoon climate with an annual rainfall of 1122 mm and an annual average temperature of 15.8 °C (in 2019: already 25 million, and 17.6 °C mean temperature). At this location, paddy soils, fluvio-aquic soils, coastal saline soils, and yellow-brown soils occur. At 273 sampling sites within the urban area, which covers 11.4% of the total city, topsoils 0–10 cm were taken, as well as roadside dust samples close by. The samples were sieved <0.125 mm and digested with $HNO_3$-$HF$-$HClO_4$. The mean levels of Pb, Zn and Cd were more than 3 times higher than the background, Cu and Cr at less extent, but not Ni. Due to the short history of industrialization, Pb levels were lower than in old European cities, whereas Cu and Ni levels were about the same. In Shanghai urban soils, only Cr and Ni data were normally distributed, but others approached a log-normal distribution. Zn and Cu in soils, and Pb and Cu in roadside dust demonstrated only weak spatial autocorrelations, whereas Cr, Cd and Ni showed medium spatial autocorrelations. The spatial distribution patterns of Pb, Zn, Cu, Cr, and Cd in Shanghai urban soils were generally similar, whereas the Ni distribution pattern differed from the others, indicating a different main source. Levels in both soils and dusts were lower at the eastside of the Huangpu river, which developed only after 1990, whereas the historic center is situated at the west side. Principal component analysis of soil data grouped Pb, Zn, Cu and Cr mainly into component 1, Cd into component 2, and Ni into component 3. Though the proportion of industrial output in urban Shanghai declined from 70% in 1983 down to 12% in 2005, because it moved to suburban or rural areas, metal emissions were still appreciable due to atmospheric transport [35].

In **Hong Kong** territory, about 7 million people live at 1067 $km^2$, but Hong Kong island itself is only 80.3 $km^2$ [81]. Major parent materials in Hong Kong are granite, granodiorite and tuff volcanic rocks with low metal concentrations due to strong leaching under the subtropical weather conditions. Industrial activities in Hong Kong have been reduced in the last 20 years (before sampling in 2000), because industrial operations moved to Mainland China [63,82].

At Hong Kong island, screening of topsoils 0–15 cm comprised urban soils, suburban soils and country park soils. The samples were sieved <2 mm and digested with $HNO_3/HClO_4$, which yielded a recovery rate of about only 60% for Al, Pb and Cr from the reference material used. Principal component analyses parted the 236 urban soil samples into 3 groups, Cd-Cu-Ni-Pb-Zn, Cr-Fe-Mg (not completely recovered by aqua regia), and Co-Mn. Cr-Fe- and Co-Mn were also grouped together in suburban soils and country park soils. Similarly, clustering found a cluster Cd-Cr-Cu-Ni-Pb-Zn, which is probably anthropogenic, and Al-Ca-Mg-Mn-Fe-Co as mainly from parent soil materials. Similarly, in the suburban soils, data from Cr-Co-Ni-Al-Mg-Fe formed a cluster probably from natural material, and Cd-Cu-Pb-Zn-Ca-Mn from mixed sources. In country park soils, element data grouped as Cd-Cr-Ni-Fe-Al, and Co-Cu-Pb-Zn. Though it can be assumed to be geochemically similar, mapping showed that Co-Ni and Cd-Zn data had different patterns of occurrence. Soil pollution indices marked the oldest residential and commercial areas, but also local traffic areas, as hotspots [63]. Leaded petrol had been banned in Hong Kong in 1999, about 5 years before sampling. The $^{206}Pb/^{207}Pb$ and $^{208}Pb/^{207}Pb$ isotope ratios of the urban, suburban and country park soils formed a linear line between the natural parent rocks and the lead additives from the Australian ore used for fuel additions in the past [81].

Urban park topsoils showed significant enrichments in Cd, Pb, Cu and Zn versus country parks, but at generally lower levels, except for Cd. As a possible Cd-source, phosphorus-fertilizers were assumed. Higher metal levels occurred in commercial areas, which are situated in older urban districts, whereas roadside dusts indicate the short-term contamination released from vehicles. Higher wearing rated of tires at high subtropical temperatures might contribute to rather high Zn levels in Hong Kong street dusts [82].

**Danang** city and Hoian are ancient neighboring towns in Vietnam. Their soils have developed from fluvial and marine Cenozoic sediments, in a tropical climate of 24–26 °C mean temperature and an annual mean rainfall of 2087 mm. Rural, urban, crop and industrial soils were sampled 0–20 cm at 35 locations, and at 20–50 cm resp. 50–70 cm at 15 locations, separated into various grain size fractions, and digested with $HF-HCl-HNO_3$. Grain size patterns were homogenous with about 75% sand and less than 5% clay. In urban soils, Cd, Co, Pb, and Zn were 2–4 times higher than in rural soils, but not substantial accumulations of Cr, Ni, V and Zr in urban soils were noted. In crop soils, Zn was doubled, and Cu-hotspots were found. Industrial soils contained the highest values of Pb, Ni, Zn and Cu, whereas Co, Mn and V levels were like at rural sites. Cd, Cr, Mn, Ni and Pb were enriched in the clay fraction, but Cu, Y and Zn in the fine silt (2–6 µm) [83].

**Bangkok** with more than 8.3 million inhabitants is situated only a few meters above sea level, and suffers from high groundwater level, frequent inundations at the end of the monsoonal period, and thus from soil leaching into the groundwater. 15 soils 0–5 cm were sampled along two transects, sieved <2 mm, and grains 2–20 mm were mechanically separated into a surface and a core fraction. All soils were gley soils, derived from deposits of both natural and anthropogenic material, a clayey substrate from paddy soils, and quartz sand. All soils contained black soot-like organics from incomplete combustion of fossil fuels, and a soil pH of about 6. Sites contaminated with Cd, Cu, Pb, and Zn were concentrated in the southern part and at the industrial branch of the sampled transect. Rather low contaminations can be explained by short accumulation time of the young city, recent deposits of non-contaminated materials, and frequent washouts. Except for Cr and Ni, the datasets were not normally distributed, but principal component analysis could be performed with the log-data. It distinguished Al-Fe-Mn-Cr-Ni as dominated by the parent material and clays, from Cd-Cu-Pb-Zn, as from probably anthropogenic emission [84].

**Sangareddy** district of Telangana state, south central **India**, is an intensively developing region, where urban population and corresponding traffic had doubled within a few years before sampling. The town has been built on lateritic soils based on basalt, receiving an annual rainfall of 910 mm, and a monthly mean temperature within 22–33 °C. 20 soils were samples at 0–10 cm, sieved <0.074 mm, and analyzed by XRF. In particular, the means of Cr and Cu were much larger than their corresponding

geochemical background values, and their spatial distribution patterns were similar. Higher Zn levels were found adjacent to the main highway and the bus station. Cu, As, Ni, and Pb showed moderate contamination levels, while Zn and Cr showed high ones, which presumably reflects the impact of the 21st century. The first principal component contained high loads of As and Pb, and the second of Cr [85].

### 3.4.3. America

**Ottawa** houses a population of about 350,000 inhabitants upon 110 km$^2$ in south-eastern Ontario, Canada, and low presence of heavy industry. Street dust and garden soils 0–5 cm were sampled in winter 1993, and the fraction 0.1–0.25 mm sieved and digested with $HNO_3/HClO_4/HF$. Presumably due to losses of soluble species by runoff and municipal street cleaning, the concentrations of many hazardous trace elements (like As, Cd, Hg, Ag, Tl) was generally low, and not higher in street dust than in the adjacent garden soils, just Cu, Sb and Mo seemed enriched in the dust. Main elements representing the geochemical background matrix, however, were different from garden soils [86].

In **New York City**, about 8 million inhabitants live upon 782 km$^2$, municipal parkland comprises 110 km$^2$, annual precipitation is 1140 mm, and mean temperature is 12 °C. Its complex geology includes crystalline bedrocks, sedimentary rocks, coastal plain sediments, glacial deposits, postglacial materials, and serpentinite. Landfills occupy 20% of the surface area, containing primarily combustion residues originating until 1954–1957, and since than un-combusted organic matter [20]. Release and transport of metals within the soil profile can hardly be evaluated because of unknown history and replacements. The area had been affected by over 100 years of urbanization. An old landfill in the Central Park had been covered by topsoil, and showed elevated total concentrations of trace metals and P. At other park study sites, the B horizon contained lower but notable amounts of trace elements, suggesting some metal movement within the soil profile, whereas the C horizon was in line with values for igneous and metamorphic rocks. At the same sites, Hg, Pb, Cr and Ni decreased with depth, whereas at others, they reached a maximum at 20 cm. In particular, Pb levels were high compared with other urban soils, but other trace metals were moderate because of high rainfall and subsequent runoff. Pb had hotspots generated by short-radius cluster processes. High P-levels might be due to frequent application of fertilizers, and failure of re-vegetation. High V derived from industrial dust and burning fuel oils. At the relatively undisturbed Van Cortlandt and Pelham Bay Park sites, the data obtained 1999 versus 1989 showed elevated Cd, Co, Cr, Pb and Zn for the surface horizon, suggesting respective deposition and accumulation over time. Zn inputs were related to tire wear. Though Pb concentrations in ambient particulate matter had decreased since the ban of leaded gasoline, Pb enrichment remained through re-suspension and sink [20].

Due to the sandy nature of **Florida** soils, metal concentrations in Florida soils are lower than in other parts of the US. **Urban soils** from Orlando and Tampa, as well as 4 smaller cities in Florida were sampled at 107 commercial sites and 107 public sites for 0–15 cm depth. Commercial sites include restaurants, shopping malls, supermarkets, cinemas and banks, while public sites include government facilities, courts, libraries, museums, and schools. Samples were sieved <2 mm and digested using $HNO_3/H_2O_2$. Compared with mean crust levels, Co was generally depleted, whereas some top values of Pb, Zn, and Se were found everywhere, and top Cr values in Ocala. High Pb-levels can be explained not just from the use of Pb-gasoline, but also of Pb-paints, which had been banned in 1978 (=40 years before sampling). Ba showed high positive correlations with all metals and Se, except Cu, whereas As did not correlate with others. Strong associations of Cu, Pb, and Zn may indicate similar contaminations sources [87].

In **Mexico** City area, about 20 million inhabitants live upon 1200 km$^2$, at an annual mean temperature of 16 °C and an average annual rainfall of 660 mm. The northern part is the industrial center, the central parts is the historic and socio-economic center, and the southern part is dominated by residential activities. 146 topsoils 0–10 cm were collected by grid sampling, sieved <0.074 mm, and analyzed by X-ray-fluorescence. High levels of V, Cr, Ni and Ba were presumably due to andesite

host rocks. High levels of Pb, Cu, Zn and Ba in the northern part appeared to be related to industrial activities. Significant correlations appeared between Co, Cr, Ni and V, which was ascertained by cluster analysis. These elements had similar distribution patterns without distinct hotspots. Another cluster included Pb, Cu, Zn and Ba, which peaked at several spots in the industrial and central part of the city, related to traffic activities. Pb-gasoline had been used in Mexico till 1989, which was 19 years before the actual sampling period. With respect to a similar sampling campaign 5 years before, Cu and Zn increased, while Pb, Cr, Ni and V remained at about the same levels. Maximum Cu and Zn was found close to high density traffic roads, presumably Zn from tires and Cu from mechanical car parts [24].

The **Cuban** urbanization process is one of the oldest in the Americas. 77% of the total residents in Cuba are urban population. In **Havana**, soil uses are urban agriculture, parks, gardens, wooded areas, and vacant areas. Urban agriculture is based on locally produced organic fertilizers with agro-ecological pest control and local seed production. The climate of Havana is tropical with a wetter and a drier season. The geology of the area is mainly sedimentary limestone and ultramafic rocks. Some of the agricultural soils have a surface layer made of compost from various organic wastes. The soils were carbonaceous with a pH of 8.27 ± 0.20. They were sampled according to morphologically different horizons, sieved <2 mm, digested with reverse aqua regia, or selectively extracted with Mehlich 3 solution or oxalate buffer pH 3. Maximum Hg was found in Habana Vieja. 4 principal components explained 78% of variance, which contained after rotation mainly Fe-Ti-(-As)-(-Hg), Ni-Cr-Co, Pb-Zn-Cd-Cu, and Mn. The pseudo-total concentrations were not significantly affected by different land-uses, but Cd-Zn-Cr-Ni-Cu were enriched in surface horizons, and Co-Cr-Fe-Ni in soils with gleyic properties [38]. With respect to average values of world soils, the majority of As, Cu, Mn, Ni, Pb, Hg, and Zn data were higher, but largely did not exceed the maximum allowable concentrations for agricultural soils, except for some Cr and Ni, which is traceable to the ultramafic rocks of the geochemical background [51]. Geochemical enrichments of Cuban soils with Ni, Cr, Cu, Co and Mn due to their ultramafic rock base leads to calculations of the geo-accumulation indices different from e.g., Europe. The geo-accumulation indices showed minimum enrichment of As, Cd, Cr, Cu, and Ni in most samples, but high enrichment of Pb, Hg, and Zn. Cu may have a partly lithogenic character, because it neither correlated significantly with Pb-Zn, nor with Co-Cr-Ni. Cuban municipal solid waste compost contained elevated concentrations of Cd, Pb, Hg, As, Se, and Ni [51].

### 3.4.4. Australia

In **Sydney**, roadside samples were collected along the city motorways on asphalted pavement by brushing of about 10 cm, were dried and sieved <0.425 mm. Pollution index as well as the enrichment factors declined in the sequence Cu > Zn > Cr > Cd. High enrichment values for Cu and Zn were presumably due to brake linings and tire wear. Less enrichment of Pb might be the result of more than 15 years restriction of leaded gasoline use. Strong positive correlations of Ni-Fe-Cr-Mn both in roadside dusts and soils indicates natural origin. Whereas in soils, Cd-Pb and Cu-Zn were found within the same cluster, in roadside dust it was Cd-Zn and Cu-Pb [36].

Roadside topsoils in and west of **Melbourne** (Australia) were sampled 0–10 cm within a distance of 2–5 m from road edges, sieved <2 mm, and digested with aqua regia. No significant differences for electrical conductivity, exchange capacity, and organic carbon were noted between roads of different traffic density and speed classes. With the age of the road, the pH tended to increase, from 5.2 to 8.4. Maximum Pb was found in old road, maximum Mo in medium high traffic roads. Cr, Cu, Mn and Pb increased with the age of the road, and Mn with permitted speed. Because Mn additions in fuel replaced Pb, the Mn showed a negative relationship versus Pb. Significant correlation between Cr, Cu, Sb and Zn suggest a common source. Zn is also emitted as a result of tire wear [25].

Within a 15 km distance to **Adelaide**, soils 0–10 cm and 10–30 cm were sampled at 12 sites at plant production areas, sieved <2 cm and digested by aqua regia and $HClO_4$. Commercially available municipal green-waste compost and/or horse manure was used at all sites. As and Cd levels were largely below the detection limit of ICP-OES. Cu, Mn, Ni, Pb and Zn were found at ambient levels,

except for some hotspots. At industrial sites, soils from raised beds had higher Cu and Mn, and at non-industrial sites more Cu and Zn, but in general less Pb than local soils. The use of raised beds with introduced soils appears to open an effective way for food production at sites with industrial histories. In deeper soil layers (10–30 cm) versus topsoils, Cu and Zn decreased, whereas Pb increased [14].

As an example of a small town in Australia, urban soils of **Lithgow** were investigated, which is a small rural city situated in a valley with extensive coal outcrops, iron and steel works, a copper smelter, brickworks and potteries in the past. Its geological background consists of mudstones and marine shales with coal-seams, and sandstone and minor shales at the sides. Tetraethyl-lead was introduced in Australia in 1932, and prohibited in 2002. Pb-based paints were banned in 1970, but still particles from their deterioration get emitted [49]. 134 samples were taken within the entire Lithgow area 0–2 cm, and some profiles in addition, sieved <2 mm and <0.18 mm, and determined by XRF, resp. Hg by direct combustion. The surface soils 0–2 cm were enriched with S, Cu, Hg, Pb and Zn with respect to local background and to subsurface soils, which showed a rather constant concentration profile except for Cu at 2 locations. Mapping of spatial distribution showed concentration decreases with distance from the railway line and main street, and also towards the western end of the town. Whereas Cu, Pb, Zn and S were strongly intercorrelated in both grain-size-fractions, Hg was different and slightly increased at the city center. Enrichment at the soil surface indicates input from atmospheric deposition, maybe from burning coal or wood [49].

In **Suva City** (Fiji), 45 road dust and 36 roadside soils 0–5 cm were sampled at 18 sites during fine weather in May, as well as background samples 0–20 cm at 50–100 m distance from the roadside at remote locations. The unsieved samples were digested with aqua regia. In roadside soils, accumulations versus background were noted for Cu, Pb, and Zn, and in road dust enrichments versus roadside soils for Zn, Pb, and Ni. Cd was quite high with respect to an expectable Zn/Cd proportion, even at the background, and to the contrary, Co and Cr seemed nowhere enriched. Ni-input was traceable to vehicle wears, and Cu-Pb-Zn to both industrial activities and traffic, particularly close to the bus terminal. Principal component analysis of roadside soils data resulted in a component of Cd-Cr-Ni-Zn, which might be associated mainly with traffic emissions and tire and brake wear, and a second component of Fe-Co-Cu-Pb from soil and industrial influence. In case of road dust, the dominant principal component contains all the studied metals except Cu (Cd, Co, Cr, Ni, Pb, Fe, Zn), and a second component mainly Cu [88].

### 3.4.5. Africa

Nationwide geochemical maps are still scarce for African countries, which makes it difficult to refer possible pollutions to a local baseline [89].

In **Ibadan**, a city housing about 4 million inhabitants in southwestern Nigeria, ferruginous tropical soils have developed upon the Precambrian basement complex of Nigeria, consisting of various acid and basic silicate rocks. 106 samples were cored 0–15 cm, sieved <0.075 mm, and digested by aqua regia. The mineral matrix was characterized by X-ray-diffraction, which identified quartz and alkali feldspars as the main soil minerals. All element concentrations investigated followed non-Gaussian distributions, i.e., many smaller values plus some extreme hotspots. Among the mean concentrations in the study area, Cd (8.4 mg/kg), Mn (1098 mg/kg), Pb (95 mg/kg) and Zn (229 mg/kg) seemed high, whereas As, Mo, Ni, and Cr were rather low. Enrichment factor and geoaccumulation factor revealed significant enrichments in Cd, Cu, Pb and Zn, particularly at the densely populated and industrial areas of Ibadan metropolis. This could not only be linked to traffic, but also to mechanic workshops, rusty vehicle remains, exhausts from power generators, rusty water pipes, and dumping of metallous wastes (e.g., cans). Mn-Ni-Cr were surely of lithogenic origin, and Cr-Ni as well as Cu-Pb-Zn were significantly positively correlated. Cd hotspots were observed in the agricultural area, as well as at metal scrap collection and plastic manufacturing sites [89].

### 3.5. Levels Met in Roadside Dusts

Road dust is a complex mixture of particles from organics, metals, other inorganics, mold spores, pollen, animal dender etc. Its sources are traffic, industrial activities, powerplants, fossil fuel burning, waste incineration, construction and demolition activities, and resuspension [90]. Topsoils and particularly roadside dusts in urban areas are indicators of airborne particulate matter metal pollution. Most trace metals settle down as surface dust from atmospheric depositions before full incorporation into the soil matrix. Thus, the extent of atmospheric contamination may be better revealed by road dust than by soils [91,92].

Though this data compilation refers to deposited dust only in terms of mg/kg, the amount of re-suspendable particulate matter should be considered in terms of inhalation risk assessment, in particular the grain-size fractions below 2.5 μm aerodynamic diameter ($PM_{2.5}$) and below 10 μm ($PM_{10}$). In the US, the $PM_{2.5}$ has been estimated at about 12% of the total dust, unless detailed investigations are available. Therefore, a detailed study of grain-size distributions was done in various roadside environments in Albuquerque NM, Atlanta GA, Birmingham AL, El Paso TX, Los Angeles CA, New York NY, and adjacent New Jersey, after sampling by a high-volume sampler and separation by a cyclone dust trap. The average composition of paved road dust from industrial sites differed from other urban regions by containing a larger fraction of elemental carbon and less-iron crustal material (Al-Ca-Si), due to coking facilities or high local emissions of trucks, trains and ships. There was slightly more diversity among urban regions than among urban site types. The portion of silicates + carbonates, as well as the $PM_{2.5}$ content, was greatest in arid regions. Among 51 species determined by XRF and ion chromatography of water-solubles (detailed data not given), nine variables were identified as most important in distinguishing samples: Cr, Ni, Al, Cl, Ca, Si, Fe, Cu, and nitrate. Paved road dust had a higher content of potentially bioreactive metals and higher elemental C at urban than at rural sites [93].

Contrary to direct air dust sampling, deposited dust yields integrated contamination values over a long period of time, usually back to the last rainfall or street cleaning action. In addition to chemical toxicity, however, some particles promote catalytic reactions at their surface, which is not traceable by chemical analysis only. Direct health implications are due to the inhalable fraction, which is rather floating in the air than depositing at surfaces. Deposited dust directly imposes health hazards via consumption of unwashed food, particularly for babies who like to lick around, for dogs which like to lick and sniff at dusty surfaces, and finally the dust-washout might harm green plants in parks, as well as river systems [94].

Permanent changes in the technologies for house construction, traffic and heating during cold seasons, as well as changes and improvements in the purification of industrial emissions, change the composition of urban dust on a long term. Within a first step, representative sampling grids have to be established to find the hotspots of contaminations, prior to rather laborious and expensive size-fractioned sampling and microlocal analysis. Contrary to direct air dust sampling, deposited dust yields integrated contamination values over a long period of time, usually back to the last rainfall or street cleaning action [3].

The extent of contaminant input from traffic activities is generally determined by comparing concentration data of road deposited sediments with those of nearby uncontaminated soils [36]. Comparisons between road dust samples from these largely different urban areas might reveal common global urbanization effects, and specific rather local influences. This can be ascertained from additional data per gram dust, available from other densely populated areas. The effect of urbanization might be traceable from differences between road dust samples from Budapest and from more rural Hungary [59].

Vehicular traffic contributes to dust emission by release of tire and brake pads wear, releasing fibers (like copper, steel, potassium titanate, glass, organics), fillers (barium and antimony sulfate, Mg and Cr oxides, ground slag), lubricants, and abrasives. However, street dust can also contain up to 60% of particles originating from soil, like quartz, feldspars, clay minerals, chlorite and muscovite. Whereas main sources of Zn and Fe are tire wearing, brake wearing release in non-asbestos brake pads reaches up to 15%, containing Ba, Cu, Fe, Pb, Zr and Sn. Ni and Cr can originate from corrosion of

cars. Technogenic magnetic particles from high temperature combustion processes have characteristic spherical shape, while those from traffic emissions and iron smelting form irregular non-spherical aggregates [25].

A compilation of road dust data easily shows that some elements are always enriched versus upper crust mean values, like As, Cd, Hg, Mo, Sb, and Sn. Though levels in road dust may be higher than background soils, Ba, Mn, P, Tl and V were found within the fluctuation of geochemical composition. Largely, but not always higher than upper crust values are Cr, Cu, Pb and Zn, and occasionally higher are Co and Ni.

In the subsequent part, examples of special investigations made upon urban and roadside soils are given, arranged from North to South.

### 3.5.1. Europe

In **Oslo**, street dust at 16 sites was collected in August 1994 by brushing, sieved <100 μm and digested with $HNO_3/HClO_4/HF$. Comparatively high Na is due to the proximity to the sea. Major sources of Mg and Ca is cement from buildings and construction works. Ga, La, Mn, Sr, Th and Y were assigned to mainly natural soil particles. Principal component analysis could combine the 29 elements determined into 3 main components, containing 70% of data variability. Together with Al, Fe and Na, a group of Ga, La, Mn, Sr, Th, and Y was interpreted as the "natural" elements, which is coherent with the alkaline composition of the most abundant rock formations around. A second group of Ba, Be, Cd, Co, Cr, Cu, Mo, P, Pb, Sb, V and Zn was assigned to be of urban origin. They show a steady increase from suburban and residential areas towards the city center and the harbor, their sources are traffic and building activities. Because Pb level in fuel was just 0.0375 g/L before prohibition, contrary to e.g., 0.4 g/L in Madrid, Pb emissions in Oslo were rather low at any time. Further contributions to urban dust come from cement for Mg and Ca, from paintings for Pb, and from metal corrosions for Zn and Cd. A third group of elements exerted a mixed urban and natural character, because alkalization and oxidation in urban geochemical surroundings lead to changes of mobility and washout [38,95].

In the city of **Ostrava** (ČR), a city of about 300,000 inhabitants living upon 214 $km^2$, more than 400 sources of air pollution occur, particularly from industry and energy production. About 14 kg of street dust per person had been swept off the streets annually. This fallout is influenced by steel works, power plants, and traffic. Street dust was sampled at 150 localities during summer season. The Ca concentration increased with the amount of dust particles at >2 mm, confirming that the Ca source might be road construction or building material. Ti and Zr showed a significant relationship between their concentration and the weight of particles <0.063 mm. Highly variable admixtures of organic pulp-bituminous substances eroded from roads caused high heterogeneity of the street dust samples [25].

Within the 17 districts of Ostrava, between sampling 2008 and 2018, concentrations of Ti and Pb in street dust decreased in all, Ca and Mn in all but one, Cr in 12, Cu in 11 and Zn in 10 districts, whereas Fe increased in 12 and magnetic particle concentrations increased in 14 districts. Decrease of Pb concentration was connected with changes in the metallurgical production processes, because of Pb-trapping in dust from the sintering plants, as well as lower contents in slags. Addition of biomass to coal in the power plant increased Mn and Zn fallout in its immediate surrounding, whereas the hotpots around the steelwork were reduced. Increase of Cu in 6 of the districts was probably influenced by traffic. The origin of Cr was not related to iron and steel production, but to the coking plant [25].

Electron microscopy revealed the presence of microparticles of iron oxides, metallic iron, microparticles of barite, zircon and rare earth phosphates in almost all samples. Cr occurred as metallic Cr as well in Fe-Cr-Ni particles. Fe-rich particles from non-vehicular sources were irregularly shaped grains and spherules of Fe-oxides and metallic Fe, whereas irregular and angular particles derived from abrasion sources of vehicles. Fe-particles from ladle furnace and oxygen converter steelwork reached about 5 μm in size and might contain some % of Zn, because in magnetite, $Fe^{2+}$ can

be substituted with $Zn^{2+}$. Blast furnace slag is composed of Ca-Mg alumosilicates, whereas steelmaking slag contains Fe-oxides plus Mn-oxides plus phosphates [25].

In order to trace anthropogenic enrichment, Ti was considered as the most suitable element to serve as a base for enrichment factor calculations. Magnetic susceptibility showed significant statistical dependence with Fe, Mn, Cr, Pb, Zn, and Ca, but not with Cu. The class <0.063 mm had, at the same Fe-content, approximately double magnetic susceptibility than the class 2–0.063 mm, due to the proportion of Fe in non-magnetic minerals [25].

At busy crossroads in **Prague**, **Česke Budejovice** and **Ostrava**, equipped with traffic lights and intense stop-and-go, road dusts from road centers and road edges as well as topsoils 0–1 m from crossroads were collected. Because of presumably toxic metal enrichment in fine particles, grain size fractions 0.315–0.1 mm and <0.1 mm were selected for XRF-analysis and aqua regia digestions. Sampling occurred 4 times in different seasons and precipitation events.

Fly ash from powerplants are composed similar to aluminosilicates, but may contain $As_2O_3$ in the % range. Brake abrasion dust mostly consists of iron oxides, organic carbon and sulfates, arising from the oxidation process during braking, and contained 4.3% of $Sb_2O_3$ beneath 10.55 of organic carbon. The pH in road edge dust was generally higher than in adjacent roadside soils, measured in $H_2O$ as well as in KCl, though soils were already alkaline. As-pollution had negligible concentration gradients between road dust and roadside soils, because it was due to atmospheric emission in an urban and industrial agglomeration. Sb, however, declined two orders of magnitude within a few meters from the edge of the road, but a positive correlation of both Sb and As with Fe and organic matter was observed. Under dry weather conditions in autumn, an increased Sb content was noted, whereas lower levels were found in winter, because the snow layer acted as a filter for atmospheric particles, protecting the ground from a direct contamination. However, As pollution showed negligible seasonal fluctuations, indicating sources different from traffic [41].

In **Budapest** (Hungary)**,** road dust samples were obtained two times from traffic focal points in Budapest, from the large bridges across the River Danube, from Margit sziget (an island in the Danube in the Northern part of Budapest, used for recreation) as well as from main roads (no highways) outside Budapest. Road dust samples were collected by hand brushing with a nylon brush and plastic collection pan directly from the road surface. The road dust samples were dried at 80 °C for 40 min. and then sieved into <0.075 mm (<200 mesh) fraction. In Budapest street dust, As, Mn, and Ni were at background levels expected for soils, Cd was slightly higher, and enrichments of Mo and Pb were significant. High Pb levels in Budapest might not only derive from use of leaded gasoline in the past, but from Pb-paintings, particularly on the historic bridges. Sb levels in road dusts around Budapest were met at about background levels, some accumulation in the city occurred [59].

In **Murcia** Province (Spain), which soil is based upon alluvial deposits of limestone and marl, road dust at non-industrial sites on the average 79–94% less Pb, Zn, Cu, Cr, Co, Ni than roadside soil 0–5 cm, in the $HNO_3$-$HClO_4$ digest. At industrially polluted areas, however, proportions were highly variable. In addition, less DTPA-soluble amounts (as a proxy for plant available fractions) were found for Pb, Cu, Co and Ni in road dust than at corresponding road dust soils, but more Zn and Cr. That means, sampling of topsoil does not necessarily dilute roadside dust at this location. Road dust sampled in **Murcia** (Spain) in about 2010, was alkaline (pH 7.5–7.9) and calcareous (17–43% $CaCO_3$). No relations were found between cation exchange capacity and organic carbon (2.2 ± 1.2%) nor clay content. All sites under human use showed higher concentrations for Pb, Cu, Zn, and Cd, than the natural surroundings. Apart from various metal inputs, industrial activities increased both organic carbon and salinity in road dust samples [91,96].

### 3.5.2. Asia

Chelyabinsk is a typical Russian city at the Asian side of the Ural, of 1.2 million inhabitants, with developed ferrous and nonferrous metallurgy and high traffic volume, but also food, chemical, and light industry production facilities. 125 road dust samples were collected across the city after at

least one week of dry weather conditions, sieved <1 mm and digested with $HNO_3$, HF and $HClO_4$. The particle size distribution curves were asymmetric and had two peaks. The peak at 180–240 μm could be attributed to traffic-related processes, and the peak at 60–90 μm to emissions from industrial enterprises and thermal power plants. The highest proportion of particles <10 μm ($PM_{10}$) were found in subareas located closest to the two largest metallurgical plants, and the smallest in the residential area. The standard variation showed low variabilities for Al, Co, Fe, Mn, moderate variabilities for As, Cr, Ni, Pb, Sb, and Sr, and high variabilities for Cd, Cu, Hg, and Zn, thus reflecting a heterogeneity of the latter group. Emissions from a thermal power station caused high concentrations of As, Hg, and Sb, where coal is used in addition to natural gas. The element concentrations in dust samples, collected by sweeping at a distance of 50, 100 and 200 m distance from the road, were mostly the same as collected from the road itself. High pollution levels were indicated for Hg, and moderate pollution levels for Sb, Zn and Cu, but in case of Cu not for the residential area [92]. Fine spherical and semispherical particles were formed as a consequence of high T industrial processes. Principal component analysis of the dataset revealed 3 factors accounting for 86% of the total variance. The first one was dominated by Cd-Cu-Mn-Ni-Pb-Sr-Zn, and identified as dust from the steel industry. The second one had strong loadings of As-Sb-Hg, reflecting coal firing. A fourth component contained mainly Cr related to the electrometallurgical plant. There was no principal component assigned to traffic only. Cluster analysis led to the same results [92].

In **Samsun** city (Turkey), where about half a million inhabitants live at the Black Sea coast of Turkey, 52 road dust samples were collected in industrial and residential areas, and digested with aqua regia without sieving. Pollution sources were Cu-smelting, steel and non-steel industries, regenerated plastics, as well as coal heating at the period of sampling. Maximum Cu-Pb-Zn-Co-Cd levels were found in the industrial area close to a copper factory. Cu also showed a rise in the residential area with high traffic. Ni was higher within the residential area, and Zn was comparatively low in residential areas with light traffic. Significant positive correlations among Cu-Pb-Zn-Co-Cd suggested a common origin from the metal processing industries, which was also ascertained by principal component analysis. The second principal component contained Ni and Mn, which might represent a mixed lithogenic and anthropogenic origin [97].

In **Delhi** (India), nine road dust samples were collected by a vacuum cleaner, and sieved minor 0.075 mm. Significant positive correlations between Ni, Cr, Cu and Cd were found, but weak ones of this group with Pb. In addition, Zn correlated poor with other metals. The pollution level was high in case of Cd, and moderate in case of Cu and Zn. Near highways, metal levels were lower than those obtained near the roads in other areas [90].

In **Dhaka**, capital city of Bangladesh, automobile exhausts, particularly two-stroke engines auto-rickshaws, factory chemicals and primitive heating are of great concern with respect to air pollution. In central Dhaka metropolitan area, seven dust samples were taken in the industrial area and Old city with medium traffic, 24 samples in the commercial area with heavy traffic, 33 in a residential area with medium traffic, and six in a residential area with low traffic, sieved <1 mm, and submitted to X-ray fluorescence analysis. The dust samples contained more Ca (about 2%) and more P (700–1500 mg/kg) than background soils, Fe and Al were at the same level. Whereas Pb, Zn, Cu and Cr showed great differences between the four areas, V, U and I abundances varied little. Highest contents of Zn, Cu, and Ni were found in the Old city, however top Pb- values in the commercial area with maximum traffic. The ranges of average contents of Pb (25–54 mg/kg), As (4–8 mg/kg), Zn (65–169 mg/kg), and Cu (14–105 mg/kg) were much lower than in many European road dust samples, which may be due to monsoonal flooding, but dust ingestion is probably higher [98].

In **Seoul** metropolitan city and from two control sites within the suburbs of Seoul, for comparison, road dust and roadside soil samples were collected from six sites of high traffic volumes. Road dust samples were collected by hand brushing with a nylon brush and plastic collection pan directly from the road surface. The road dust samples were dried at 80 °C for 40 min. and then sieved into <0.075 mm (<200 mesh) fraction. In Seoul road dusts, Sb-levels were higher throughout. As and Ni-levels in

Seoul were only slightly elevated, but strong enrichments of Sb and Cd were noticed, the latter was obviously not correlated to traffic intensity. Due to a long period (1993–2005) of the use of unleaded gasoline between prohibition and sampling, or rapid development of traffic and road constructions meanwhile, Pb in Seoul dust was not among top levels, and Na in spite of a small distance to the sea-side, lower than e.g., in Oslo, but higher than at continental sites [59].

Top Ni concentrations in street dusts were found in **Tokyo** at 540 mg/kg [99], whereas in Seoul it was just 62 mg/kg (2005; range 42–109 mg/kg), and in Budapest it was 27.5 mg/kg (2010; range 19.2—49.9 mg/kg) [59]; the latter was within the range of adjacent soils.

In **Shanghai**, roadside dust contained trace metal concentrations even two to three fold higher than the adjacent urban soils. Zn and Cu in soils, and Pb and Cu in roadside dust demonstrated only weak spatial autocorrelations, whereas Cr, Cd and Ni showed medium spatial autocorrelations. In roadside dusts, Zn and Cu had the same pattern than in urban soils, and Ni in dusts was evenly distributed. Principal component analysis of data from the dust samples yielded a component 1, which was dominated by Pb, Zn and Cu, whereas Cr and Ni were mainly found in component 2, and Cd in component 3. High Ni in road dust might be due to petrochemical plants close to respective sampling sites. High Cd in road dust was found close to steel works [33].

### 3.5.3. America

In **Buenos Aires** (Argentina), dust was sampled from pavement edges by brushing, sieved to 4 grain size fractions and digested with $HNO_3$-HCl-HF. Cu, Mn, Mo, Ni, Pb, Sb, and Zn were reported to be traffic related [35]. After stop of leaded gasoline use, Pb wheel weights have been considered as the main Pb-source. In Buenos Aires, Cu and Sn in road dust were one order of magnitude higher than those of local soil and also Cd, Mo, Sb and Zn, but mean Ni was similar to local soil at a level of about 25 mg/kg, and Mn even lower. The cluster Cd-S-Sb was higher in finer fractions in all regions, but there was no general trend of enrichment in smaller grain size fractions, except for Mo. Maximum Cu and S were found in the residential area at a main road of high traffic density, but not in the city center close to a large park, nor close to the La-Plata river with heavy traffic of all types, nor at a zone of mixed residential and industrial areas [35].

### 3.5.4. Africa

Luanda, the capital of Angola, is the home of 2.5 million inhabitants (at the time of sampling in 2002), in a tropical environment of 26 °C mean temperature and 400 mm annual precipitation. Natural soils are marine fluvial sands containing quartz, marl, shell limestone and grey-greenish clays, resp. red sands containing also hematite and goethite. Dust samples were taken within a grid, sieved <0.1 mm, and digested with $HNO_3$:HCl:$H_2O$ = 1 + 1 + 1. Most trace elements seemed to be at ambient levels, and high Ca, Na and Mg was due to shells and the seaside. The trace elements were homogenously distributed across the city, and the emission of the industrial facilities north of the center (oil refinery, cement plant, zinc smelter) seemed to be carried away largely with winds. Extraordinary high levels of Pb and Zn were found, however, probably due to traffic. Principal component analysis assigned Sc-Al-Ga-V-Th-Co-La to a silicate matrix factor, and Pb-Cd-Sb-Cu to an urban influence factor [100].

### *3.6. Risk Assessment*

Pollutants can enter the food chain by dust ingestion, dermal contact, or inhalation as well as groundwater contamination, and may have toxic effects as a consequence. Contaminated soils in cities can pose risks to drinking water supplies, to children at playgrounds and to citizens growing vegetables on private allotments [16,31]. Apart from single element mappings, the pollution index, which is the averaging of the ratios of metal concentrations to the permissible level, yields a rapid view of the overall contaminant situation [76].

The composition and solubility of street dust is influenced by climatic conditions, soil environment, bedrock, and human activities. Resuspension of street dust is the main source of particles < 2.5 μm ($PM_{2.5}$) and <10 μm ($PM_{10}$) in the environment, and thus of health risk [25].

Direct ingestion of soil represents a significant route of exposure for children up to the age of six. Ingestion of dust as a result of hand-to-mouth behavior accounted for up to 50% of a child's daily Pb intake [8]. Metal pollution brought huge implications to roadside venders in less developed countries [88].

Local variations of soil composition might be overcome using the bioconcentration factor, which has been defined as the concentration in the plant over the respective concentration in the adjacent soil [101].

Both background concentrations and risk-based values have been used to establish cleanup target levels [87]. The Swedish Environmental Protection Agency has defined urban background levels as the natural soil metal content together with the normal load of anthropogenic metals, thus including diffuse metal inputs [8]. Dutch target values were taken as limit values for pollutants in soils stemming from road traffic, which were for Cd 0.8 mg/kg, Cr 100 mg/kg, Cu 36 mg/kg, Ni 35 mg/kg, Pb 85 mg/kg, and Zn 140 mg/kg [3].

Additions of tetraethyl-lead to gasoline had been used in the US in 1924 for the first time, and admitted in all states in 1928, in Australia in 1932, and in Germany in 1936. Health concerns and incompatibility with Pt-Pd-Rh based catalytic converters led to gradual banning, e.g., 1989 in Mexico, 1993 in Austria and South Korea, 1995 in Sweden, 1996 in Germany and the US, 1999 in Hong Kong and Hungary, 2000 in the EU in general as well in China + Taiwan and Switzerland, 2003 in Russia, and 2006 in Croatia and Turkey. However, additions and consumptions varied widely between countries. However, as Pb is very immobile, Pb in road dust is still enriched from this period and has to be considered among health hazards.

### 3.7. Mobile Soil and Dust Fractions

### 3.7.1. Assignment of Speciation from Selective Leaching

For decades, sequential extraction procedures have been done to discriminate different mobilities under exchangeable, reducible, and oxidizable conditions (after Tessier [102]), in order to assign traces to carbonates, oxides, and organics and sulfides. Though the real speciation depends on the properties of the trace ion, as well desorption and resorption on the remaining solid, mobilities under different conditions give more insights than a single mobile fraction, or total contents. Currently, the author favors a modified version given in Table 1 [103]. If there is dolomite, or highly calcareous matrix, it will not dissolve completely at pH 5 originally introduced for carbonates. On the other hand, mobility under weak acid conditions is much less from a calcareous matrix, because an acid pH cannot be reached so easily.

**Table 1.** Example of a sequential leaching sequence.

| | | |
|---|---|---|
| (1) | 0.5 M magnesium chloride adjusted to pH 7 | Exchangeable |
| (2) | 0.16 M acetic acid | Carbonates |
| (3) | 0.5 M hydroxylamine pH 2, or in 25% acetic acid | Mn oxide |
| (4) | 0.2 M oxalate buffer pH 3 | Fe oxide |
| (5) | $H_2O_2$-oxidation, then like step 2 | Organics + sulfides |
| (6) | aqua regia, or $KClO_3/HNO_3$ digest | Residual |
| (7) | total amount obtained by $HClO_4/HNO_3/HF$-digestion, or by XRF directly from the solid | |

Based upon Tessier-like sequences, the BCR had certified some reference materials according to a simplified procedure, which omits step 1 and implies steps 2 and 3 only [104]. In case steps 2 and 4 are chosen, however, all Fe/Mn-oxide bound parts will appear in oxalate [27]. Substitution of aqua regia

by an almost saturated KClO$_3$-solution acidified with nitric acid, permits to determine all elements except K-Rb-Cs, but in addition also the non-metals sulfur, boron, iodine, and germanium [105].

Human bioaccessibility by direct ingestion was estimated by a gastric simulation test, consisting of wettening with 0.013 M acetic acid, and stepwise acidification with hydrochoric acid to reach pH 6, pH 4, and pH 2.5 for 20 min each, done at 37 °C [23]. In case of road dust, removal of lipophilic organics, which might cover the inorganic grains, by extraction with ethanol (or else) is recommended beforehand [26].

In case of Cr, Be and V, and to a lesser amount in case of Ni, total digests and aqua regia yield different amounts, and data should be compared with caution. Of course, this is also valid for the main elements Al, Fe, Ti, Mg, Mn, K and the rare earths [27].

Selective leaching shows **Pb** mainly bound to reducible forms in Aveiro, Glasgow and Turin, but in oxidizable and residual forms in Sevilla. Like in agriculturals soils, **Cu** shows a high affinity to the organic (=oxidizable) fraction (Stockholm, Bangkok, Da Nang, Ljubljana, Sevilla, Glasgow, Turin), whereas in Nanjing, 66% were found residual. **Zn** speciation due to sequential leaching is variable and seems to depend on source and soil characteristics, but it tends to accumulate in the fine fraction. Zn was easily extractable from soils of Stockholm and Seoul, mainly residual in Nanjing and Honolulu, but rather equally distributed in Aveiro, Glasgow, Ljubljana, Sevilla, and Turin. **Cd** can be labile up to half, which had been found in samples from Stockholm or Hong Kong [2].

### 3.7.2. Examples from Europe

In sequential leaching of roadside soils in Southern **Sweden**, Cd, Cu and Zn were largely mobilizable with weak acid (0.11 M acetic acid), Cu and Pb largely released in the reducible fraction with hydroxylamine, and Cr-Ni in the oxidizable fraction. Though oxidizable Cu was not the main fraction, it correlated with ignition loss. Sb ranged largely within 1–2 mg/kg, with top values at 8 mg/kg versus a background of 0.46 mg/kg and was found largely refractory [30].

The urban soil samples of **Warsaw** were sequentially extracted with 0.05 M KNO$_3$, H$_2$O, 0.5 M NaOH, 0.05 M EDTA, and 4 M HNO$_3$, to obtain exchangeable, organically bound, carbonates/oxides, and residuals. Means of 21% Cd, 15% Cr and 7% were found as exchangeable with KNO$_3$. According to this sequence, main amounts of Cr, Cu, and Zn were found as residual, but Pb and Mn in the EDTA-extract. Pb pollutants from industrial sources are likely in the form of PbS, PbO and PbSO$_4$, while automobile exhausts contain PbBr$_2$, PbBrCl, and Pb(OH)Br, which can convert or at least found by sequential leaching among oxides, carbonates and sulfates [7].

In road dusts from **Murcia** (Spain), Tessier-like sequential leaching found Pb and Cd mainly bound to Fe/Mn-oxides (hydroxylamine in 25% acetic acid), Cu in urban areas equally to Fe/Mn-oxides and residual, while Cu in industrial areas was mainly residual, and along highways also significantly enriched in the organic (oxidizable) fraction. Zn was found mainly as residual. Salinity (Murcia is located at the seaside) was positively correlated with Cu bound to oxides and negatively with exchangeable Zn, and organic C positively with organic Cu. The Zn/Cd proportions indicated significant inputs of Zn, particularly in the carbonate fraction, whereas Cd levels were rather ambient. Due to the geochemical characteristics of the test sites in Murcia, sequential leaching assigned major parts of Pb to the carbonate and reducible phase fractions, Cu to the oxidizable (organically bound) fraction, Cr as reducible, and Zn as residual. Emissions from industrial activities increased Cr in the more mobile phases, did not affect the distribution pattern of Pb, and affected Zn variably [91,96].

In **Sevilla** (Spain), 35 urban soil samples were collected at 0–10 cm and 10–20 cm in the main public parks and gardens, together with grass samples on site. They were sieved <2 mm and submitted to aqua regia digestion as well as to 0.05 M EDTA/NH$_4$-acetate pH 7 selective extraction. No differences between 0–10 cm and 10–20 cm were found for aqua regia and EDTA pH7 extractable metals, except for Mn and Ni in EDTA, and K in NH$_4$-acetate (without EDTA). The soil texture was not related to the geographical location. Many metals had positively skewed data distributions, except Cr-EDTA and aqua regia Cr-Ni-Fe. Hotspots of Cu, Pb, and Zn were found in the older green areas within

the inner historic quarters of the city, though traffic was largely stopped 12 years before sampling. Between sampling in autumn and subsequent spring at the same sites, decrease for Zn and Mn mobilizable with EDTA, as well as increases in pH and Fe-EDTA were found, due to winter rain or irrigation. Principal component analysis using log-transformed data resulted in grouping EDTA and aqua regia—extractable Cu-Pb-Zn together with organic carbon in factor 1, clay content associated with aqua regia—extractable Cr-Mn-Ni-Fe in factor 2, as well as Mn-Ni-Fe in EDTA negative to carbonate in factor 3, and solely pH in factor 4. Cu and Zn contents in the grass samples were significantly related both to the EDTA and the aqua regia extractable concentrations in the soils, but Cr, Mn, Ni, and Fe showed no relation. The mean concentrations found in the green plants were at about the same levels which had been reported for wild weeds from unpolluted areas [9].

Contrary to soils, the easily mobilizable fraction of nickel found in street dusts could be about half and thus rather high, which was measured in **Sevilla** 1996 [106].

In urban soil samples from **Zielona Gora** (Poland), the proportions of water or HCl-soluble amounts of Cu, Pb and Zn were surprisingly highest in parks, and lowest in industrial areas, due to combined effects of pollution level and pH [11].

### 3.7.3. Examples from Asia, Australia and America

In order to elucidate speciation and mobility, in urban soils (0–15 cm) from **Seoul** (Korea), a sequential leaching sequence of 0.16 M acetic acid, 0.1 M hydroxylamine pH2, oxidation with $H_2O_2$, and aqua regia was applied. Zn was largely mobilizable already with dilute acetic acid, Pb with acid hydroxylamine, and Cu was rather uniformly distributed among the obtained fractions. Whereas the summed total concentrations of each metal in soils and dust decreased, the patterns of sequential leaching varied only slightly, and just the most mobile forms were leached by rainfall [58,76].

Mobile phases in **Hong Kong** road dusts are influenced by high Ca, which was about 5 times higher than in adjacent soils. In Tessier-like sequential leaching, Pb was at 70% associated with carbonates and Fe-Mn-oxides, Zn at 60% with carbonates, but Cd only at 40%, whereas in the soils, Zn and Pb were mainly bound to Fe-Mn-oxides, and more than half of the Cd was found exchangeable. Cu was found at 70% bound to the organic/sulphidic phase within street dust, and at 40% within soils. Thus, metal mobility in street dust was much higher than in soils [82].

In urban soils from **Hangzhou** (China), in the gastric juice simulation test, 21% of Cd, 18% of Cu, 10% of Zn, and 4% of Pb were eluted (Lu, Bai 2010). 0,1 M HCl mobile fractions decreased from Cd at 86% to Zn at 72% to Cu at 60% down to Pb at 42%, which should indicate a long-term mobility [23].

Mobilizable amounts in soils from **Danang** (Vietnam) were investigated within a sequential leaching sequence of 1 M $NH_4NO_3$, 1 M $NH_4$-actetate pH 6, $NH_2OH$ pH 6, $NH_4.EDTA$ pH 4.6, oxalate pH 3.25, and finally ascorbic acid/oxalate pH 3.25. Cu was found at about 40% leachable with $NH_4$-EDTA, and 25% as oxalate-leachable, Zn was mainly mobilizable with $NH_4$-EDTA, and Cr mobilizable with oxalate but not with hydroxylamine, within this given sequence [83].

In **Bangkok** soils, environmental mobilities of trace metals were investigated from a leaching sequence of 1 M $NH_4NO_3$, 1 M $NH_4$-acetate pH6, 0.1 M $NH_2OH$ pH 6, 0.025 M EDTA pH 4.6, 0.2 M oxalate pH 3.25, hot oxalate/ascorbic acid pH 3.25, and $HNO_3/HClO_4$. Whereas exchangeables were low, except for Cd at pH 6, presumably because of washout from frequent inundations, reduction with hydroxylamine released considerable amounts of Cd, Pb, and Zn. These can be assigned to Mn-oxides, and would be mobilizable to the groundwater during floodings. High Pb and Cu were released with EDTA pH 4.6, but appreciable amounts of Cr, Cu, Mn, Ni, Pb and Zn were more tightly bound to Fe-oxides, found in the acid oxalate extracts [84].

In **Sydney**, roadside samples were collected along the city motorways on asphalted pavement by brushing of about 10 cm. The samples were dried, sieved <0.425 mm, and submitted to a simplified sequential leaching sequence of 0.1 M acetic acid, followed by 0,1 M hydroxylamine pH2 and $H_2O_2$ oxidation. Maximum Zn was found exchangeable, Cu organically bound, and Fe-Cr in the residual fractions [36].

In soils from parks sampled in **New York City**, sequential leaching, the predominant fraction for Co, Cr, Cu, and Zn was residual (41–51%), just Cd (46%) and Ni (63%) were largely mobilizable in acetate buffer pH 5. Among exchangeables, Cd was largest at 14%. Major amounts of Hg were released by oxidation as the organic-sulfidic fraction [20].

In **Havana** (Cuba), the pseudo-total contents and Mehlich3 extracts were significantly correlated, except for Fe, As, and Hg. Mehlich 3 extracted 30% of Pb and Cd, 22% of Mn, and 18% of Zn and Cu, but negligible amounts of As, Cr, Fe, Ti, and Hg. Acid oxalate extracted more than 10% of Hg, Co, Mn and Zn, with respect to reverse aqua regia [51].

## 4. Platinum Metals

### 4.1. General

Pt occurs associated with Ir, Os, Pd, Ru, and Rh, usually below 1 µg/kg, and 0.4 µg/kg for both platinum and palladium are widely regarded as the geological background. Though they are recovered as a byproduct from gold refining, local enrichment of gold in urban dust samples from England 1993/94 was not specifically attributable to roads or comparable with the Pt distribution, but it might be related to urban infrastructure development [107,108]. Similarly, nor in street dusts of Seoul nor of Zadar, correlations between Au and Pt were found [73,94].

Total dissolution of the sample results in the increase of elements interfering with the ICP-MS final determination of Pt and Pd, e.g., Zr, Hf, Y and Cu, which yield additional signals at the Pd resp. Pt mass. Because Pt and Pd are emitted from catalytic converters mainly in the elementary form, total digestion of road dusts and topsoils by hydrofluoric acid is not necessary, but rather sufficient oxidation. Aqua regia or reverse aqua regia digests leave substantial amounts of Zr, Hf and Y in the solid. Anion exchange separation via chloro complexes is interfered by fluoride complexes of Zr, Hf and Y, but works well after aqua regia digestion [58,109].

Molecular interferences from Cu-Ar, Y-O, Sr-O and Zr-O yielding positive signals upon the mass of Pd in the ICP-MS can also be overcome by reductive co-precipitation with Hg, and use of He as a collision gas [110].

Whereas the enhancement of $HfO^+$ upon the Pt-signal could be mathematically corrected, respective interferences of $^{87}Sr^{16}O^+$, $^{87}Rb^{16}O^+$, $^{89}YO^+$ and $^{68}Zn^{35}Cl^+$ on the mass signals of $^{103}Rh$ and $^{105}Pd$ were overcome by reductive co-precipitation with Hg. The influence of $^{65}Cu^{40}Ar^+$ upon $^{105}Pd$ was small. The Pd-data could be confirmed by total-reflection x-ray fluorescence [111].

Digestion by reflux in glass with reverse aqua regia and 20 µL of elemental bromine was successful to dissolve also the platinum metals present in the metallic state. Whereas ICP-OES was sufficiently sensitive to determine a lot of contaminant metals, ICP-MS was needed to get Pd, Pt, Ir, Au, Hg, Bi, and Tl. Positive interference of Y-O upon Pd-masses necessitated a respective separation, which was achieved by co-precipitation with dithizon [58]. Platinum metals remained stable in solution in glass flasks in aqua regia in presence of some elemental bromine, but interactions with plastics should be checked.

Pt in road dust occurs as oxide, chloride or metallic Pt, its solubility in rainwater is within 0.010–0.025%; sulfur in soil or water increases its solubility. Pt(IV) with its high redox potential can oxidize sulfur in side chains of proteins and denature them, causing inhibition of all functions and destructions of all membranes. Steadily increasing intensity of traffic leads to increase of Pt in environmental matrices, like roadside soil and grass samples [112].

### 4.2. Use of Platinum Group Elements

For platinum group metals, catalyst materials used in cars are currently regarded as the main emission source. However, palladium emissions from industry cover a larger part than platinum, compared with emissions from traffic [113]. Platinum metals may also be emitted from the jewelry production (e.g., like around Pforzheim city, Germany). Gold-palladium and palladium-based

alloys have been used in dental labs and surgeries, which enter the urban dust as abrasion from pedestrians [114,115].

Iridium and ruthenium are not constituents of the 3-way catalyst, they originate from noble metal processing industries [113]. For municipal sludge in England, it is unlikely that Ir, Ru and Os originated from automobile catalysts [115].

### 4.3. The Catalyst Technology

Pt- containing catalysts were introduced to reduce air pollution [37]. The catalysts remove about 90% of CO, unburned hydrocarbons, and nitrogen oxides from vehicle exhausts [116]. Pt and Pd support the oxidation of organics to $CO_2$ and $H_2O$, whereas Rh catalyzes the reduction of $NO_x$ to yield $N_2$. Alternatively, Pd-Rh as the active layer of the monolithic catalyst body can be also used [114,117]. Modern 3-way catalysts contain 2–3 g of platinum group elements where the ratio Pt/Rh or Pd/Rh is usually 5:1. The ratios of Rh, Pd and Pt in catalytic converters are constantly changed to optimize performance [118].

The metal catalyst, containing 1–3 g of platinum group metal, is supported on a ceramic honeycomb monolith housed in a stainlesssteel box, and is designed to function for a minimum of 80,000 km [107]. Zr-oxide in the autocatalyst acts as a stabilizer, and Ce-oxide as an oxygen reservoir, they are present as well in Pt-based as well as in Pd-based catalysts [116,119].

Platinum–group elements are contained in the catalytic layer at a percentage of 0.10–0.15% (weight). They are emitted primarily as particulate matter in consequence of the thermal and mechanical losses of the catalytic material when the engine is working. Engine test-bench experiments showed platinum- emissions to be higher at higher driving speed and engine temperature. Moreover, age of the converters and malfunctioning of the engines affected the platinum-group-emission rates. In particular, large temperature gradients result in the release of Pt from the monolith in form of small particles [108,120].

Other elements like Ce, La, Nd, Zr etc. are identified as emitted together with platinum group elements from catalysts [121], but currently, data about rare earths present in urban soils, are scarce.

In Japan, automobile catalysts based on palladium as the active layer have been introduced in the early 1970's, and in Germany, autocatalyst technology was introduced in 1984 [114]. Catalytic converters became mandatory in all new cars sold in Australia from 1986, and in Austria since 1987 [116,122]. Since 1993, catalytic converters for automotive traction are mandatory on all new gasoline-driven cars in the EU [120].

In Germany 1990–1994, 35% of the admitted cars were equipped with catalysts [123], in Austria in 2000 about 52% [122], and in Hong Kong 2003, 97% of petrol and 5% of diesel vehicles [119]. In addition to Pt-Pd-Rh, the use of Ir as a novel active metal in catalytic converters ought to be reflected in roadside soils [122].

Palladium has increasingly substituted platinum in automotive catalysts, and iridium has recently been used in Japan in so-called De-NOx catalysts which have been developed to reduce the emission of nitrogen oxides in the exhaust of lean-burn engines [121].

Automobile catalytic converters are the important source of Pt in roadside environment, and there is a tendency to increase Pt levels in road dusts along with traffic volume.

Merget and Rosner [124] estimated the emission of platinum from cars equipped with catalysts as 60 ng/km for the city, 20 ng/km for a street tunnel at constant speed at 80 km/h, and as 140 ng/km for a highway at constant speed at 130 km/h. Platinum is emitted from automotive catalytic converters in particulate form, mainly as elemental platinum. The nanocrystalline particles are attached to μm-sized aluminum oxide particles.

Pt and Pd levels depend on the time and frequency of the use of catalytic converters for automobile exhausts, and still start to accumulate. This is documented by data from Zadar and Novy Sad, which were obtained before cars were equipped with catalysts, from Budapest with a few of them, and Seoul, where catalysts were already quite common (see below).

Higher relative solubility of Pd compared with Pt and Rh means probably higher uptake by flora and fauna. The majority of emitted platinum group metals are in metallic form.

### 4.4. Pt-Metals in Urban Soils

In **Brno**, a town of about 400 thousand inhabitants, roadside soil and grass samples as well as road dust and tunnel dust from about 1 m height above the road were sampled and digested with aqua regia, and preconcentrated as chloro-complex on a C-18 column in presence of a cationic detergent. In the vicinity of roads with high traffic, Pt in soils ranged within 10.5–15.7 µg/kg and in respective grass within 10.0–11.6 µg/kg. Similarly, in regions of medium traffic, soils contained 6.02–8.44 µg/kg in soil and 5.61–7.85 µg/kg in the grass, and finally in city outskirts within 4.00–6.48 µg/kg in the soils and 3.99–5.02 µg/kg in the grass. Pt levels in parks showed Pt below the quantification limits. Equal levels of Pt in soil and respective grass implies easy transfer to the biomass (or maybe an inefficient plant-washing procedure) [112].

In the city of **Zadar** (Croatia), topsoil samples 0–10 cm were sampled parallel to the coastline near the shore of the city, sieved <2 mm, and digested with reverse aqua regia in presence of bromine, in order to dissolve Pt-metals also. Because samples were taken in 2003–2004, before the onset of the use of Pt catalysts in that region, Pt-contents was just about 1 µg/kg, and Ir was still less. Pt, Ir, Au, Tl and Bi in the topsoils were found at ambient levels and showed symmetrical frequency distributions [73].

In **Salerno** (Italy), platinum-group element strongly correlated with typical traffic emitter indicators Cr, Cu, Sb, and also Sn, whereas areas with low population density and low traffic intensity had concentrations close to background values < 1 µg/kg for Pt and Pd, and <0.4 µg/kg for Rh, resp. [75].

In **Rome**, topsoils contained some Pt (3–15 µg/kg) and Pd (4–16 µg/kg) due to the introduction of catalytic converters some years before sampling, whereas this was negligible in **Novi Sad** (<1 µg/kg) [74].

In the ash stored in the landfill site at **Sheffield**, platinum and rhodium as well as the Pt/Pd ratio were highest in 1998 and decreased since then. Ru, Ir and Os varied very little over a 6 years period. Normalized data show that the influence of point sources is negligible, and palladium is more mobile than platinum [115].

In roadside dust and roadside topsoil samples from **Ioannina** (Greece), the proportions Pt/Rh, Pt/Pd and Pd/Rh were not in agreement with the potential catalytic converter manufacturing ratios. The platinum group elements over Pb ratio showed a similar pattern at all sites, because a part of the vehicle fleet still used leaded gasoline. Tsogas et al. normalized their data to local background levels, which was obtained from a presumably non-contaminated sample of the same soil type. For Ioannina in Northwest Greece, they took 6.7 µg/kg Pt, 1.28 µg/kg Pd, 1.21 µg/kg Rh, and 12.3 mg/kg Pb as the background for the soil, and 2.1 µg/kg Pt, 0.98 µg/kg Pd, 0.27 µg/kg Rh, and 11.8 mg/kg Pb as the background for the upper crust. In roadside topsoils from Ioannina (Greece), the palladium distribution was affected by pH, organic matter and water, which suggests increased mobility of Pd in comparison to other PGEs. Maximum concentrations were reached during summer and spring, and minimum during winter and autumn months. Not just the traffic load, but driving stile, matrix compositions and diurnal variation of traffic volume as a function of climatic conditions affect the platinum group element levels [125].

In **Brazil**, catalytic converters have been in use since 1996, but not for buses and trucks. Contrary to cars in Europe, Asia or North America, Brazilian vehicles use an admixture with 20% ethanol, and the catalytic converters contain mainly Pd and Rh, instead of Pt. Along seven main avenues of high density traffic in Sao Paulo (Brazil), soil of 0–5 cm were sampled in grass strips at 15 and 115 cm distance from the asphalt, sieved <2 mm, digested with aqua regia, and measured by high resolution ICP-MS. Soil samples were characterized by pH 6–7, 4–7% organic matter, and a high amount of clay-size particles of 40–50%. Higher organic matter tended towards higher platinum group element levels, but there was no characterization with granulometric soil characteristics, Sampling occurred about 15 years after introduction of catalytic converters, and lead to 3–378 µg/kg Pd, 1–208 µg/kg Pt, and 0.2–45 µg/kg Rh. No differences between high speed and stop and go avenues were observed.

The Pt/Pd ratios ranged within 0.2–0.4, and were lower than from many comparable studies from other regions of the globe [38].

Tunnels represent specific urban sites, where dust containing Pt accumulates. At tunnel wall about 1 m above the road, 110 µg/kg of Pt ware found before cleaning, and 35 µg/kg after one week after cleaning [112].

*4.5. Road Dusts*

Surface dust samples around **London** were collected with a plastic scoop, oven dried and sieved <75 µm, 75–125 µm and 125–250 µm. At most sites, maximum Pt was found in the finest fraction, and for the rest in the medium fraction [108]. Street dust samples from England, at the start of the mandatory use of car catalysts, contained 0.42–29.8 µg/kg Pt. The lowest values were found from dusts from side streets with lower traffic density and were generally comparable to local soils. The highest Pt concentrations occurred at major road intersections. There were no significant relationships between Pt, Pd and Au [107].

**Białystok**, a town of about 300,000 inhabitants in northeastern Poland, sampling of road dust occurred 3–4 years afterwards, but already 34–111 µg/kg Pt were found at sites of high traffic, sampled after a dry period of 3 weeks [111]. The road dust samples were sieved <0.3 mm, <0.15 mm, and <0.075 mm, and digested with HCl/HNO$_3$/HF, and determined by high-resolution ICP-MS, which improved the detection limit significantly, with respect to a quadrupole tool [108].

Tunnels represent specific urban sites, where dust containing Pt accumulates. At tunnel walls of a city highway in **Brno**, about 1 m above the road, 110 µg/kg of Pt were found before cleaning, and 35 µg/kg after one week after cleaning [112].

In dust samples from **Germany**, platinum up to 308 µg/kg close to motorways, 257 µg/kg at city sites, 189 µg/kg in parking garages, and 141 µg/kg in roadway tunnels were found, whereas platinum-rhodium levels in urban dust from living quarters were quite low [123]. In urban dust from Germany, ample proportions of Pt/Pd = 0.04–31 were found [114]. Along German highways, the peak concentration of platinum group metals in dust collected by Bergerhoff vessels, at 2 m was attached on a coarse grain size fraction, and at 15 m on a fine fraction. 65 % of the traffic platinum metal emissions were bound to particles >10 µm, and most of the platinum group metals at <25 µm. After deposition, no significant speciation changes took place anymore [113,123]. In effluents from various roads (pH 5.8–6.7), platinum concentrations were marginal (0.010–0.078 µg/L) [123].

At various sites in **Budapest** (Hungary), road dust samples were sampled by simple brushing, two times from traffic focal points in Budapest, from the large bridges across the River Danube, from Margitsziget (an island in the Danube in the Northern part of Budapest, used for recreation) as well as from main roads (no highways) outside Budapest. After sieving minor 0.075 mm and digestion by reflux with reverse aqua regia plus elemental bromine, they were analyzed by ICP-OES and ICP-MS. In Budapest, maximum Pt levels were found in dust collected at the central bridges across the River Danube, where urban traffic jams are focused, both more wind can be expected also. Note the low level at the most northern bridge Margit Hid, which is close to a recreational area at Margit Sziget (Margit island). Pd maxima were rather met at the big squares in the East of Budapest. In Budapest. Pt and Pd concentrations hardly correlated with other contaminant metals, except at the Danube-bridges and the center. Hotspots of Pt and Pd occurrence were found at sites of intense traffic, particularly at stop-and-go sites, which seem to be the bridges across the Danube [59,94].

Topsoils (0–10 cm) in the **Emilia** (Northern Italy) contained 2.5–2.8 µg/kg Pt and 4.0–4.2 µg/kg Pd at remote areas, but 63–73 µg/kg Pt and 83–112 µg/kg Pd at urban or high traffic sites [126], beneath accumulations of Cd, Ni, Pb, and Zn. Like platinum, there were also maxima of zinc, lead and copper in soils 0.1 m and 0.5 m aside from roads, but in these cases also migration to deeper layers was detected. Migration of platinum group metals is strongly retarded at the pH 7–8 of the street dust [117].

Significant platinum- and rhodium concentrations in urban dust collected in the city of **Rome**, were found in the vicinity of traffic signals, indicating that the "stop and go" conditions might affect their release. Neither Pb not Cu were strongly correlated with any of the platinum group metals [120].

Road dust samples from various sites in **Seoul** (Korea) were collected from 1 $m^2$ directly from the road surface by using a plastic hand brush and a dustpan. The samples were digested by reflux in glass like sewage sludge resp. soils with reverse aqua regia plus elemental bromine. The study also suggested that not only traffic volume but also driving style have a great influence on Pt levels in road dusts. The concentration levels of Pt and Pd were in the range of 2.3–444 (median 76) and 172–1215 (median 609) ng/g, respectively. Palladium also showed similar distribution trend with Pt, and remarkably high concentration of Pd and Pt in dust was found in the heavy traffic areas. The closely correlated elements with Pt and Pd in dust were Be, Cr, Cu, Fe, Mo, Ni, Au, Hg, and Bi (higher than r = 0.50). Road dusts with high Pt and Pd levels were enriched in traffic-related elements compared with those from control suburb areas. The top levels could be assigned to erratic stop-start driving conditions. Contrary to heavy contaminations with Zn and Cu, Pb was comparatively lower, because unleaded gasoline had been in use during 17 years before actual sampling [59,94].

Catalytic converters became mandatory on all new cars sold in Australia from 1986. About 15 years afterwards, in **Perth** (Western Australia) samples of road dust and roadside soils 0–1 cm as well as a reference soil were taken, sieved <0.063 mm, and digested with aqua regia. The platinum group metals were separated from the matrix by sorption of interferents upon a cation-exchanger from 0.6 M HCl, but Hf-fluoride eluted together with Pt, and Pt-mass had to be corrected from $HfO^+$. Shifts of the Pt/Pd proportion towards higher Pd levels compared with data found in European cities might be due to lower washout in dry Western Australia. Pt in road dust and roadside soils ranged 30–420 µg/kg, and Pd 13–440 µg/kg. No straightforward correlations of platinum group element levels with traffic volume, but with stop and go frequencies, were found. The proportions among the platinum group metals were narrow, Pt/Rh = 5.1 (range 4.6–6.3), Pt/Pd = 1.04 (range 0.7–1.5), and Pd/Rh = 5.17 (range 3.3–6.9). In the background soil from the remnant bushland above the city, aqua regia and $HF/HNO_3$ digests yielded the same results for Pd and Rh, but Pt in aqua regia was slightly less [116]. Systematic seasonal variations in platinum group element levels were observed both in road dust as well as in roadside soil samples. The levels were also affected by surface morphology and rainfall [116].

*4.6. Solubilities and Mobilities*

Emitted nanocrystalline particles from automotive catalysts contain mainly metallic platinum, the oxidic fraction is usually below 5% [124]. In the automotive test stand, liberation of platinum, both as a metal and in quadrivalent form, was detected caused by temperature increase and vibrations. New cars emit approximately 43–133 ng/km, and used cars 6–162 ng/km of platinum, of which less than 10% are water soluble [123].

In sewage sludges sampled all over England, platinum group element concentrations were found 3–7 times lower than in corresponding incinerator ashes, which means, they are not volatile during incineration. The concentrations of rhodium, ruthenium, iridium and osmium in sewage sludge were much lower than platinum and palladium, with maxima < 10 µg/kg for Sheffield and <20 µg/kg for Birmingham. The concentrations found in the stored ashes were highly variable in platinum and palladium. The proportion Pt/Pd ranged between 0.1 and 2.2 in incinerator ash and sewage sludge. For many sampling sites, many significant correlations between platinum group element concentrations could be established. Pt and Pd concentrations were about 1/10, whereas Ru and Ir were at the same level, and increased in the incinerator ashes 2–10 fold [115].

In order to investigate speciation transformation of various emitted platinum species, 25 g homogenized soil were mixed at room temperature with either Pt-black, or $K_2PtCl_4$, or $Na_2PtCl_6$, or $PtO_2$, or tunnel dust. The mixtures were shaken with 100 mL $H_2O$ up to 60 days, and the supernatants centrifuged, filtered, digested, and measured by ICP-MS. The 30-days residues were also further extracted with organic solvents resp. complexing agents. As a result, the solubility of $PtO_2$ was

very low at 0.01%. $Na_2PtCl_6$ and $K_2PtCl_4$ solubilities rapidly decreased within 1 day below 1%. The solubility of Pt-black was lower than 0.1%, but had a maximum of 0.3% after 7 days. The solubility of Pt released from tunnel dust varied between 0.5–4%. Organic solvents did not release any Pt from the soil mixtures, except 3.1% from the Pt-tunnel dust in methanol. EDTA released less than 5%, but about half from the soil- tunnel dust mixture, however just 10% from tunnel dust alone [127].

In physiological saline solution, however, tetra- and hexachloroplatinate can be formed [124]. In simulated human stomach and intestine conditions, simulated bioaccessibility increased from the acidic stomach to the neutral, carbonate-rich intestine. It was controlled by dissolution rates of metallic nanoparticles in the stomach, and solubility as well as undefined organic complexes in the intestine. After 1 h, the model solution pepsin-citrate-malate-lactate-acetate-hydrochloric acid pH 2.5 predicted a solubility of 40% Rh, 40% Pt, and 60% Pd. At pH = 1, Pt dissolution is favored by the formation of simple chlorides. Sparingly $Pd(OH)_2$ becomes increasingly important at neutral conditions, as well as mixed chloro-carbonato-complexes $PdCl_3CO_3{}^{3-}$. In case of Pt and Pd, organic complexation is kinetically hindered [128].

## 5. Roadside Flora

Roadside plants have not been investigated very often, and this chapter should encourage readers to do further respective investigations. Roadside trees and herbs are usually cared for as a means of street maintenance, and leaves and cuttings are put to waste or compost. In poorer countries, however, this biomass may be used as feedstuff as well.

### 5.1. Roadside Trees

In the cities, tree root space is limited by pavements, and polluted by salt application during wintertime. In order to optimize an urban tree soil substrate to set young trees in prepared ditches, requirements to be achieved are 83–87% degree of density, $>5.10^{-6}$ m/sec water permeability, >25% water capacity, >15% air capacity, pH within 5.0–8.5, and salt content < 1.5 g/kg. Planting of urban trees (*Celtis australis*) in Vienna was successfully done in a pre-mixed substrate from fluviatile sediment, compost, dolomite chippings, and sand [129].

At non-disturbed sites, the root space exceeds the volume of the aerial part of the tree and increases with increasing height and age of the tree down to several meters. Setting of trees necessitates digging respective big ditches and filling with an optimized substrate. A 25 m high tree can reach e.g., down to 25 m, and an 18 mg height tree down to 15 m. The mixed substrate should be stored for some time before setting of the trees. Big trees filter the air and cool down in summer because of transpiration and shadowing, but need sufficient root space, water and oxygen supply [129].

In urban environment, root growth of trees is limited by overlying pavement as well as compaction of deeper layers. An asymptotic non-linear model was used to describe the vertical distribution of root mass of *platanus orientalis* trees on the outskirts of Christchurch, New Zealand. In the uppermost 20 cm of soil, all pavement treatments altered root abundance. Intensified shallow root growth beneath pavement cracks is traceable to high water infiltration rate, as well as the barrier effect of the pavement to $O_2$ diffusion. Most of the root biomass was found within 0–30 cm, and fine roots go deeper, but age and height of the trees was not recorded [130].

Along urban roads, big roadside trees can be used to indicate cumulative contaminations from traffic emissions, but this is site- and species dependent. In the city of Novi Sad (Serbia), *Platanus acerifolia*, *Celtis occidentalis*, *Tilia argentea* and *Quercus robur* were sampled for leaves at 2 m height as well as adjacent soils within 30 cm distance, at 0–15 cm depth. Sampling only 0–15 cm assumed that the roots of urban trees do not penetrate more than 0.9 m because of urban soil compaction. In the soils, positive correlations between Cu, Zn and Pb were found, but not necessarily in the corresponding leaves analyzed. The levels encountered in the leaves indicated enrichment of Pb, but not of Cu and Zn, and species-specific differences emerged. *Tilia argentea* had significantly less Cu than *Quercus robur* and *Celtis occidentalis*, *Celtis occidentalis* and *Platanus acerifolia* had less Zn than the others, *Celtis* had highest

Pb, and *Tilia* had the lowest Pb. Apart from slightly different soil levels, root uptake, translocation and leaf surface adsorption are responsible for different metal levels in the leaves. Leaves with good bioaccumulation abilities can be used for phytoremedation of urban soils, if they are removed in autumn and then safely deposited [101].

Leaves of sour cherry (*prunus laurus cerasus*) have been found as effective collectors of platinum group element emission. After washings with water and 2-propanol, the dust could be separated from the leaves by centrifugation [131].

Tree bark has been shown to be an effective substrate for collection of airborne contaminants. 57 bark samples were taken by scalpel at about 1 m from the ground. Uncontaminated tree barks contained <0.1 µg/kg, and increased to several µg/kg at contaminated sites [132]. Tree barks sampled in Seoul urban area ranged between 0.9–4.5 µg/kg, indicating the deposition of Pt-containing particles [94]. Similarly, grass samples from rural areas ranged between 0.1 and 0.3 µg/kg, but rose to 800–3000 µg/kg in areas exposed to traffic. The rare earth concentrations were highest for Ce (13 mg/kg), Nd (7 mg/kg) and La (6.5 mg/kg) in the case of a bark collected near a coal fired power station, where Pb was 280 mg/kg [132].

## 5.2. Roadside Herbs

Roadside flora as well as soil and air in the vicinity of roads undergo contamination by oil derivatives, exhaust gases like $CO$, $CO_2$, $NO_x$ and $SO_2$, as well as metal abrasions from vehicles (Zn, Cu, Pb, Cd, Mn), till a distance of 50–100 m distance from the road. This leads to a metallophyte roadside flora. In Szczecin Lowland in 0–0.3 m distance from the road, Poaceae (*Arrhenaterum elatius, Calamagrostis epigejos, festuca pratensis*), Asteraceae (*Bellis perennis, Cichorium intybus, Leontodon autumnalis*) as well as *Equisetum arvense, Cerastium semidecandrum, Convolvulus arvensis*, and *Carex hirta* were found. *Leontodon autumnalis* obviously requires the increased content of heavy metals in soils and grew up in masses in the proper roadside zone. Both at field and forest roadsides, pH in 0–10 cm soil depth decreased with distance to the road at 0.5 to 2.1 pH units from neutral to acid. Alkalinity close to roadside may result from de-icing agents in winter. Mn, Pb, and Cu strongly declined from zone A (0–0.3 m) to zone D (5–8 m). The accumulation depended on the deposition of traffic pollutants on the bare soil along field roads or at plant surfaces. Participation and cover of metallophyte species was larger in all zones along field roads compared to forest roads. The largest number of metallophyte species were found in 0.3–2 m distance from the road [133].

Roadside soils can reach potentially toxic concentrations to harm roadside flora and earthworms. Mn, Sb and Pt-group metals show increasing trends in urban roadside soils [37]. Within 15 years of catalyst use, the concentrations of platinum and palladium in roadside dandelions and plantains in Germany increased 100–200 fold [121].

Along some highways in Germany, less than 1 m from the road edge, 5 kinds of plant samples were collected in September 2009 after a 2 weeks dry period: dandelion (*taraxacum officinale*), plantain (*plantago lanceolata*), moss (*rhytidiadelphus squarrosus*), mushrooms (*vascellum pratense*), and annual ryegrass (*lolium multiflorum*). Pt, Pd, Rh, Ru, Ir, as well as Ce, La, Nd, Pb, and Zr were determined. Dandelion had the highest concentrations for all sampling sites in comparison to the other plants from the same site, and annual ryegrass had the lowest. Pb, La, Ce, Nd, which are also released by cars showed the same behavior at the plants than the platinum group metals. After careful washing with tap and bi-distilled water, a decrease in the concentrations of all elements was established, both for dandelion and plantain, between 2-fold (Ru, Rh), and 3.3 (Zr) [121].

Plants are able to assimilate Pt, Pd and Rh. Ryegrass (Lolium perenne) pots were exposed along a highway with a daily average traffic of 65,000 vehicles to determine the effects of the exposure duration and the evolution of the pollutant concentration in the plants. From day 0–8, Pd and Pt increased rapidly by a factor of 3 [134].

Pt gets enriched in the topsoil layers and the small grain size fraction, in this work at <0.075 mm. In hydroponic cultures, Pt-group compounds are readily taken by green plants, with highest concentrations in the roots, but the availability of emissions from converters is only about 1%. [113].

In order to test the soil to plant transfer of platinum group metals, spinach, cress, phacelia and stinging nettle were cultivated in pot experiments in a greenhouse on different soils collected from areas adjacent to a German highway, as well as on uncontaminated sandy and clayey soils. Sampling of plant directly from the road would record the deposition on the surface rather than the plant uptake. Just deionized water was added, and the plants harvested after 6 weeks about 1–2 cm above the ground. For all test cultures, the transfer for Pd was larger than for platinum, rhodium and copper. The transfer for Pt was still one order of magnitude higher than for Pb [135].

The soil plant transfer of $PtCl_6^{2-}$ was tested in pot experiments with cabbage, utilizing 50–60 kg of eutric fluvisol and chromic luvisol. The transfer factor for Pt to the cabbage was 0.0081 for the eutric fluvisol and 0.0041 for the chromic luvisol [121].

## 6. Conclusions

Contrary to soil inventory studies for agricultural purposes, which follows standardized procedures, sampling and analysis of urban soils has been done highly variable. To find some conclusions, the individual datasets given in the tables of the Appendix A are evaluated with respect to sampling depth, sieved grain size, digestion resp. analytical method, annual precipitation and year of sampling. In order to exclude bias from the outliers, median values (see Appendix A) were compared wherever possible, and top values kept in mind.

Whereas the road dust data (see Appendix B) refer to samples brushed from the surface, urban soils have been cored 0–2 cm, 0–5 cm, 0–10 cm, 0–20 cm and 0–30 cm, which leads to different proportions of dilution of the road dust and possible pollutants with original soil. From the current data, no effect of the median concentration found for different **sampling depths** were found for As, Cu, and Zn, because pollution dominates, for Ba, because digestion method is dominant, and for Co, Cr and Mn for mixed reasons. In case of Cu and Hg, some top values occurred at 0–5 cm and 0–10 cm, but no general trend could be noticed. Ni tends to decrease with increasing sampling depth, apart from two high outliers. Maximum Pb levels were found in cores 0–10 cm, indicating a slight downward trend of the pollution peak in the past. With respect to Ag, Mo, Sb, Sn, P, and V, not enough data were available from different core depths to come to conclusions. Just for Uppsala [8], soil profile studies have been found, sampled 0–5 cm, 5–10 cm and 10–20 cm at the same spot. In Uppsala, median concentrations of total Al, Fe, Mn, Ni and Zn rise with increasing depth, possibly because of geogenic reasons, whereas As, Cd, Cu and Pb remain constant, apart from some top Cu-values in 5–10 cm.

XRF (X-ray fluorescence) and **digestions** using acid mixtures with HF (hydrofluoric acid) yield total concentrations, whereas mixtures without HF yield quasi-total concentrations. Expectable differences are seen for Al, Fe, Ba, and Cr up to twofold, and respective datasets should not be mixed into one table.

Most urban soil samples were sieved to **grain size** < 2 mm. No significant differences were found for samples < 1 mm, <0.425 mm, <0.18 mm and <0.075 mm, because not many data were available from this classes, and larger effects might dominate. For Ba, all finer fractions are totals. V might be enriched in finer fractions. Just for Lithgow in Australia [49], data from different grain size fractions have been reported for same spots and time. Median Pb and Cu concentrations were found enriched in the fraction < 0.18 mm versus <2 mm, but top values occurred in both.

Contents met in urban soil equals geogenic contents plus emission minus plant transfer minus washout. Therefore, all median data were correlated with mean **annual precipitation** data, obtained from the papers itself or from Wikipedia by internet. It makes sense to treat "urban soils" (the medians obtained from regular sampling grids), park soils and roadside soils separately. Some negative correlations were seen between mean annual precipitation and As (r = −0.510) and Zn (r = −0.255) in urban soils, Cu (r = −0.374), Pb (r = −0.212) and Zn (r = −0.515) in park soils, which indicates more washout, whereas Cu in roadside soils increased with annual precipitation

(r = +0.596). If specified, park soils were not in all cases lower in heavy metals than roadside soils, as expected.

Levels of **park soils** were lower than **roadside** soils for Cd, Ni, Zn (except Vienna [53]) and Pb (except Kielce [69]), for others, no differences were seen. The two papers from Beijing [78,79] used too different sampling methods to permit comparison (roadside 0–20 cm/HF-digestion/unsieved versus park soils 0–5 cm/aqua regia/<0.125 mm).

Some soils from urban regions have been sampled and analyzed by approximately the same method in different years, thus some **time trends** can be estimated. Levels of As, Cd, and Ni remained about constant in Vienna between 1997 [70] and 2003 [16], as well as in Trondheim between 1994 and 2004 [66], but maxima in Trondheim were greatly reduced. Cd, however, was reduced about 10-fold in Hangzhou from 2009 (0–10 cm; [23]) to 2019 (0–30 cm; [63]) and in Hong Kong from 2000 [82] to 2004 [81]. Co and Cr remained at the same level in Vienna between 1997 [70] and 2003 [16]. Cu levels decreased only slightly in Trondheim [66] and Vienna [16,79], but extreme top concentrations disappeared. In Hangzhou, Cu levels in industrial, roadside and residential urban soils decreased to ambient levels [23,63]. Pb levels decrease anywhere, particularly the maxima. In Hong Kong and Hangzhou, Pb decreased to 1/3 within 4 (2000 to 2004) resp. 10 (2009–2019) years. In Trondheim and Vienna, Zn median concentrations decreased slightly, but maxima considerably. In Hangzhou and Hong Kong urban soils, Zn got lowered to ambient levels within 2009 to 2019, or 2000 to 2004 resp.

Within the cited texts, also indications about pollution changes have been found. New York City had passed a maximum emission of Hg, Pb, Cr and Ni [20] some decades before. In Adelaide garden soils, maximum Pb was found at 10–30 cm depth [14]. In Kielce, a 3-fold decrease of Pb and a 2-fold decrease of Zn was noted along transportation routes compared with data obtained 16 years beforehand (69). In Tyumen, highest Co, Ni, Cr, Cu, and Pb were typical for areas formed in about 1940–1960, and Zn and As maxima for the historical center [64]. In Havana, top Hg was found in Habana Vieja [51], the historic center.

For risk assessment reasons, grain size and solubilities are important, and it remains unclear to the author, based upon which sampling parameters respective risk factors had been derived. The estimation that about 12% of the fraction < 2 mm is inhalable, seems quite rough when globally assumed, particularly if 0–30 cm had been sampled.

**Funding:** This research received no external funding.

**Conflicts of Interest:** The author declares no conflict of interest.

## Appendix A. Urban Soil Data

Within the subsequent tables in Appendix A, results of urban soils are given in mg/kg. In case the sampling year is not given in the text, it is assumed to be 2 years before publication. Mean crust values are taken from Wedepohl and Rudnick/Gao [40,136].

Determination methods are ICP-OES, ICP-MS after digestion, resp. XRF = X-ray fluorescence, or DC-arc = direct current arc emission spectrography.

Urban soil: sampling along a sampling grid within the entire area.

Roadside soil: sampling close to roads.

**Table A1.** Aluminum in urnan soils.

| | sampling year | reference | Sampling depth cm | grain size mm | Digestion | % Al |
|---|---|---|---|---|---|---|
| Upper crust | | [136] | | | | 8.10 |
| Continental crust | | [40] | | | | 7.96 |
| Aberdeen parkland | 1994 | [65] | 0–10 | <2 | HNO$_3$/HCl | 1.39 ± 0.11 |
| Aberdeen roadside | 1994 | [65] | 0–10 | <2 | HNO$_3$/HCl | 1.28 ± 0.08 |
| Uppsala | 2003 | [8] | 0–5 | <2 | HNO$_3$/HCl | 1.84/1.02–3.26 |
| Uppsala | 2003 | [8] | 5–10 | <2 | HNO$_3$/HCl | 2.25/0.96–3.53 |
| Uppsala | 2003 | [8] | 10–20 | <2 | HNO$_3$/HCl | 2.47/1.03–3.45 |
| Zadar urban soils | 2003/2004 | [73] | 0–10 | <2 | HNO$_3$/HCl | 2.77/0.65–4.23 |
| Salerno urban soils | 2018 | [75] | 0–20 | <2 | HNO$_3$/HCl | 3.00/1.00–6.30 |
| Tyumen urban soils | 2016 | [64] | 0–10 | <1 | XRF | 4.56/0.79–6.88 |
| Xuzhou urban soils | 2004 | [80] | 0–10 | <2 | HF/HNO$_3$/HCl-XRF | 5.87/5.13–8.04 |
| Danang urban soils | 1995 | [83] | 0–20 | <0.063 | HF/HNO$_3$/HCl | 6.94/3.07–9.37 |
| Bangkok urban soils | 1996 | [84] | 0–5 | <2 | HNO$_3$/HClO$_4$ | 1.25/0.14–4.34 |
| Ottawa garden soils | 1993 | [86] | 0–5 | 0.1–0.25 | HNO$_3$/HF/HClO$_4$ | 5.58/4.54–6.22 |

**Table A2.** Iron in urban soils.

| | sampling year | reference | Sampling depth cm | grain size mm | Digestion | % Fe |
|---|---|---|---|---|---|---|
| Upper crust | | [136] | | | | 5.20 |
| Continental crust | | [40] | | | | 4.32 |
| Aberdeen parkland | 1994 | [65] | 0–10 | <2 | HNO$_3$/HCl | 1.85 ± 0.011 |
| Aberdeen roadside | 1994 | [65] | 0–10 | <2 | HNO$_3$/HCl | 1.81 ± 0.08 |
| Uppsala | 2003 | [8] | 0–5 | <2 | HNO$_3$/HCl | 2.49/1.56–3.79 |
| Uppsala | 2003 | [8] | 5–10 | <2 | HNO$_3$/HCl | 2.76/1.33–4.07 |
| Uppsala | 2003 | [8] | 10–20 | <2 | HNO$_3$/HCl | 3.03/1.36–4.11 |
| Warsaw urban soils | 1996 | [7] | 0–20 | <1 | 450°/HCl | 0.90/0.35–3.10 |
| Zadar urban soils | 2003/2004 | [73] | 0–10 | <2 | HNO$_3$/HCl | 3.33/0.87–4.11 |
| Sevilla parks-gardens | 2000 | [9] | 0–20 | <2 | HNO$_3$/HCl | 2.09/1.45–2.71 |
| Vigo parks-gardens | 2013 | [12] | 0–20 | <2 | XRF | 2.58/1.17–5.20 |
| Salerno urban soils | 2018 | [75] | 0–20 | <2 | HNO$_3$/HCl | 2.32/1.04–3.23 |
| Tyumen urban soils | 2016 | [64] | 0–10 | <1 | XRF | 1.53/0.31–3.36 |
| Xuzhou urban soils | 2004 | [80] | 0–10 | <2 | HF/HNO$_3$/HCl-XRF | 3.29/2.66–4.15 |
| Hangzhou industrial | 2009 | [23] | 0–10 | <2 | HNO$_3$/HClO$_4$/HF | 3.91 ± 1.43 |
| Hangzhou roadside | 2009 | [23] | 0–10 | <2 | HNO$_3$/HClO$_4$/HF | 3.05 ± 0.69 |
| Hangzhou residential | 2009 | [23] | 0–10 | <2 | HNO$_3$/HClO$_4$/HF | 2.89 ± 0.63 |
| Hangzhou parks | 2009 | [23] | 0–10 | <2 | HNO$_3$/HClO$_4$/HF | 2.80 ± 0.68 |
| Danang urban soils | 1995 | [83] | 0–20 | <0.063 | HF/HNO$_3$/HCl | 3.99/1.40–4.48 |
| Bangkok urban soils | 1996 | [84] | 0–5 | <2 | HNO$_3$/HClO$_4$ | 1.84/0.39–2.67 |
| Sydney roadside soils | 2009 | [36] | 0–10 | <0.425 | HNO$_3$/HCl | 3.11/0.62–4.72 |
| Suva(Fiji)roadside soil | 2015 | [88] | 0–5 | none | HNO$_3$/HCl | 3.95/2.93–8.67 |
| Ottawa garden soils | 1993 | [86] | 0–5 | 0.1–0.25 | HNO$_3$/HF/HClO$_4$ | 2.08/1.53–2.81 |
| Havana urban soils | 2018 | [51] | horizons | <2 | HNO$_3$/HCl | 2.55 ± 1.00 |

**Table A3.** Phosphorus in urban soils.

| | sampling year | reference | Sampling depth cm | grain size mm | Digestion | P mg/kg |
|---|---|---|---|---|---|---|
| Upper crust | | [136] | | | | 654 |
| Continental crust | | [40] | | | | 757 |
| Aberdeen parkland | 1994 | [65] | 0–10 | <2 | HNO$_3$/HCl | 957 ± 170 |
| Aberdeen roadside | 1994 | [65] | 0–10 | <2 | HNO$_3$/HCl | 861 ± 70 |
| Warsaw urban soils | 1996 | [7] | 0–20 | <1 | 450°/HCl | 53/18–163 |
| Zadar urban soils | 2003/2004 | [73] | 0–10 | <2 | HNO$_3$/HCl | 796/309–2719 |
| Salerno urban soils | 2018 | [75] | 0–20 | <2 | HNO$_3$/HCl | 1000/400–2600 |
| Ottawa garden soils | 1993 | [86] | 0–5 | 0.1–0.25 | HNO$_3$/HF/HClO$_4$ | 1160/570–1909 |
| New York Parks | 1999 | [20] | 0–15 | <2 | HNO$_3$/HCl | 643/198–8158 |

**Table A4.** Boron in urban soils.

| Location | sampling year | reference | Sampling depth cm | grain size mm | | **B mg/kg** |
|---|---|---|---|---|---|---|
| Upper crust | | [136] | | | | |
| Continental crust | | [40] | | | | 11 |
| Tallinn | 1987–90 | [67] | 0–10 | <1 | XRF | 20/<10–100 |
| Vilnius central | 2000 | [68] | 0–10 | <1 | DC-arc | 21.3 |
| Vilnius peripheral | 2000 | [68] | 0–10 | <1 | DC-arc | 24.5 |
| Siauliai | 2000 | [68] | 0–10 | <1 | DC-arc | 28.7 |
| Mazeikiai | 2000 | [68] | 0–10 | <1 | DC-arc | 25.1 |
| Joniskis | 2000 | [68] | 0–10 | <1 | DC-arc | 24.9 |
| Berlin low-D residential | 1993–96 | [29] | 0–20 | <2 | XRF | 14/max 820 |
| Berlin high-D residential | 1993–96 | [29] | 0–20 | <2 | XRF | 15/max 46 |
| Berlin allotment | 1993–96 | [29] | 0–20 | <2 | XRF | 16/max 61 |
| Berlin industrial | 1993–96 | [29] | 0–20 | <2 | XRF | 15/max 570 |
| Zadar | 2003/04 | [73] | 0–10 | <2 | $HNO_3/HCl$ | 19.5/15.3–36.7 |
| Salerno urban soils | 2018 | [75] | 0–20 | <2 | $HNO_3/HCl$ | 10/5–17.4 |
| Beijing | 2008 | [78] | 0–20 | fine soil | $HNO_3/HClO_4/HF$ | 8.1 ± 0.9 |

**Table A5.** Silver in urban soils.

| Location | sampling year | reference | Sampling depth cm | grain size mm | | **Ag mg/kg** |
|---|---|---|---|---|---|---|
| Continental crust | | [40] | | | | 0.07 |
| Tallinn urban soils | 1987–90 | [67] | 0–10 | <1 | XRF | 0.15/<0.1–3 |
| Vilnius central | 2000 | [68] | 0–10 | <1 | DC-arc | 0.31 |
| Vilnius peripheral | 2000 | [68] | 0–10 | <1 | DC-arc | 0.12 |
| Siauliai urban soils | 2000 | [68] | 0–10 | <1 | DC-arc | 0.09 |
| Mazeikiai urban soils | 2000 | [68] | 0–10 | <1 | DC-arc | 0.07 |
| Joniskis urban soils | 2000 | [68] | 0–10 | <1 | DC-arc | 0.08 |
| Salerno urban soils | 2018 | [75] | 0–20 | <2 | $HNO_3/HCl$ | 0.09/0.02–0.72 |
| Xuzhou urban soils | 2004 | [80] | 0–10 | <2 | $HF/HNO_3/HCl$-XRF | 0.19/0.06–0.72 |
| Ottawa garden soils | 1993 | [86] | 0–5 | 0.1–0.25 | $HNO_3/HF/HClO_4$ | 0.30/0.20–0.43 |
| New York Parks | 1999 | [20] | 0–15 | <2 | $HNO_3/HCl$ | 0.2/0.02–15.0 |

**Table A6.** Arsenic in urban soils.

| Location | sampling year | reference | sampling depth cm | grain size mm | | **As mg/kg** |
|---|---|---|---|---|---|---|
| Upper crust | | [136] | | | | 5.7 |
| Continental crust | | [40] | | | | 1.7 |
| Trondheim urban soils | 1994 | [66] | 0–2 | <2 | $HNO_3/HCl$ | 2.8/0.5–83 |
| Trondheim urban soils | 2004 | [66] | 0–2 | <2 | $1/1\ HNO_3$ | 3.3/0.3–23 |
| Uppsala | 2003 | [8] | 0–5 | <2 | $HNO_3/HCl$ | 3.5/1.4–15.0 |
| Uppsala | 2003 | [8] | 5–10 | <2 | $HNO_3/HCl$ | 3.9/1.4–16.2 |
| Uppsala | 2003 | [8] | 10–20 | <2 | $HNO_3/HCl$ | 3.9/1.3–23.7 |
| Kielce-residential | 2016 | [69] | 0–20 | <2 | $HNO_3$ (HF?) | 2.3/1.1–3.4 |
| Kielce road soils | 2016 | [69] | 0–20 | <2 | $HNO_3$ (HF?) | 1.2/0.5–2.8 |
| Kielce urban greenery | 2016 | [69] | 0–20 | <2 | $HNO_3$ (HF?) | 2.5/0.3–8.4 |
| Kielce allotment gardens | 2016 | [69] | 0–20 | <2 | $HNO_3$ (HF?) | 2.7/0.3–10.0 |
| Kielce agricultural areas | 2016 | [69] | 0–20 | <2 | $HNO_3$ (HF?) | 1.3/0.5–3.7 |
| Berlin low-D residential | 1993–96 | [29] | 0–20 | <2 | unknown | 3.4/max 58.6 |
| Berlin high-D residential | 1993–96 | [29] | 0–20 | <2 | unknown | 4.3/max 42.3 |
| Berlin allotment | 1993–96 | [29] | 0–20 | <2 | unknown | 3.7/max 18.7 |
| Berlin industrial | 1993–96 | [29] | 0–20 | <2 | unknown | 4.1/max 126 |
| Vienna roadside soils | 1997 | [70] | 0–10 | none | $HNO_3/HCl$ | 8.2/3.9–10.5 |
| Vienna parks | 1997 | [70] | 0–10 | none | $HNO_3/HCl$ | 8.8/4.7–12.1 |
| Vienna urban soils | 2003 | [16] | 0–10 | <2 | $HNO_3/HCl$ | 8.1 ± 1.2 |
| Novi Sad urban soils | 2017 | [72] | 0–15 | <2 | $HNO_3/H_2O_2$ | 6.3/2.1–11.1 |
| Zadar urban soils | 2003–04 | [73] | 0–10 | <2 | $HNO_3/HCl$ | 12.4/5.0–18.7 |
| Salerno urban soils | 2018 | [75] | 0–20 | <2 | $HNO_3/HCl$ | 10.4/4.3–17.3 |
| Sangareddy urban soils | 2019 | [85] | 0–10 | <0.074 | XRF | 3.65/2.3–4.8 |
| Tyumen urban soils | 2016 | [64] | 0–10 | <1 | XRF | 7.7/1.5–81 |
| Beijing parks | 2018 | [79] | 0–5 | <0.125 | $HNO_3/HCl$ | 12/1–26 |
| Xian urban soils | 2016 | [1] | 0–10 | none | XRF | 12.7/7.5–14.5 |
| Xuzhou urban soils | 2004 | [80] | 0–10 | <2 | $HF/HNO_3/HCl$-XRF | 13/9–17 |
| Hangzhou Industrial | 2019 | [63] | 0–30 | <0.15 | $HNO_3/HClO_4/HF$ | 9.0/6.7–13.3 |
| Hangzhou Roadside | 2019 | [63] | 0–30 | <0.15 | $HNO_3/HClO_4/HF$ | 9.9/6.7–15.6 |
| Hangzhou Residential | 2019 | [63] | 0–30 | <0.15 | $HNO_3/HClO_4/HF$ | 0.16/0.11–0.23 |

**Table A6.** *Cont.*

| | | | | | | |
|---|---|---|---|---|---|---|
| Ottawa garden soils | 1993 | [86] | 0–5 | 0.1–0.25 | $HNO_3/HF/HClO_4$ | 2.8/1.7–4.4 |
| New York Parks | 1999 | [20] | 0–15 | <2 | $HNO_3/HCl$ | 13/1–46 |
| Clay County urban soils | 2016/18 | [87] | 0–15 | <2 | $HNO_3/H_2O_2$ | 0.81/0.20–1.68 |
| Ocala urban soils | 2016/18 | [87] | 0–15 | <2 | $HNO_3/H_2O_2$ | 1.42/0.40–4.78 |
| Orlando urban soils | 2016/18 | [87] | 0–15 | <2 | $HNO_3/H_2O_2$ | 0.95/0.17–2.41 |
| Pensacola urban soils | 2016/18 | [87] | 0–15 | <2 | $HNO_3/H_2O_2$ | 3.52/0.86–24.2 |
| Tampa urban soils | 2016/18 | [87] | 0–15 | <2 | $HNO_3/H_2O_2$ | 1.01/0.05–4.52 |
| West Palm Beach urban soils | 2016/18 | [87] | 0–15 | <2 | $HNO_3/H_2O_2$ | 1.63/0.26–6.16 |
| Havana urban soils | 2018 | [51] | horizons | <2 | $HNO_3/HCl$ | 8.11 ± 3.9 |
| Ibadan | 2006 | [89] | 0–15 cm | <0.075 | $HNO_3/HCl$ | 3.0/<–22 |

**Table A7.** Barium in urban soils.

| Location | sampling year | reference | sampling depth cm | grain size mm | | Ba mg/kg |
|---|---|---|---|---|---|---|
| Upper crust | | [136] | | | | 628 |
| Continental crust | | [40] | | | | 584 |
| Aberdeen parkland | 1994 | [65] | 0–10 | <2 | $HNO_3/HCl$ | 99 ± 15 |
| Aberdeen roadside | 1994 | [65] | 0–10 | <2 | $HNO_3/HCl$ | 204 ± 43 |
| Tallinn urban soils | 1987–90 | [67] | 0–10 | <1 | XRF | 250/<100–600 |
| Vilnius central | 2000 | [68] | 0–10 | <1 | DC-arc | 433 |
| Vilnius peripheral | 2000 | [68] | 0–10 | <1 | DC-arc | 412 |
| Siauliai urban soils | 2000 | [68] | 0–10 | <1 | DC-arc | 381 |
| Mazeikiai urban soils | 2000 | [68] | 0–10 | <1 | DC-arc | 332 |
| Joniskis urban soils | 2000 | [68] | 0–10 | <1 | DC-arc | 401 |
| Kielce residential | 2016 | [69] | 0–20 | <2 | $HNO_3$ (HF?) | 52/12–129 |
| Kielce road soils | 2016 | [69] | 0–20 | <2 | $HNO_3$ (HF?) | 39/23–58 |
| Kielce urban greenery | 2016 | [69] | 0–20 | <2 | $HNO_3$ (HF?) | 43/7–171 |
| Kielce allotment gardens | 2016 | [69] | 0–20 | <2 | $HNO_3$ (HF?) | 55/16–135 |
| Kielce agricultural areas | 2016 | [69] | 0–20 | <2 | $HNO_3$ (HF?) | 40/8–169 |
| Zadar urban soils | 2003/04 | [73] | 0–10 | <2 | $HNO_3/HCl$ 3:1 | 146/115–409 |
| Vigo parks-gardens | 2013 | [12] | 0–20 | <2 | XRF | 528/238–1145 |
| Salerno urban soils | 2018 | [75] | 0–20 | <2 | $HNO_3/HCl$ | 284/96–556 |
| Xian urban soil | 2016 | [1] | 0–10 | none | XRF | 560/495–896 |
| Xuzhou urban soils | 2004 | [80] | 0–10 | <2 | $HF/HNO_3/HCl$-XRF | 470/425–614 |
| Ottawa garden soils | 1993 | [86] | 0–5 | 0.1–0.25 | $HNO_3/HF/HClO_4$ | 772/609–854 |
| New York Parks | 1999 | [20] | 0–15 | <2 | $HNO_3/HCl$ | 106/39–307 |
| Clay County urban soils | 2016/18 | [87] | 0–15 | <2 | $HNO_3/H_2O_2$ | 23.4/4.9–52.7 |
| Ocala urban soils | 2016/18 | [87] | 0–15 | <2 | $HNO_3/H_2O_2$ | 119/23.9–346 |
| Orlando urban soils | 2016/18 | [87] | 0–15 | <2 | $HNO_3/H_2O_2$ | 20.3/2.0–58.8 |
| Pensacola urban soils | 2016/18 | [87] | 0–15 | <2 | $HNO_3/H_2O_2$ | 48.1/6.6–322 |
| Tampa urban soils | 2016/18 | [87] | 0–15 | <2 | $HNO_3/H_2O_2$ | 23.7/4.1–259 |
| West Palm Beach urban soils | 2016/18 | [87] | 0–15 | <2 | $HNO_3/H_2O_2$ | 29.1/8.1–129 |
| Mexico | 2008 | [24] | 0–10 | <0.074 | XRF | 505/321–1098 |

**Table A8.** Cadmium in urban soils.

| Location | sampling year | reference | sampling depth cm | grain size mm | | Cd mg/kg |
|---|---|---|---|---|---|---|
| Upper crust | | [136] | | | | 0.06 |
| Continental crust | | [40] | | | | 0.10 |
| Trondheim urban soils | 1994 | [66] | 0–2 | <2 | $HNO_3/HCl$ | 0.16/<0.01–11.3 |
| Trondheim urban soils | 2004 | [66] | 0–2 | <2 | 1/1 $HNO_3$ | 0.12/0.002–5.6 |
| Uppsala | 2003 | [8] | 0–5 | <2 | $HNO_3/HCl$ | 0.21/0.08–0.71 |
| Uppsala | 2003 | [8] | 5–10 | <2 | $HNO_3/HCl$ | 0.21/0.09–0.99 |
| Uppsala | 2003 | [8] | 10–20 | <2 | $HNO_3/HCl$ | 0.22/0.07–0.40 |
| Warsaw urban soils | 1996 | [7] | 0–20 | <1 | 450°/HCl | 1.0/<–5.5 |
| Kielce residential | 2016 | [69] | 0–20 | <2 | $HNO_3$ (HF?) | 0.104/0.001–0.23 |
| Kielce road soils | 2016 | [69] | 0–20 | <2 | $HNO_3$ (HF?) | 0.141/0.032–0.37 |
| Kielce urban greenery | 2016 | [69] | 0–20 | <2 | $HNO_3$ (HF?) | 0.081/0.001–0.21 |
| Kielce allotment gardens | 2016 | [69] | 0–20 | <2 | $HNO_3$ (HF?) | 0.182/0.079–0.49 |
| Kielce agricultural areas | 2016 | [69] | 0–20 | <2 | $HNO_3$ (HF?) | 0.050/0.001–0.19 |
| Berlin low-D residential | 1993–96 | [29] | 0–20 | <2 | unknown | 0.21/max 6.7 |
| Berlin high-D residential | 1993–96 | [29] | 0–20 | <2 | unknown | 0.41/max 20.3 |
| Berlin allotment | 1993–96 | [29] | 0–20 | <2 | unknown | 0.32/max 2.5 |
| Berlin industrial | 1993–96 | [29] | 0–20 | <2 | unknown | 0.50/max 131 |
| Vienna roadside soils | 1997 | [70] | 0–10 | none | $HNO_3/HCl$ | 0.6/0.2–3.4 |
| Vienna parks | 1997 | [70] | 0–10 | none | $HNO_3/HCl$ | 0.55/0.3–1.3 |
| Vienna urban soils | 2003 | [16] | 0–10 | <2 | $HNO_3/HCl$ | 0.5 ± 0.2 |
| Sopron urban soils | 2012 | [39] | 0–10 | none | $HNO_3/H_2O_2$ | 1.62/0.37–6.74 |
| Sopron urban soils | 2012 | [39] | 0–20 | none | $HNO_3/H_2O_2$ | 1.52/0.17–6.14 |
| Zadar urban soils | 2003/04 | [73] | 0–10 | <2 | $HNO_3/HCl$ | 1.06/0.16–3.24 |

**Table A8.** *Cont.*

| | | | | | | |
|---|---|---|---|---|---|---|
| Salerno urban soils | 2018 | [75] | 0–20 | <2 | HNO$_3$/HCl | 0.42/0.15–1.07 |
| Palermo public parks | 2000 | [31] | 0–10 | <2 | HNO$_3$/HCl | 0.68/0.27–1.86 |
| Seoul-Uijeongbu urban soils | 1995 | [58] | 0–15 | <0.18 | HNO$_3$/HClO$_4$ | 1.4/0.7–3.8 |
| Seoul-Koyang urban soils | 1995 | [58] | 0–15 | <0.18 | HNO$_3$/HClO$_4$ | 2.0/1.3–3.7 |
| Shenyang urban soils | 2008 | [77] | 0–5 | <1 | HNO$_3$/HClO$_4$ | 0.04–2.08 |
| Beijing roadside soil | 2008 | [78] | 0–20 | fine soil | HNO$_3$/HClO$_4$/HF | 0.215 ± 0.070 |
| Beijing Parks | 2018 | [79] | 0–5 | <0.125 | HNO$_3$/HCl | 0.47/0.17–0.87 |
| Xuzhou urban soils | 2004 | [80] | 0–10 | <2 | HF/HNO$_3$/HCl-XRF | 0.37/0.11–1.00 |
| Hangzhou Industrial | 2009 | [23] | 0–10 | <2 | HNO$_3$/HClO$_4$/HF | 1.99 ± 0.65 |
| Hangzhou Roadside | 2009 | [23] | 0–10 | <2 | HNO$_3$/HClO$_4$/HF | 1.31 ± 0.29 |
| Hangzhou Residential | 2009 | [23] | 0–10 | <2 | HNO$_3$/HClO$_4$/HF | 1.08 ± 0.21 |
| Hangzhou Parks | 2009 | [23] | 0–10 | <2 | HNO$_3$/HClO$_4$/HF | 0.94 ± 0.13 |
| Hangzhou Industrial | 2019 | [63] | 0–30 | <0.15 | HNO$_3$/HClO$_4$/HF | 0.16/0.10–0.26 |
| Hangzhou Roadside | 2019 | [63] | 0–30 | <0.15 | HNO$_3$/HClO$_4$/HF | 0.20/0.14–0.41 |
| Hangzhou Residential | 2019 | [63] | 0–30 | <0.15 | HNO$_3$/HClO$_4$/HF | 0.16/0.11–0.23 |
| Shanghai urban soils | 2006 | [33] | 0–10 | <0.125 | HF/HNO$_3$/HClO$_4$ | 0.52/0.19–3.66 |
| Hong Kong urban parks | 2000 | [82] | 0–10 | <2 | HNO$_3$ | 2.18 ± 1.02 |
| Hong Kong Urban | 2004 | [81] | 0–15 | <2 | HNO$_3$/HClO$_4$ | 0.33/0.11–1.36 |
| Hong Kong suburban | 2004 | [81] | 0–15 | <2 | HNO$_3$/HClO$_4$ | 0.31/0.23–0.80 |
| Hong Kong country park | 2004 | [81] | 0–15 | <2 | HNO$_3$/HClO$_4$ | 0.32/0.20–0.58 |
| Danang urban soils | 1995 | [83] | 0–20 | <0.063 | HF/HNO$_3$/HCl | 0.4/0.1–4.6 |
| Bangkok urban soils | 1996 | [84] | 0–5 | <2 | HNO$_3$/HClO$_4$ | 0.15/0.05–2.53 |
| Sydney roadside soils | 2009 | [36] | 0–10 | <0.425 | HNO$_3$/HCl | 0.18/0.01–0.49 |
| Adelaide garden soils | 2017 | [14] | 0–10 | <20 | HNO$_3$/HClO$_4$/HCl | <0.1/<0.1–0.38 |
| Suva(Fiji)roadside soil | 2015 | [88] | 0–5 | none | HNO$_3$/HCl | 3.1/2.0–6.2 |
| Ottawa garden soils | 1993 | [86] | 0–5 | 0.1–0.25 | HNO$_3$/HF/HClO$_4$ | 0.27/0.11–0.59 |
| New York Parks | 1999 | [20] | 0–15 | <2 | HNO$_3$/HCl | 0.4/0.1–3.0 |
| Clay County urban soils | 2016/18 | [87] | 0–15 | <2 | HNO$_3$/H$_2$O$_2$ | 0.12/0.01–0.58 |
| Ocala urban soils | 2016/18 | [87] | 0–15 | <2 | HNO$_3$/H$_2$O$_2$ | 0.33/0.01–1.21 |
| Orlando urban soils | 2016/18 | [87] | 0–15 | <2 | HNO$_3$/H$_2$O$_2$ | 0.16/0.01–1.60 |
| Pensacola urban soils | 2016/18 | [87] | 0–15 | <2 | HNO$_3$/H$_2$O$_2$ | 0.27/0.01–1.67 |
| Tampa urban soils | 2016/18 | [87] | 0–15 | <2 | HNO$_3$/H$_2$O$_2$ | 0.18/0.01–1.54 |
| West Palm Beach urban soils | 2016/18 | [87] | 0–15 | <2 | HNO$_3$/H$_2$O$_2$ | 0.31/0.03–1.33 |
| Havana urban soils | 2018 | [51] | horizons | <2 | HNO$_3$/HCl | 0.62 ± 0.56 |
| Ibadan urban soils | 2006 | [89] | 0–15 | <0.075 | HNO$_3$/HCl | 0.15/0.1–69 |

**Table A9.** Cobalt in urban soils.

| Location | sampling year | reference | sampling depth cm | grain size mm | | Co mg/kg |
|---|---|---|---|---|---|---|
| Upper crust | | [136] | | | | 15 |
| Continental crust | | [40] | | | | 24 |
| Aberdeen parkland | 1994 | [65] | 0–10 | <2 | HNO$_3$/HCl | 6.4 ± 0.7 |
| Aberdeen roadside | 1994 | [65] | 0–10 | <2 | HNO$_3$/HCl | 6.2 ± 0.3 |
| Tallinn urban soils | 1987–90 | [67] | 0–10 | <1 | XRF | 5/<3–50 |
| Vilnius central | 2000 | [68] | 0–10 | <1 | DC-arc | 3.6 |
| Vilnius peripheral | 2000 | [68] | 0–10 | <1 | DC-arc | 3.9 |
| Siauliai urban soils | 2000 | [68] | 0–10 | <1 | DC-arc | 6.2 |
| Mazeikiai urban soils | 2000 | [68] | 0–10 | <1 | DC-arc | 4.1 |
| Joniskis urban soils | 2000 | [68] | 0–10 | <1 | DC-arc | 5.0 |
| Kielce residential | 2016 | [69] | 0–20 | <2 | HNO$_3$ (HF?) | 1.5/0.5–3.0 |
| Kielce road soils | 2016 | [69] | 0–20 | <2 | HNO$_3$ (HF?) | 1.0/0.5–1.9 |
| Kielce urban greenery | 2016 | [69] | 0–20 | <2 | HNO$_3$ (HF?) | 1.1/0.2–2.5 |
| Kielce allotment gardens | 2016 | [69] | 0–20 | <2 | HNO$_3$ (HF?) | 1.7/0.5–3.7 |
| Kielce agricultural areas | 2016 | [69] | 0–20 | <2 | HNO$_3$ (HF?) | 0.8/0.2–1.6 |
| Vienna roadside soils | 1997 | [70] | 0–10 | none | HNO$_3$/HCl | 7.8/4.5–9.8 |
| Vienna parks | 1997 | [70] | 0–10 | none | HNO$_3$/HCl | 8.0/5.2–9.8 |
| Vienna urban soils | 2003 | [16] | 0–10 | <2 | HNO$_3$/HCl | 7.9 ±1.1 |
| Sopron urban soils | 2012 | [39] | 0–10 | none | HNO$_3$/H$_2$O$_2$ | 20.6/3.6–64.2 |
| Sopron urban soils | 2012 | [39] | 0–20 | none | HNO$_3$/H$_2$O$_2$ | 21.1/5.4–55.9 |
| Szeged urban soils | 2005 | [17] | 0–10 | none | HNO$_3$/HCl | 3.5/0.2–8.5 |
| Novi Sad urban soils | 2017 | [72] | 0–15 | <2 | HNO$_3$/H$_2$O$_2$ | 7.2/3.5–11.2 |
| Zadar urban soils | 2003/04 | [73] | 0–10 | <2 | HNO$_3$/HCl | 12.9/2.4–19.2 |
| Salerno urban soils | 2018 | [75] | 0–20 | <2 | HNO$_3$/HCl | 8.0/3.8–13.0 |
| Palermo public parks | 2000 | [31] | 0–10 | <2 | HNO$_3$/HCl | 5.2/1.5–14.8 |
| Tyumen urban soils | 2016 | [64] | 0–10 | <1 | XRF | 19.5/3.5–55.6 |
| Seoul-Uijeongbu | 1995 | [58] | 0–15 | <0.18 | HNO$_3$/HClO$_4$ | 9.6/1.2–20 |
| Seoul-Koyang | 1995 | [58] | 0–15 | <0.18 | HNO$_3$/HClO$_4$ | 23.2/13.2–39.2 |
| Xian urban soils | 2016 | [1] | 0–10 | none | XRF | 23.0/13.8–53.0 |
| Xuzhou urban soils | 2004 | [80] | 0–10 | <2 | HF/HNO$_3$/HCl-XRF | 11/9–19 |

**Table A9.** *Cont.*

| Hangzhou Industrial | 2019 | [63] | 0–30 | <0.15 | $HNO_3/HClO_4/HF$ | 15.0/12.1–16.8 |
|---|---|---|---|---|---|---|
| Hangzhou Roadside | 2019 | [63] | 0–30 | <0.15 | $HNO_3/HClO_4/HF$ | 15.1/12.8–16.4 |
| Hangzhou Residential | 2019 | [63] | 0–30 | <0.15 | $HNO_3/HClO_4/HF$ | 14.4/10.8–17.1 |
| Hong Kong Urban | 2004 | [81] | 0–15 | <2 | $HNO_3/HClO_4$ | 3.3/0.6–10.9 |
| Hong Kong suburban | 2004 | [81] | 0–15 | <2 | $HNO_3/HClO_4$ | 2.9/1.7–16.3 |
| Hong Kong country park | 2004 | [81] | 0–15 | <2 | $HNO_3/HClO_4$ | 2.7/1.4–8.1 |
| Danang urban soils | 1995 | [83] | 0–20 | <0.063 | $HF/HNO_3/HCl$ | 17/7–217 |
| Suva(Fiji)roadside soil | 2015 | [88] | 0–5 | none | $HNO_3/HCl$ | 33/24–38 |
| Ottawa garden soils | 1993 | [86] | 0–5 | 0.1–0.25 | $HNO_3/HF/HClO_4$ | 8.1/5.6–11.6 |
| New York Parks | 1999 | [20] | 0–15 | <2 | $HNO_3/HCl$ | 7/3–13 |
| Clay County urban soils | 2016/18 | [87] | 0–15 | <2 | $HNO_3/H_2O_2$ | 0.64/0.11–2.11 |
| Ocala urban soils | 2016/18 | [87] | 0–15 | <2 | $HNO_3/H_2O_2$ | 1.54/0.32–5.48 |
| Orlando urban soils | 2016/18 | [87] | 0–15 | <2 | $HNO_3/H_2O_2$ | 0.41/0.07–2.28 |
| Pensacola urban soils | 2016/18 | [87] | 0–15 | <2 | $HNO_3/H_2O_2$ | 1.04/0.32–1.97 |
| Tampa urban soils | 2016/18 | [87] | 0–15 | <2 | $HNO_3/H_2O_2$ | 0.51/0.06–1.41 |
| West Palm Beach urban soils | 2016/18 | [87] | 0–15 | <2 | $HNO_3/H_2O_2$ | 0.78/0.20–2.62 |
| Mexico | 2008 | [24] | 0–10 | <0.074 | XRF | 19/11–42 |
| Havana urban soils | 2018 | [51] | horizons | <2 | $HNO_3/HCl$ | 11.7 ± 7.8 |

**Table A10.** Cr in urban soils.

| Location | sampling year | reference | sampling depth cm | grain size mm | | Cr mg/kg |
|---|---|---|---|---|---|---|
| Upper crust | | [136] | | | | 73 |
| Continental crust | | [40] | | | | 126 |
| Aberdeen parkland | 1994 | [65] | 0–10 | <2 | $HNO_3/HCl$ | 23.9 ± 2.3 |
| Aberdeen roadside | 1994 | [65] | 0–10 | <2 | $HNO_3/HCl$ | 22.9 ± 2.8 |
| Trondheim urban soils | 1994 | [66] | 0–2 | <2 | $HNO_3/HCl$ | 69/8–199 |
| Trondheim urban soils | 2004 | [66] | 0–2 | <2 | $1/1\ HNO_3$ | 58/23–296 |
| Uppsala | 2003 | [8] | 0–5 | <2 | $HNO_3/HCl$ | 32/14–62 |
| Uppsala | 2003 | [8] | 5–10 | <2 | $HNO_3/HCl$ | 38/13–61 |
| Uppsala | 2003 | [8] | 10–20 | <2 | $HNO_3/HCl$ | 43/14 - 65 |
| Tallinn urban soils | 1987–90 | [67] | 0–10 | <1 | XRF | 33/<6–300 |
| Vilnius central | 2000 | [68] | 0–10 | <1 | DC-arc | 33.8 |
| Vilnius peripheral | 2000 | [68] | 0–10 | <1 | DC-arc | 32.8 |
| Siauliai urban soils | 2000 | [68] | 0–10 | <1 | DC-arc | 41.9 |
| Mazeikiai urban soils | 2000 | [68] | 0–10 | <1 | DC-arc | 32.5 |
| Joniskis urban soils | 2000 | [68] | 0–10 | <1 | DC-arc | 49.6 |
| Warsaw urban soils | 1996 | [7] | 0–20 | <1 | 450°/HCl | 13/5–70 |
| Kielce residential | 2016 | [69] | 0–20 | <2 | $HNO_3$ (HF?) | 7.9/3.2–12.9 |
| Kielce road soils | 2016 | [69] | 0–20 | <2 | $HNO_3$ (HF?) | 6.8/3.8–12.5 |
| Kielce urban greenery | 2016 | [69] | 0–20 | <2 | $HNO_3$ (HF?) | 6.9/1.4–19.1 |
| Kielce allotment gardens | 2016 | [69] | 0–20 | <2 | $HNO_3$ (HF?) | 7.9/2.6–18.4 |
| Kielce agricultural areas | 2016 | [69] | 0–20 | <2 | $HNO_3$ (HF?) | 3.6/0.8–6.8 |
| Berlin low-D residential | 1993–96 | [29] | 0–20 | <2 | XRF | 22/max 214 |
| Berlin high-D residential | 1993–96 | [29] | 0–20 | <2 | XRF | 27/max 168 |
| Berlin allotment | 1993–96 | [29] | 0–20 | <2 | XRF | 23/max 135 |
| Berlin industrial | 1993–96 | [29] | 0–20 | <2 | XRF | 27/max 1840 |
| Vienna roadside soils | 1997 | [70] | 0–10 | none | $HNO_3/HCl$ | 30/10–68 |
| Vienna parks | 1997 | [70] | 0–10 | none | $HNO_3/HCl$ | 26/14–46 |
| Vienna urban soils | 2003 | [16] | 0–10 | <2 | $HNO_3/HCl$ | 36 ± 8 |
| Szeged urban soils | 2005 | [17] | 0–10 | none | $HNO_3/HCl$ | 53/41–69 |
| Novi Sad urban soils | 2017 | [72] | 0–15 | <2 | $HNO_3/H_2O_2$ | 28.4/10.6–51 |
| Zadar urban soils | 2003/04 | [73] | 0–10 | <2 | $HNO_3/HCl$ | 80/42–158 |
| Torino urban soils | 1999 | [6] | 0–20 | <2 | $HNO_3/HCl$ | 157/67–870 |
| Salerno urban soils | 2018 | [75] | 0–20 | <2 | $HNO_3/HCl$ | 17.2/10.9–35 |
| Palermo public parks | 2000 | [31] | 0–10 | <2 | $HNO_3/HCl$ | 34/12–100 |
| Sevilla parks-gardens | 2000 | [9] | 0–20 | <2 | $HNO_3/HCl$ | 42/24–67 |
| Vigo parks-gardens | 2013 | [12] | 0–20 | <2 | XRF | 65/33–195 |
| Tyumen urban soils | 2016 | [64] | 0–10 | <1 | XRF | 107/25–348 |
| Sangareddy urban soil | 2019 | [85] | 0–10 | <0.074 | XRF | 198/158–482 |
| Seoul-Uijeongbu urban soils | 1995 | [58] | 0–15 | <0.18 | $HNO_3/HClO_4$ | 23/4–85 |
| Seoul-Koyang urban soils | 1995 | [58] | 0–15 | <0.18 | $HNO_3/HClO_4$ | 45/23–85 |
| Beijing roadside soil | 2008 | [78] | 0–20 | fine soil | $HNO_3/HClO_4/HF$ | 61.9 ± 2.3 |
| Beijing Parks | 2018 | [79] | 0–5 | <0.125 | $HNO_3/HCl$ | 53/21–489 |
| Xian urban soils | 2016 | [1] | 0–10 | none | XRF | 69/58–148 |
| Xuzhou urban soils | 2004 | [80] | 0–10 | <2 | $HF/HNO_3/HCl$-XRF | 71/63–102 |
| Hangzhou Industrial | 2019 | [63] | 0–30 | <0.15 | $HNO_3/HClO_4/HF$ | 102/80–153 |
| Hangzhou Roadside | 2019 | [63] | 0–30 | <0.15 | $HNO_3/HClO_4/HF$ | 99/85–120 |
| Hangzhou Residential | 2019 | [63] | 0–30 | <0.15 | $HNO_3/HClO_4/HF$ | 95/77–110 |
| Shanghai urban soils | 2006 | [33] | 0–10 | <0.125 | $HF/HNO_3/HClO_4$ | 108/26–233 |
| Hong Kong Urban | 2004 | [81] | 0–15 | <2 | $HNO_3/HClO_4$ | 16.8/2.6–51.4 |

**Table A10.** *Cont.*

| | | | | | | |
|---|---|---|---|---|---|---|
| Hong Kong suburban | 2004 | [81] | 0–15 | <2 | HNO$_3$/HClO$_4$ | 19.7/10.1–49 |
| Hong Kong country park | 2004 | [81] | 0–15 | <2 | HNO$_3$/HClO$_4$ | 20.2/13.7–48 |
| Danang urban soils | 1995 | [83] | 0–20 | <0.063 | HF/HNO$_3$/HCl | 104/23–175 |
| Bangkok urban soils | 1996 | [84] | 0–5 | <2 | HNO$_3$/HClO$_4$ | 25.4/4.3–57.4 |
| Sydney roadside soils | 2009 | [36] | 0–10 | <0.425 | HNO$_3$/HCl | 35/10–79 |
| Suva(Fiji)roadside soil | 2015 | [88] | 0–5 | none | HNO$_3$/HCl | 34/14–63 |
| Ottawa garden soils | 1993 | [86] | 0–5 | 0.1–0.25 | HNO$_3$/HF/HClO$_4$ | 43.8/28.8–59 |
| New York Parks | 1999 | [20] | 0–15 | <2 | HNO$_3$/HCl | 37/16–71 |
| Clay County urban soils | 2016/18 | [87] | 0–15 | <2 | HNO$_3$/H$_2$O$_2$ | 9.9/3.5–24.0 |
| Ocala urban soils | 2016/18 | [87] | 0–15 | <2 | HNO$_3$/H$_2$O$_2$ | 56.7/6.4–289 |
| Orlando urban soils | 2016/18 | [87] | 0–15 | <2 | HNO$_3$/H$_2$O$_2$ | 9.6/1.9–28.2 |
| Pensacola urban soils | 2016/18 | [87] | 0–15 | <2 | HNO$_3$/H$_2$O$_2$ | 12.6/6.0–24.4 |
| Tampa urban soils | 2016/18 | [87] | 0–15 | <2 | HNO$_3$/H$_2$O$_2$ | 11.1/1.5–41.5 |
| West Palm Beach urban soils | 2016/18 | [87] | 0–15 | <2 | HNO$_3$/H$_2$O$_2$ | 15.9/5.0–86.9 |
| Mexico | 2008 | [24] | 0–10 | <0.074 | XRF | 135/65–559 |
| Havana urban soils | 2018 | [51] | horizons | <2 | HNO$_3$/HCl | 82.9 ± 119.3 |
| Ibadan urban soils | 2006 | [89] | 0–15 | <0.075 | HNO$_3$/HCl | 56/10–436 |

**Table A11.** Copper in urban soils.

| | sampling year | reference | sampling depth cm | grain size mm | | Cu mg/kg |
|---|---|---|---|---|---|---|
| Upper crust | | [136] | | | | 27 |
| Continental crust | | [40] | | | | 25 |
| Aberdeen parkland | 1994 | [65] | 0–10 | <2 | HNO$_3$/HCl | 27 ± 64 |
| Aberdeen roadside | 1994 | [65] | 0–10 | <2 | HNO$_3$/HCl | 45 ± 11 |
| Trondheim urban soils | 1994 | [66] | 0–2 | <2 | HNO$_3$/HCl | 35/1.7–706 |
| Trondheim urban soils | 2004 | [66] | 0–2 | <2 | 1/1 HNO$_3$ | 32/5–383 |
| Uppsala | 2003 | [8] | 0–5 | <2 | HNO$_3$/HCl | 25/11–110 |
| Uppsala | 2003 | [8] | 5–10 | <2 | HNO$_3$/HCl | 26/13–356 |
| Uppsala | 2003 | [8] | 10–20 | <2 | HNO$_3$/HCl | 26/11–54 |
| Tallinn urban soils | 1987–90 | [67] | 0–10 | <1 | XRF | 35/7–621 |
| Vilnius central | 2000 | [68] | 0–10 | <1 | DC-arc | 18.5 |
| Vilnius peripheral | 2000 | [68] | 0–10 | <1 | DC-arc | 13.6 |
| Siauliai urban soils | 2000 | [68] | 0–10 | <1 | DC-arc | 17.9 |
| Mazeikiai urban soils | 2000 | [68] | 0–10 | <1 | DC-arc | 11.3 |
| Joniskis urban soils | 2000 | [68] | 0–10 | <1 | DC-arc | 15.6 |
| Warsaw urban soils | 1996 | [7] | 0–20 | <1 | 450°/HCl | 25/7–65 |
| Kielce residential | 2016 | [69] | 0–20 | <2 | HNO$_3$ (HF?) | 8.2/1.8–12.3 |
| Kielce road soils | 2016 | [69] | 0–20 | <2 | HNO$_3$ (HF?) | 13.8/8.7–22.7 |
| Kielce urban greenery | 2016 | [69] | 0–20 | <2 | HNO$_3$ (HF?) | 9.9/0.9–24.1 |
| Kielce allotment gardens | 2016 | [69] | 0–20 | <2 | HNO$_3$ (HF?) | 11.5/3.1–41.9 |
| Kielce agricultural areas | 2016 | [69] | 0–20 | <2 | HNO$_3$ (HF?) | 4.7/0.6–11.1 |
| Berlin low-D residential | 1993–96 | [29] | 0–20 | <2 | XRF | 19/max 1340 |
| Berlin high-D residential | 1993–96 | [29] | 0–20 | <2 | XRF | 37/max 3230 |
| Berlin allotment | 1993–96 | [29] | 0–20 | <2 | XRF | 25/max 1280 |
| Berlin industrial | 1993–96 | [29] | 0–20 | <2 | XRF | 46/max 6470 |
| Vienna roadside soils | 1997 | [70] | 0–10 | none | HNO$_3$/HCl | 47/17–228 |
| Vienna parks | 1997 | [70] | 0–10 | none | HNO$_3$/HCl | 46/23–135 |
| Vienna urban soils | 2003 | [16] | 0–10 | <2 | HNO$_3$/HCl | 39.5 ± 12.5 |
| Sopron urban soils | 2012 | [39] | 0–10 | <2 | HNO$_3$/H$_2$O$_2$ | 118/11–1221 |
| Sopron urban soils | 2012 | [39] | 0–20 | none | HNO$_3$/H$_2$O$_2$ | 121/11–1449 |
| Szeged urban soils | 2005 | [17] | 0–10 | none | HNO$_3$/HCl | 36/26–88 |
| Novi Sad urban soils | 2017 | [72] | 0–15 | <2 | HNO$_3$/H$_2$O$_2$ | 27.7/4.4–459 |
| Zadar urban soils | 2003/04 | [73] | 0–10 | <2 | HNO$_3$/HCl | 83/43.5–893 |
| Torino urban soils | 1999 | [6] | 0–20 | <2 | HNO$_3$/HCl | 76/34–283 |
| Salerno urban soils | 2018 | [75] | 0–20 | <2 | HNO$_3$/HCl | 60.6/17.2–181 |
| Palermo public parks | 2000 | [31] | 0–10 | <2 | HNO$_3$/HCl | 63/10–344 |
| Sevilla parks-gardens | 2000 | [9] | 0–20 | <2 | HN O3/HCl | 42/11–374 |
| Vigo parks-gardens | 2013 | [12] | 0–20 | <2 | XRF | 62/23–208 |
| Tyumen urban soils | 2016 | [64] | 0–10 | <1 | XRF | 39/5–224 |
| Sangareddy urban soils | 2019 | [85] | 0–10 | <0.074 | XRF | 112/84–214 |
| Seoul Uijeongbu urban soils | 1996 | [58] | 0–15 | <0.18 | HNO$_3$/HClO$_4$ | 37/10–283 |
| Seoul Koyang urban soils | 1996 | [58] | 0–15 | <0.18 | HNO$_3$/HClO$_4$ | 48/26–220 |
| Shenyang urban soils | 2008 | [77] | 0–5 | <1 | HNO$_3$/HClO$_4$ | 33/19–275 |
| Beijing roadside soil | 2006 | [78] | 0–20 | fine soil | HNO$_3$/HClO$_4$/HF | 29.7 ± 5.7 |
| Beijing Parks | 2018 | [79] | 0–5 | <0.125 | HNO$_3$/HCl | 32/15–91 |
| Xian urban soils | 2016 | [1] | 0–10 | none | XRF | 29.4/22.2–97 |
| Xuzhou urban soils | 2004 | [80] | 0–10 | <2 | HF/HNO$_3$/HCl-XRF | 32/17–80 |
| Hangzhou Industrial | 2009 | [23] | 0–10 | <2 | HNO$_3$/HClO$_4$/HF | 58 ± 35 |
| Hangzhou Roadside | 2009 | [23] | 0–10 | <2 | HNO$_3$/HClO$_4$/HF | 50 ± 27 |

**Table A11.** *Cont.*

| | | | | | | |
|---|---|---|---|---|---|---|
| Hangzhou Residential | 2009 | [23] | 0–10 | <2 | HNO$_3$/HClO$_4$/HF | 38 ± 20 |
| Hangzhou Parks | 2009 | [23] | 0–10 | <2 | HNO$_3$/HClO$_4$/HF | 52 ± 31 |
| Hangzhou Industrial | 2019 | [63] | 0–30 | <0.15 | HNO$_3$/HClO$_4$/HF | 30/20–55 |
| Hangzhou Roadside | 2019 | [63] | 0–30 | <0.15 | HNO$_3$/HClO$_4$/HF | 30/25–49 |
| Hangzhou Residential | 2019 | [63] | 0–30 | <0.15 | HNO$_3$/HClO$_4$/HF | 26/20–31 |
| Shanghai urban soils | 2006 | [33] | 0–10 | <0.125 | HF/HNO$_3$/HClO$_4$ | 59/23–152 |
| Hong Kong Urban | 2004 | [81] | 0–15 | <2 | HNO$_3$/HClO$_4$ | 10.4/1.3–277 |
| Hong Kong suburban | 2004 | [81] | 0–15 | <2 | HNO$_3$/HClO$_4$ | 4.9/1.4–89 |
| Hong Kong country park | 2004 | [81] | 0–15 | <2 | HNO$_3$/HClO$_4$ | 4.8/2.0–20.2 |
| Danang urban soils | 1995 | [83] | 0–20 | <0.063 | HF/HNO$_3$/HClO$_4$ | 56/37–208 |
| Bangkok urban soils | 1996 | [84] | 0–5 | <2 | HNO$_3$/HClO$_4$ | 27/5–283 |
| Sydney roadside soils | 2009 | [36] | 0–10 | <0.425 | HNO$_3$/HCl | 69/6–225 |
| Adelaide garden soils | 2017 | [14] | 0–10 | <20 | HNO$_3$/HClO$_4$/HCl | 25.2/0.3–183 |
| Lithgow roadside soils | 2011 | [49] | 0–2 | <0.18 | XRF | 39/16–509 |
| Lithgow roadside soils | 2011 | [49] | 0–2 | <2 | XRF | 28/11–682 |
| Suva(Fiji)roadside soil | 2015 | [88] | 0–5 | none | HNO$_3$/HCl | 266/120–847 |
| Ottawa garden soils | 1993 | [86] | 0–5 | 0.1–0.25 | HNO$_3$/HF/HClO$_4$ | 12.1/6.3–19.4 |
| New York Parks | 1999 | [20] | 0–15 | <2 | HNO$_3$/HCl | 46/14–348 |
| Clay County urban soils | 2016/18 | [87] | 0–15 | <2 | HNO$_3$/H$_2$O$_2$ | 5.4/1.0–32.5 |
| Ocala urban soils | 2016/18 | [87] | 0–15 | <2 | HNO$_3$/H$_2$O$_2$ | 6.6/1.4–25.5 |
| Orlando urban soils | 2016/18 | [87] | 0–15 | <2 | HNO$_3$/H$_2$O$_2$ | 9.8/1.4–104 |
| Pensacola urban soils | 2016/18 | [87] | 0–15 | <2 | HNO$_3$/H$_2$O$_2$ | 10.4/2.8–29.9 |
| Tampa urban soils | 2016/18 | [87] | 0–15 | <2 | HNO$_3$/H$_2$O$_2$ | 7.8/0.9–43.8 |
| West Palm Beach urban soils | 2016/18 | [87] | 0–15 | <2 | HNO$_3$/H$_2$O$_2$ | 19.9/2.0–75.9 |
| Mexico | 2008 | [24] | 0–10 | <0.074 | XRF | 93/26–461 |
| Havana urban soils | 2018 | [51] | horizons | <2 | HNO$_3$/HCl | 73.5 ± 37.5 |
| Ibadan urban soils | 2006 | [89] | 0–15 | <0.075 | aqua regia | 32/7–248 |

**Table A12.** Mercury in urban soils.

| Location | sampling year | reference | sampling depth cm | grain size mm | | Hg mg/kg |
|---|---|---|---|---|---|---|
| Continental crust | | [40] | | | | 0.040 |
| Trondheim urban soils | 1994 | [66] | 0–2 | <2 | HNO$_3$/HCl | 0.13/0.02–4.5 |
| Trondheim urban soils | 2004 | [66] | 0–2 | <2 | 1/1 HNO$_3$ | 0.09/0.02–2.2 |
| Uppsala | 2003 | [8] | 0–5 | <2 | HNO$_3$/HCl | 0.14/<–3.66 |
| Uppsala | 2003 | [8] | 5–10 | <2 | HNO$_3$/HCl | 0.15/<–5.41 |
| Uppsala | 2003 | [8] | 10–20 | <2 | HNO$_3$/HCl | 0.13/<–1.11 |
| Berlin low-D residential | 1993–96 | [29] | 0–20 | <2 | unknown | 0.10/max 5.0 |
| Berlin high-D residential | 1993–96 | [29] | 0–20 | <2 | unknown | 0.34/max 3.5 |
| Berlin allotment | 1993–96 | [29] | 0–20 | <2 | unknown | 0.17/max 5.0 |
| Berlin industrial | 1993–96 | [29] | 0–20 | <2 | unknown | 0.21/max 71.2 |
| Vienna roadside soils | 1997 | [70] | 0–10 | none | HNO$_3$/HCl | 0.3/0.1–1.7 |
| Vienna parks | 1997 | [70] | 0–10 | none | HNO$_3$/HCl | 0.5/0.1–4.1 |
| Vienna urban soils | 2003 | [16] | 0–10 | <2 | HNO$_3$/HCl | 0.15 ±0.11 |
| Zadar urban soils | 2003/04 | [73] | 0–10 | <2 | HNO$_3$/HCl | 0.25/0.17–1.15 |
| Salerno urban soils | 2018 | [75] | 0–20 | <2 | HNO$_3$/HCl | 0.056/0.021–0.256 |
| Palermo public parks | 2000 | [31] | 0–10 | <2 | HNO$_3$/HCl | 0.68/0.04–6.96 |
| Beijing Parks | 2018 | [79] | 0–5 | <0.125 | HNO$_3$/HCl | 0.44/0.1–15.2 |
| Xuzhou urban soils | 2004 | [80] | 0–10 | <2 | HF/HNO$_3$/HCl-XRF | 0.17/0.02–0.83 |
| Hangzhou Industrial | 2019 | [63] | 0–30 | <0.15 | HNO$_3$/HClO$_4$/HF | 0.12/0.05–0.40 |
| Hangzhou Roadside | 2019 | [63] | 0–30 | <0.15 | HNO$_3$/HClO$_4$/HF | 0.09/0.06–0.19 |
| Hangzhou Residential | 2019 | [63] | 0–30 | <0.15 | HNO$_3$/HClO$_4$/HF | 0.12/0.06–0.28 |
| Lithgow roadside soils | 2011 | [49] | 0–2 | <0.18 | Cold vapour | 0.044/0.019–14.9 |
| Ottawa garden soils | 1993 | [86] | 0–5 | 0.1–0.25 | HNO$_3$/HF/HClO$_4$ | 0.048/0.018–0.111 |
| New York Parks | 1999 | [20] | 0–15 | <2 | HNO$_3$/HCl | 0.3/0.1–1.0 |
| Havana urban soils | 2018 | [51] | horizons | <2 | HNO$_3$/HCl | 0.51 ± 0.57 |

**Table A13.** Manganese in urban soils.

| Location | sampling year | reference | sampling depth cm | grain size mm | | Mn mg/kg |
|---|---|---|---|---|---|---|
| Upper crust | | [136] | | | | 774 |
| Continental crust | | [40] | | | | 716 |
| Aberdeen parkland | 1994 | [65] | 0–10 | <2 | HNO$_3$/HCl | 286 ± 52 |
| Aberdeen roadside | 1994 | [65] | 0–10 | <2 | HNO$_3$/HCl | 264 ± 24 |
| Uppsala | 2003 | [8] | 0–5 | <2 | HNO$_3$/HCl | 494/199–833 |
| Uppsala | 2003 | [8] | 5–10 | <2 | HNO$_3$/HCl | 526/162–940 |
| Uppsala | 2003 | [8] | 10–20 | <2 | HNO$_3$/HCl | 573/145–968 |
| Tallinn urban soils | 1987–90 | [67] | 0–10 | <1 | XRF | 320/76–1750 |
| Vilnius central | 2000 | [68] | 0–10 | <1 | DC-arc | 427 |

**Table A13.** *Cont.*

| | | | | | | |
|---|---|---|---|---|---|---|
| Vilnius peripheral | 2000 | [68] | 0–10 | <1 | DC-arc | 558 |
| Siauliai urban soils | 2000 | [68] | 0–10 | <1 | DC-arc | 478 |
| Mazeikiai urban soils | 2000 | [68] | 0–10 | <1 | DC-arc | 329 |
| Joniskis urban soils | 2000 | [68] | 0–10 | <1 | DC-arc | 418 |
| Warsaw urban soils | 1996 | [7] | 0–20 | <1 | 450°/HCl | 280/18–992 |
| Kielce residential | 2016 | [69] | 0–20 | <2 | $HNO_3$ (HF?) | 657/84–1804 |
| Kielce road soils | 2016 | [69] | 0–20 | <2 | $HNO_3$ (HF?) | 330/96–711 |
| Kielce urban greenery | 2016 | [69] | 0–20 | <2 | $HNO_3$ (HF?) | 365/39–1308 |
| Kielce allotment gardens | 2016 | [69] | 0–20 | <2 | $HNO_3$ (HF?) | 616/122–1674 |
| Kielce agricultural areas | 2016 | [69] | 0–20 | <2 | $HNO_3$ (HF?) | 441/99–1303 |
| Novi Sad urban soils | 2017 | [73] | 0–15 | <2 | $HNO_3/H_2O_2$ | 364/200–623 |
| Zadar urban soils | 2003/04 | [73] | 0–10 | <2 | $HNO_3/HCl$ | 871/142–1334 |
| Salerno urban soils | 2018 | [75] | 0–20 | <2 | $HNO_3/HCl$ | 716/302–1095 |
| Palermo public parks | 2000 | [31] | 0–10 | <2 | $HNO_3/HCl$ | 519/142–1241 |
| Sevilla parks-gardens | 2000 | [9] | 0–20 | <2 | $HNO_3/HCl$ | 468/335–893 |
| Vigo parks-gardens | 2013 | [12] | 0–20 | <2 | XRF | 517/168–879 |
| Xian urban soils | 2016 | [1] | 0–10 | none | XRF | 662/425–1126 |
| Xuzhou urban soils | 2004 | [80] | 0–10 | <2 | HF/$HNO_3$/HCl-XRF | 507/430–902 |
| Hangzhou Industrial | 2019 | [63] | 0–30 | <0.15 | $HNO_3/HClO_4$/HF | 931/698–2240 |
| Hangzhou Roadside | 2019 | [63] | 0–30 | <0.15 | $HNO_3/HClO_4$/HF | 816/598–1150 |
| Hangzhou Residential | 2019 | [63] | 0–30 | <0.15 | $HNO_3/HClO_4$/HF | 861/703–1300 |
| Danang urban soils | 1995 | [83] | 0–20 | <0.063 | HF/$HNO_3$/HCl | 387/155–775 |
| Bangkok urban soils | 1996 | [84] | 0–5 | <2 | $HNO_3/HClO_4$ | 290/50–810 |
| Sydney roadside soils | 2009 | [36] | 0–10 | <0.425 | $HNO_3/HCl$ | 750/16–2460 |
| Adelaide garden soils | 2017 | [14] | 0–10 | <20 | $HNO_3/HClO_4$/HCl | 169/0.1–750 |
| Ottawa garden soils | 1993 | [86] | 0–5 | 0.1–0.25 | $HNO_3$/HF/$HClO_4$ | 532/320–718 |
| New York Parks | 1999 | [20] | 0–15 | <2 | $HNO_3/HCl$ | 406/19–3117 |
| Havana urban soils | 2018 | [51] | horizons | <2 | $HNO_3/HCl$ | 578 ± 235 |
| Ibadan urban soils | 2006 | [89] | 0–15 | <0.075 | $HNO_3/HCl$ | 993/114–3053 |

**Table A14.** Molybdenum in urban soils.

| Location | sampling year | reference | sampling depth cm | grain size mm | | **Mo** **mg/kg** |
|---|---|---|---|---|---|---|
| Upper crust | | [136] | | | | 0.6 |
| Continental crust | | [40] | | | | 1.1 |
| Tallinn urban soils | 1987–90 | [67] | 0–10 | <1 | XRF | 1.5/<1–30 |
| Vilnius central | 2000 | [68] | 0–10 | <1 | DC-arc | 1.01 |
| Vilnius peripheral | 2000 | [68] | 0–10 | <1 | DC-arc | 0.83 |
| Siauliai urban soils | 2000 | [68] | 0–10 | <1 | DC-arc | 0.72 |
| Mazeikiai urban soils | 2000 | [68] | 0–10 | <1 | DC-arc | 0.64 |
| Joniskis urban soils | 2000 | [68] | 0–10 | <1 | DC-arc | 0.86 |
| Vienna roadside soils | 1997 | [70] | 0–10 | none | $HNO_3/HCl$ | 0.9/0.6–6.4 |
| Vienna parks | 1997 | [70] | 0–10 | none | $HNO_3/HCl$ | 0.9/0.6–1.8 |
| Vienna urban soils | 2003 | [16] | 0–10 | <2 | $HNO_3/HCl$ | 0.8 ±0.2 |
| Zadar urban soils | 2003/04 | [73] | 0–10 | <2 | $HNO_3/HCl$ | 0.97/0.40–5.77 |
| Salerno urban soils | 2018 | [75] | 0–20 | <2 | $HNO_3/HCl$ | 1.78/0.74–4.37 |
| Xuzhou urban soils | 2004 | [80] | 0–10 | <2 | HF/$HNO_3$/HCl-XRF | 1.2/0.7–2.5 |
| Ottawa garden soils | 1993 | [86] | 0–5 | 0.1–0.25 | $HNO_3$/HF/$HClO_4$ | 0.60/0.30–1.26 |
| New York Parks | 1999 | [20] | 0–15 | <2 | $HNO_3/HCl$ | 2.0/0.4–6.0 |
| Ibadan urban soils | 2006 | [89] | 0–15 | <0.075 | $HNO_3/HCl$ | 1.4/0.1–35 |

**Table A15.** Nickel in urban soils.

| Location | sampling year | reference | sampling depth cm | grain size mm | | **Ni** **mg/kg** |
|---|---|---|---|---|---|---|
| Upper crust | | [136] | | | | 34 |
| Continental crust | | [40] | | | | 56 |
| Aberdeen parkland | 1994 | [65] | 0–10 | <2 | $HNO_3/HCl$ | 14.9 ± 1.6 |
| Aberdeen roadside | 1994 | [65] | 0–10 | <2 | $HNO_3/HCl$ | 15.9 ± 1.7 |
| Trondheim urban soils | 1994 | [66] | 0–2 | <2 | $HNO_3/HCl$ | 45/6–231 |
| Trondheim urban soils | 2004 | [66] | 0–2 | <2 | 1/1 $HNO_3$ | 43/17–153 |
| Uppsala | 2003 | [8] | 0–5 | <2 | $HNO_3/HCl$ | 19/7–39 |
| Uppsala | 2003 | [8] | 5–10 | <2 | $HNO_3/HCl$ | 21/6–57 |
| Uppsala | 2003 | [8] | 10–20 | <2 | $HNO_3/HCl$ | 23/7–43 |
| Tallinn urban soils | 1987–90 | [67] | 0–10 | <1 | XRF | 15/4–65 |
| Vilnius central | 2000 | [68] | 0–10 | <1 | DC-arc | 14.1 |
| Vilnius peripheral | 2000 | [68] | 0–10 | <1 | DC-arc | 12.5 |
| Siauliai urban soils | 2000 | [68] | 0–10 | <1 | DC-arc | 14.6 |
| Mazeikiai urban soils | 2000 | [68] | 0–10 | <1 | DC-arc | 13.9 |
| Joniskis urban soils | 2000 | [68] | 0–10 | <1 | DC-arc | 13.8 |
| Kielce residential | 2016 | [69] | 0–20 | <2 | $HNO_3$ (HF?) | 6.4/2.0–13.7 |

**Table A15.** *Cont.*

| | | | | | | |
|---|---|---|---|---|---|---|
| Kielce road soils | 2016 | [69] | 0–20 | <2 | HNO$_3$ (HF?) | 5.2/3.1–9.9 |
| Kielce urban greenery | 2016 | [69] | 0–20 | <2 | HNO$_3$ (HF?) | 5.2/1.2–11.7 |
| Kielce allotment gardens | 2016 | [69] | 0–20 | <2 | HNO$_3$ (HF?) | 5.0/1.5–9.8 |
| Kielce agricultural areas | 2016 | [69] | 0–20 | <2 | HNO$_3$ (HF?) | 2.5/0.6–5.9 |
| Berlin low-D residential | 1993–96 | [29] | 0–20 | <2 | XRF | 5.8/max 91 |
| Berlin high-D residential | 1993–96 | [29] | 0–20 | <2 | XRF | 9.0/max 45 |
| Berlin allotment | 1993–96 | [29] | 0–20 | <2 | XRF | 6.1/max 51 |
| Berlin industrial | 1993–96 | [29] | 0–20 | <2 | XRF | 8.7/max 769 |
| Vienna roadside soils | 1997 | [70] | 0–10 | none | HNO$_3$/HCl | 31/20–45 |
| Vienna parks | 1997 | [70] | 0–10 | none | HNO$_3$/HCl | 29/21–38 |
| Vienna urban soils | 2003 | [16] | 0–10 | <2 | HNO$_3$/HCl | 28 ±4 |
| Sopron urban soils | 2012 | [39] | 0–10 | none | HNO$_3$/H$_2$O$_2$ | 26/6–99 |
| Sopron urban soils | 2012 | [39] | 0–20 | none | HNO$_3$/H$_2$O$_2$ | 25/4–71 |
| Szeged urban soils | 2005 | [17] | 0–10 | none | HNO$_3$/HCl | 34/17–44 |
| Novi Sad urban soils | 2017 | [72] | 0–15 | <2 | HNO$_3$/H$_2$O$_2$ | 27.7/10.2–74 |
| Zadar urban soils | 2003/04 | [73] | 0–10 | <2 | HNO$_3$/HCl | 62/35–120 |
| Torino urban soils | 1999 | [6] | 0–20 | <2 | HNO$_3$/HCl | 175/103–790 |
| Salerno urban soils | 2018 | [75] | 0–20 | <2 | HNO$_3$/HCl | 15.3/9.9–29.3 |
| Palermo public parks | 2000 | [31] | 0–10 | <2 | HNO$_3$/HCl | 17.8/7.0–38.6 |
| Sevilla | 2000 | [9] | 0–20 | <2 | HNO$_3$/HCl | 23/16–32 |
| Vigo parks-gardens | 2013 | [12] | 0–20 | <2 | XRF | 32.0/11.5–60 |
| Tyumen urban soils | 2016 | [64] | 0–10 | <1 | XRF | 44/5–283 |
| Sangareddy urban soils | 2019 | [85] | 0–10 | <0.074 | XRF | 31.5/19–51 |
| Seoul-Uijeongbu urban soils | 1995 | [58] | 0–15 | <0.18 | HNO$_3$/HClO$_4$ | 20/8–130 |
| Seoul-Koyang urban soils | 1995 | [58] | 0–15 | <0.18 | HNO$_3$/HClO$_4$ | 45/24–81 |
| Beijing Roadside soil | 2008 | [78] | 0–20 | fine soil | HNO$_3$/HClO$_4$/HF | 26.7 ±2.4 |
| Beijing Parks | 2018 | [79] | 0–5 | <0.125 | HNO$_3$/HCl | 27/18–39 |
| Xian urban soils | 2016 | [1] | 0–10 | none | XRF | 31.1/21.7–34.6 |
| Xuzhou urban soils | 2004 | [80] | 0–10 | <2 | HF/HNO$_3$/HCl-XRF | 30/25–54 |
| Hangzhou Industrial | 2019 | [63] | 0–30 | <0.15 | HNO$_3$/HClO$_4$/HF | 43/31–74 |
| Hangzhou Roadside | 2019 | [63] | 0–30 | <0.15 | HNO$_3$/HClO$_4$/HF | 39/32–46 |
| Hangzhou Residential | 2019 | [63] | 0–30 | <0.15 | HNO$_3$/HClO$_4$/HF | 37/28–43 |
| Shanghai urban soils | 2006 | [33] | 0–10 | <0.125 | HF/HNO$_3$/HClO$_4$ | 31/5–66 |
| Hong Kong Urban | 2004 | [81] | 0–15 | <2 | HNO$_3$/HClO$_4$ | 3.7/0.2–19.9 |
| Hong Kong suburban | 2004 | [81] | 0–15 | <2 | HNO$_3$/HClO$_4$ | 3.1/1.3–6.8 |
| Hong Kong country park | 2004 | [81] | 0–15 | <2 | HNO$_3$/HClO$_4$ | 4.8/1.8–9.6 |
| Danang urban soils | 1995 | [83] | 0–20 | <0.063 | HF/HNO$_3$/HCl | 15/9–68 |
| Bangkok urban soils | 1996 | [84] | 0–5 | <2 | HNO$_3$/HClO$_4$ | 23/4–52 |
| Sydney roadside soils | 2009 | [36] | 0–10 | <0.425 | HNO$_3$/HCl | 147/27–242 |
| Adelaide garden soils | 2017 | [14] | 0–10 | <20 | HNO$_3$/HClO$_4$/HCl | 7.8/<0.03–32.6 |
| Suva(Fiji)roadside soil | 2015 | [88] | 0–5 | none | HNO$_3$/HCl | 32/22–66 |
| Ottawa garden soils | 1993 | [86] | 0–5 | 0.1–0.25 | HNO$_3$/HF/HClO$_4$ | 15.8/10.5–23.1 |
| New York Parks | 1999 | [20] | 0–15 | <2 | HNO$_3$/HCl | 29/8–97 |
| Clay County urban soils | 2016/18 | [87] | 0–15 | <2 | HNO$_3$/H$_2$O$_2$ | 2.5/0.9–4.4 |
| Ocala urban soils | 2016/18 | [87] | 0–15 | <2 | HNO$_3$/H$_2$O$_2$ | 8.7/2.8–29.9 |
| Orlando urban soils | 2016/18 | [87] | 0–15 | <2 | HNO$_3$/H$_2$O$_2$ | 2.3/0.3–5.0 |
| Pensacola urban soils | 2016/18 | [87] | 0–15 | <2 | HNO$_3$/H$_2$O$_2$ | 4.2/1.8–8.9 |
| Tampa urban soils | 2016/18 | [87] | 0–15 | <2 | HNO$_3$/H$_2$O$_2$ | 3.0/0.5–10.3 |
| West Palm Beach urban soils | 2016/18 | [87] | 0–15 | <2 | HNO$_3$/H$_2$O$_2$ | 3.1/1.2–7.9 |
| Mexico | 2008 | [24] | 0–10 | <0.074 | XRF | 49/29–151 |
| Havana urban soils | 2018 | [51] | horizons | <2 | HNO$_3$/HCl | 72.1± 131 |
| Ibadan urban soils | 2006 | [59] | 0–15 | <0.075 | HNO$_3$/HCl | 16.5/7–118 |

**Table A16.** Lead in urban soils.

| Location | sampling year | reference | sampling depth cm | grain size mm | | **Pb** **mg/kg** |
|---|---|---|---|---|---|---|
| Upper crust | | [136] | | | | 17 |
| Continental crust | | [40] | | | | 14.8 |
| Aberdeen parkland | 1994 | [65] | 0–10 | <2 | HNO$_3$/HCl | 94 ± 216 |
| Aberdeen roadside | 1994 | [65] | 0–10 | <2 | HNO$_3$/HCl | 173 ± 34 |
| Trondheim urban soils | 1994 | [66] | 0–2 | <2 | HNO$_3$/HCl | 35/9–976 |
| Trondheim urban soils | 2004 | [66] | 0–2 | <2 | 1/1 HNO$_3$ | 32/16–1025 |
| Uppsala | 2003 | [8] | 0–5 | <2 | HNO$_3$/HCl | 26/9–358 |
| Uppsala | 2003 | [8] | 5–10 | <2 | HNO$_3$/HCl | 25/9–163 |
| Uppsala | 2003 | [8] | 10–20 | <2 | HNO$_3$/HCl | 26/10–160 |
| Tallinn urban soils | 1987–1990 | [67] | 0–10 | <1 | XRF | 50/<6–602 |
| Vilnius central | 2000 | [68] | 0–10 | <1 | DC-arc | 51.6 |
| Vilnius peripheral | 2000 | [68] | 0–10 | <1 | DC-arc | 24.8 |
| Siauliai urban soils | 2000 | [68] | 0–10 | <1 | DC-arc | 29.6 |
| Mazeikiai urban soils | 2000 | [68] | 0–10 | <1 | DC-arc | 15.3 |
| Joniskis urban soils | 2000 | [68] | 0–10 | <1 | DC-arc | 33.9 |
| Kielce residential | 2016 | [69] | 0–20 | <2 | HNO$_3$ (HF?) | 34/6.4–95 |

**Table A16.** *Cont.*

| | | | | | | |
|---|---|---|---|---|---|---|
| Kielce road soils | 2016 | [69] | 0–20 | <2 | HNO$_3$ (HF?) | 29/9–45 |
| Kielce urban greenery | 2016 | [69] | 0–20 | <2 | HNO$_3$ (HF?) | 36/1–102 |
| Kielce allotment gardens | 2016 | [69] | 0–20 | <2 | HNO$_3$ (HF?) | 23/7–103 |
| Kielce agricultural areas | 2016 | [69] | 0–20 | <2 | HNO$_3$ (HF?) | 9.9/4.7–28.9 |
| Berlin low-D residential | 1993–1996 | [29] | 0–20 | <2 | XRF | 50/max 2070 |
| Berlin high-D residential | 1993–1996 | [29] | 0–20 | <2 | XRF | 109/max 1490 |
| Berlin allotment | 1993–1996 | [29] | 0–20 | <2 | XRF | 62/max 722 |
| Berlin industrial | 1993–1996 | [29] | 0–20 | <2 | XRF | 87/max 4710 |
| Vienna roadside soils | 1997 | [70] | 0–10 | none | HNO$_3$/HCl | 86/20–354 |
| Vienna parks | 1997 | [70] | 0–10 | none | HNO$_3$/HCl | 86/38–243 |
| Vienna urban soils | 2003 | [16] | 0–10 | <2 | HNO$_3$/HCl | 64 ± 27 |
| Sopron urban soils | 2012 | [39] | 0–10 | none | HNO$_3$/H$_2$O$_2$ | 125/28–559 |
| Sopron urban soils | 2012 | [39] | 0–20 | none | HNO$_3$/H$_2$O$_2$ | 120/25–287 |
| Szeged urban soils | 2005 | [17] | 0–10 | none | HNO$_3$/HCl | 40/23–136 |
| Novi Sad urban soils | 2017 | [72] | 0–15 | <2 | HNO$_3$/H$_2$O$_2$ | 49/9–999 |
| Zadar urban soils | 2003/2004 | [73] | 0–10 | <2 | HNO$_3$/HCl | 80/44–553 |
| Torino urban soils | 1999 | [6] | 0–20 | <2 | HNO$_3$/HCl | 117/31–870 |
| Salerno urban soils | 2018 | [75] | 0–20 | <2 | HNO$_3$/HCl | 67/16–538 |
| Palermo public parks | 2000 | [31] | 0–10 | <2 | HNO$_3$/HCl | 202/57–682 |
| Sevilla parks-gardens | 2000 | [9] | 0–20 | <2 | HNO$_3$/HCl | 103/14–791 |
| Vigo parks-gardens | 2013 | [12] | 0–20 | <2 | XRF | 82/34–259 |
| Tyumen urban soils | 2016 | [64] | 0–10 | <1 | XRF | 19.6/8–430 |
| Sangareddy urban soils | 2019 | [85] | 0–10 | <0.074 | XRF | 17/3–32 |
| Seoul-Uijeongbu urban soils | 1995 | [58] | 0–15 | <0.18 | HNO$_3$/HClO$_4$ | 57/28–444 |
| Seoul-Koyang urban soils | 1995 | [58] | 0–15 | <0.18 | HNO$_3$/HClO$_4$ | 67/36–956 |
| Shenyang urban soils | 2008 | [77] | 0–5 | <1 | HNO$_3$/HClO$_4$ | 48/0.1–340 |
| Beijing roadside soil | 2008 | [78] | 0–20 | fine soil | HNO$_3$/HClO$_4$/HF | 35.4 ± 13.5 |
| Beijing Parks | 2018 | [79] | 0–5 | <0.125 | HNO$_3$/HCl | 33/8–92 |
| Xian urban soils | 2016 | [1] | 0–10 | none | XRF | 32.2/20.6–70.7 |
| Xuzhou urban soils | 2004 | [80] | 0–10 | <2 | HF/HNO$_3$/HCl-XRF | 36/16–120 |
| Hangzhou Industrial | 2009 | [23] | 0–10 | <2 | HNO$_3$/HClO$_4$/HF | 139 ± 107 |
| Hangzhou Roadside | 2009 | [23] | 0–10 | <2 | HNO$_3$/HClO$_4$/HF | 95 ± 81 |
| Hangzhou Residential | 2009 | [23] | 0–10 | <2 | HNO$_3$/HClO$_4$/HF | 91 ± 71 |
| Hangzhou Parks | 2009 | [23] | 0–10 | <2 | HNO$_3$/HClO$_4$/HF | 56 ± 39 |
| Hangzhou Industrial | 2019 | [63] | 0–30 | <0.15 | HNO$_3$/HClO$_4$/HF | 29/21–70 |
| Hangzhou Roadside | 2019 | [63] | 0–30 | <0.15 | HNO$_3$/HClO$_4$/HF | 30/25–39 |
| Hangzhou Residential | 2019 | [63] | 0–30 | <0.15 | HNO$_3$/HClO$_4$/HF | 27/20–45 |
| Shanghai urban soils | 2006 | [33] | 0–10 | <0.125 | HF/HNO$_3$/HClO$_4$ | 71/14–192 |
| Hong Kong urban parks | 2000 | [82] | 0–10 | <2 | HNO$_3$ | 93 ± 37 |
| Hong Kong Urban | 2004 | [81] | 0–15 | <2 | HNO$_3$/HClO$_4$ | 71/8–496 |
| Hong Kong Suburban | 2004 | [81] | 0–15 | <2 | HNO$_3$/HClO$_4$ | 49/16–161 |
| Hong Kong country park | 2004 | [81] | 0–15 | <2 | HNO$_3$/HClO$_4$ | 37/11–124 |
| Danang urban soils | 1995 | [83] | 0–20 | <0.063 | HF/HNO$_3$/HCl | 1.8/0.4–20.1 |
| Bangkok urban soils | 1996 | [84] | 0–5 | <2 | HNO$_3$/HClO$_4$ | 29/12–269 |
| Sydney roadside soils | 2009 | [36] | 0–10 | <0.425 | HNO$_3$/HCl | 64/24–198 |
| Adelaide garden soils | 2017 | [14] | 0–10 | <20 | HNO$_3$/HClO$_4$/HCl | 30/<0.1–268 |
| Lithgow roadside soils | 2011 | [49] | 0–2 | <0.18 | XRF | 46/<5–3490 |
| Lithgow roadside soils | 2011 | [49] | 0–2 | <2 | XRF | 27/<5–3200 |
| Suva(Fiji) roadside soil | 2015 | [88] | 0–5 | none | HNO$_3$/HCl | 59/21–135 |
| Ottawa Parks | 1993 | [86] | 0–5 | 0.1–0.25 | HNO$_3$/HF/HClO$_4$ | 33.8/15.6–205 |
| New York Parks | 1999 | [20] | 0–15 | <2 | HNO$_3$/HCl | 178/40–730 |
| Clay County urban soils | 2016/2018 | [87] | 0–15 | <2 | HNO$_3$/H$_2$O$_2$ | 28.4/5.5–165 |
| Ocala urban soils | 2016/2018 | [87] | 0–15 | <2 | HNO$_3$/H$_2$O$_2$ | 35.9/5.3–271 |
| Orlando urban soils | 2016/2018 | [87] | 0–15 | <2 | HNO$_3$/H$_2$O$_2$ | 18.7/1.8–245 |
| Pensacola urban soils | 2016/2018 | [87] | 0–15 | <2 | HNO$_3$/H$_2$O$_2$ | 86.3/5.2–466 |
| Tampa urban soils | 2016/2018 | [87] | 0–15 | <2 | HNO$_3$/H$_2$O$_2$ | 38.4/2.3–552 |
| West Palm Beach urban soils | 2016/2018 | [87] | 0–15 | <2 | HNO$_3$/H$_2$O$_2$ | 53.7/4.3–433 |
| Mexico | 2008 | [24] | 0–10 | <0.074 | XRF | 116/15–693 |
| Havana urban soils | 2018 | [51] | horizons | <2 | HNO$_3$/HCl | 73.5 ± 79.4 |
| Ibadan urban soils | 2006 | [89] | 0–15 | <0.075 | HNO$_3$/HCl | 47/9–648 |

**Table A17.** Antimony in urban soils.

| Location | sampling year | reference | sampling depth cm | grain size mm | | Sb mg/kg |
|---|---|---|---|---|---|---|
| Upper crust | | [136] | | | | 0.4 |
| Continental crust | | [40] | | | | 0.30 |
| Zadar urban soils | 2003/04 | [73] | 0–10 | <2 | HNO$_3$/HCl | 0.87/0.41–10.02 |
| Salerno urban soils | 2018 | [75] | 0–20 | <2 | HNO$_3$/HCl | 1.22/0.38–7.57 |
| Palermo public parks | 2000 | [31] | 0–10 | <2 | HNO$_3$/HCl | 3.0/1.1–15.7 |
| Xuzhou urban soils | 2004 | [80] | 0–10 | <2 | HF/HNO$_3$/HCl-XRF | 0.96/0.79–1.60 |
| Ottawa garden soils | 1993 | [86] | 0–5 | 0.1–0.25 | HNO$_3$/HF/HClO$_4$ | 0.22/0.11–1.00 |
| New York Parks | 1999 | [20] | 0–15 | <2 | HNO$_3$/HCl | 2.0/0.4–6.0 |

**Table A18.** Tin in urban soils.

| Location | sampling year | reference | sampling depth cm | grain size mm | | Sn mg/kg |
|---|---|---|---|---|---|---|
| Upper crust | | [136] | | | | 2.2 |
| Continental crust | | [40] | | | | 2.3 |
| Tallinn urban soils | 1987–90 | [67] | 0–10 | <1 | XRF | 3/<1–4 |
| Vilnius central | 2000 | [68] | 0–10 | <1 | DC-arc | 4.39 |
| Vilnius peripheral | 2000 | [68] | 0–10 | <1 | DC-arc | 2.87 |
| Siauliai urban soils | 2000 | [68] | 0–10 | <1 | DC-arc | 2.98 |
| Mazeikiai urban soils | 2000 | [68] | 0–10 | <1 | DC-arc | 2.35 |
| Joniskis urban soils | 2000 | [68] | 0–10 | <1 | DC-arc | 3.11 |
| Berlin low-D residential | 1993–96 | [29] | 0–20 | <2 | XRF | 3.4/max 267 |
| Berlin high-D residential | 1993–96 | [29] | 0–20 | <2 | XRF | 8.6/max 150 |
| Berlin allotment | 1993–96 | [29] | 0–20 | <2 | XRF | 4.9/max 112 |
| Berlin industrial | 1993–96 | [29] | 0–20 | <2 | XRF | 7.4/max 409 |
| Zadar urban soils | 2003/04 | [73] | 0–10 | <2 | $HNO_3/HCl$ | 8.6/6.5–19.2 |
| Salerno urban soils | 2018 | [75] | 0–20 | <2 | $HNO_3/HCl$ | 4.6/1.3–15.2 |
| Xuzhou urban soils | 2004 | [80] | 0–10 | <2 | $HF/HNO_3/HCl$-XRF | 4.2/2.2–11 |
| Ottawa garden soils | 1993 | [86] | 0–5 | 0.1–0.25 | $HNO_3/HF/HClO_4$ | 1.32/0.77–2.65 |
| New York Parks | 1999 | [20] | 0–15 | <2 | $HNO_3/HCl$ | 9/2–41 |

**Table A19.** Vanadium in urban soils.

| Location | sampling year | reference | sampling depth cm | grain size mm | | V mg/kg |
|---|---|---|---|---|---|---|
| Upper Crust | | [136] | | | | 106 |
| Continental crust | | [40] | | | | 98 |
| Tallinn urban soils | 1987–90 | [67] | 0–10 | <1 | XRF | 30/6–90 |
| Vilnius central | 2000 | [68] | 0–10 | <1 | DC-arc | 27 |
| Vilnius peripheral | 2000 | [68] | 0–10 | <1 | DC-arc | 32 |
| Siauliai urban soils | 2000 | [68] | 0–10 | <1 | DC-arc | 36 |
| Mazeikiai urban soils | 2000 | [68] | 0–10 | <1 | DC-arc | 40 |
| Joniskis urban soils | 2000 | [68] | 0–10 | <1 | DC-arc | 44 |
| Vienna roadside soils | 1997 | [70] | 0–10 | none | $HNO_3/HCl$ | 30/17–45 |
| Vienna parks | 1997 | [70] | 0–10 | none | $HNO_3/HCl$ | 28/19–41 |
| Vienna urban soils | 2003 | [16] | 0–10 | <2 | $HNO_3/HCl$ | 34 ± 5 |
| Zadar urban soils | 2003/2004 | [73] | 0–10 | <2 | $HNO_3/HCl$ | 99/29–125 |
| Salerno urban soils | 2018 | [75] | 0–20 | <2 | $HNO_3/HCl$ | 55/26.2–92.2 |
| Palermo public parks | 2000 | [31] | 0–10 | <2 | $HNO_3/HCl$ | 54/21–124 |
| Tyumen urban soils | 2016 | [64] | 0–10 | <1 | XRF | 95/5–162 |
| Xian urban soils | 2016 | [1] | 0–10 | none | XRF | 79/54–90 |
| Xuzhou urban soils | 2004 | [80] | 0–10 | <2 | $HF/HNO_3/HCl$-XRF | 72/5–101 |
| Danang urban soils | 1995 | [83] | 0–20 | <0.063 | $HF/HNO_3/HCl$ | 101/21–119 |
| New York Parks | 1999 | [20] | 0–15 | <2 | $HNO_3/HCl$ | 82/8–355 |
| Mexico | 2008 | [24] | 0–10 | <0.074 | XRF | 186/60–229 |
| Ottawa garden soils | 1993 | [86] | 0–5 | 0.1–0.25 | $HNO_3/HF/HClO_4$ | 46/29–71 |

**Table A20.** Zinc in urban soils.

| Location | sampling year | reference | sampling depth cm | grain size mm | | Zn mg/kg |
|---|---|---|---|---|---|---|
| Upper crust | | [136] | | | | 75 |
| Continental crust | | [40] | | | | 65 |
| Aberdeen parkland | 1994 | [65] | 0–10 | <2 | $HNO_3/HCl$ | 58 ± 8 |
| Aberdeen roadside | 1994 | [65] | 0–10 | <2 | $HNO_3/HCl$ | 113 ± 15 |
| Trondheim urban soils | 1994 | [66] | 0–2 | <2 | $HNO_3/HCl$ | 98/7–3420 |
| Trondheim urban soils | 2004 | [66] | 0–2 | <2 | 1/1 $HNO_3$ | 80/4–1056 |
| Uppsala | 2003 | [8] | 0–5 | <2 | $HNO_3/HCl$ | 84/45–149 |
| Uppsala | 2003 | [8] | 5–10 | <2 | $HNO_3/HCl$ | 90/38–245 |
| Uppsala | 2003 | [8] | 10–20 | <2 | $HNO_3/HCl$ | 99/27–191 |
| Tallinn urban soils | 1987–90 | [67] | 0–10 | <1 | XRF | 114/11–1560 |
| Vilnius central | 2000 | [68] | 0–10 | <1 | DC-arc | 136 |
| Vilnius peripheral | 2000 | [68] | 0–10 | <1 | DC-arc | 99 |
| Siauliai urban soils | 2000 | [68] | 0–10 | <1 | DC-arc | 107 |
| Mazeikiai urban soils | 2000 | [68] | 0–10 | <1 | DC-arc | 39 |
| Joniskis urban soils | 2000 | [68] | 0–10 | <1 | DC-arc | 76 |
| Warsaw urban soils | 1996 | [7] | 0–20 | <1 | 450°/HCl | 140/20–426 |
| Kielce residential | 2016 | [69] | 0–20 | <2 | $HNO_3$ (HF?) | 91/19–274 |
| Kielce road soils | 2016 | [69] | 0–20 | <2 | $HNO_3$ (HF?) | 87/39–235 |
| Kielce urban greenery | 2016 | [69] | 0–20 | <2 | $HNO_3$ (HF?) | 58/7–193 |
| Kielce allotment gardens | 2016 | [69] | 0–20 | <2 | $HNO_3$ (HF?) | 107/27–290 |
| Kielce agricultural areas | 2016 | [69] | 0–20 | <2 | $HNO_3$ (HF?) | 28/9–50 |
| Berlin low-D residential | 1993–96 | [29] | 0–20 | <2 | XRF | 85/max 3160 |

**Table A20.** *Cont.*

| | | | | | | |
|---|---|---|---|---|---|---|
| Berlin high-D residential | 1993–96 | [29] | 0–20 | <2 | XRF | 163/max 6040 |
| Berlin allotment | 1993–96 | [29] | 0–20 | <2 | XRF | 121/max 3160 |
| Berlin industrial | 1993–96 | [29] | 0–20 | <2 | XRF | 169/max 25210 |
| Vienna roadside soils | 1997 | [70] | 0–10 | none | $HNO_3/HCl$ | 157/77–688 |
| Vienna parks | 1997 | [70] | 0–10 | none | $HNO_3/HCl$ | 156/84–374 |
| Vienna urban soils | 2003 | [16] | 0–10 | <2 | $HNO_3/HCl$ | 141 ± 43 |
| Sopron urban soils | 2012 | [39] | 0–10 | none | $HNO_3/H_2O_2$ | 133/27–607 |
| Sopron urban soils | 2012 | [39] | 0–20 | none | $HNO_3/H_2O_2$ | 102/16–579 |
| Szeged urban soils | 2005 | [17] | 0–10 | none | $HNO_3/HCl$ | 203/137–228 |
| Novi Sad urban soils | 2017 | [72] | 0–15 | <2 | $HNO_3/H_2O_2$ | 101/46–194 |
| Zadar urban soils | 2003/04 | [73] | 0–10 | <2 | $HNO_3/HCl$ | 191/76–932 |
| Torino urban soils | 1999 | [6] | 0–20 | <2 | $HNO_3/HCl$ | 149/78–545 |
| Salerno urban soils | 2018 | [75] | 0–20 | <2 | $HNO_3/HCl$ | 129/47–633 |
| Palermo public parks | 2000 | [31] | 0–10 | <2 | $HNO_3/HCl$ | 138/52–433 |
| Sevilla parks-gardens | 2000 | [9] | 0–20 | <2 | $HNO_3/HCl$ | 86/26–450 |
| Vigo parks-gardens | 2013 | [12] | 0–20 | <2 | XRF | 150/59–234 |
| Tyumen urban soils | 2016 | [64] | 0–10 | <1 | XRF | 70/5–368 |
| Sangareddy urban soils | 2019 | [85] | 0–10 | <0.074 | XRF | 104/84–134 |
| Seoul-Uijeongbu urban soils | 1995 | [58] | 0–15 | <0.18 | $HNO_3/HClO_4$ | 164/60–864 |
| Seoul-Koyang urban soils | 1995 | [58] | 0–15 | <0.18 | $HNO_3/HClO_4$ | 189/87–1400 |
| Shenyang urban soils | 2008 | [77] | 0–5 | <1 | $HNO_3/HClO_4$ | 115/61–265 |
| Beijing roadside soil | 2008 | [78] | 0–20 | fine soil | $HNO_3/HClO_4/HF$ | 92 ± 19 |
| Beijing Parks | 2018 | [79] | 0–5 | <0.125 | $HNO_3/HCl$ | 137/69–288 |
| Xian urban soils | 2016 | [1] | 0–10 | none | XRF | 90/63–245 |
| Xuzhou urban soils | 2004 | [80] | 0–10 | <2 | $HF/HNO_3/HCl$-XRF | 102/53–380 |
| Hangzhou Industrial | 2009 | [23] | 0–10 | <2 | $HNO_3/HClO_4/HF$ | 346 ± 314 |
| Hangzhou Roadside | 2009 | [23] | 0–10 | <2 | $HNO_3/HClO_4/HF$ | 215 ± 121 |
| Hangzhou Residential | 2009 | [23] | 0–10 | <2 | $HNO_3/HClO_4/HF$ | 211 ± 74 |
| Hangzhou Parks | 2009 | [23] | 0–10 | <2 | $HNO_3/HClO_4/HF$ | 94 ± 61 |
| Hangzhou Industrial | 2019 | [63] | 0–30 | <0.15 | $HNO_3/HClO_4/HF$ | 103/72–187 |
| Hangzhou Roadside | 2019 | [63] | 0–30 | <0.15 | $HNO_3/HClO_4/HF$ | 108/89–165 |
| Hangzhou Residential | 2019 | [63] | 0–30 | <0.15 | $HNO_3/HClO_4/HF$ | 95/68–119 |
| Shanghai urban soils | 2006 | [33] | 0–10 | <0.125 | $HF/HNO_3/HClO_4$ | 301/103–1025 |
| Hong Kong urban parks | 2000 | [82] | 0–10 | <2 | $HNO_3$ | 168 ± 75 |
| Hong Kong Urban | 2004 | [81] | 0–15 | <2 | $HNO_3/HClO_4$ | 78/23–930 |
| Hong Kong suburban | 2004 | [81] | 0–15 | <2 | $HNO_3/HClO_4$ | 52/26–173 |
| Hong Kong country park | 2004 | [81] | 0–15 | <2 | $HNO_3/HClO_4$ | 44/25–136 |
| Danang urban soils | 1995 | [83] | 0–20 | <0.063 | $HF/HNO_3/HCl$ | 81/48–465 |
| Bangkok urban soils | 1996 | [84] | 0–5 | <2 | $HNO_3/HClO_4$ | 38/3–814 |
| Sydney roadside soils | 2009 | [36] | 0–10 | <0.425 | $HNO_3/HCl$ | 152/71–238 |
| Adelaide garden soils | 2017 | [14] | 0–10 | <20 | $HNO_3/HClO_4/HCl$ | 103/<0.6–662 |
| Lithgow roadside soil | 2011 | [49] | 0–2 | <0.18 | XRF | 120/40–2170 |
| Lithgow roadside soil | 2011 | [49] | 0–2 | <2 | XRF | 97/34–4950 |
| Suva(Fiji) roadside soil | 2015 | [88] | 0–5 | none | $HNO_3/HCl$ | 507/60–1617 |
| Ottawa garden soils | 1993 | [86] | 0–5 | 0.1–0.25 | $HNO_3/HF/HClO_4$ | 100/50–223 |
| New York Parks | 1999 | [20] | 0–15 | <2 | $HNO_3/HCl$ | 81/19–300 |
| Clay County urban soils | 2016/18 | [87] | 0–15 | <2 | $HNO_3/H_2O_2$ | 32.6/5.8–158 |
| Ocala urban soils | 2016/18 | [87] | 0–15 | <2 | $HNO_3/H_2O_2$ | 48.3/6.0–297 |
| Orlando urban soils | 2016/18 | [87] | 0–15 | <2 | $HNO_3/H_2O_2$ | 37.0/0.6–253 |
| Pensacola urban soils | 2016/18 | [87] | 0–15 | <2 | $HNO_3/H_2O_2$ | 69.8/7.6–331 |
| Tampa urban soils | 2016/18 | [87] | 0–15 | <2 | $HNO_3/H_2O_2$ | 67.2/4.8–1001 |
| West Palm Beach urban soils | 2016/18 | [87] | 0–15 | <2 | $HNO_3/H_2O_2$ | 70.9/9.5–383 |
| Mexico | 2008 | [24] | 0–10 | <0.074 | XRF | 447/95–1890 |
| Havana urban soils | 2018 | [51] | horizons | <2 | $HNO_3/HCl$ | 126 + 88 |
| Ibadan urban soils | 2006 | [89] | 0–15 | <0.075 | $HNO_3/HCl$ | 94/28–2643 |

## Appendix B. Urban Dust Data

Medians and ranges given in mg/kg, together with digestion/determination method. Severe pollutions have been marked red.

**Table A21.** Aluminum in road dust.

| | sampling year | reference | grain size | Digestion | % Al |
|---|---|---|---|---|---|
| Upper crust | | [136] | | | 8.10 |
| Continental crust | | [40] | | | 7.96 |
| Budapest city | 2003/2004 | [59] | <0.075 mm | $HNO_3/HCl$ | 0.85/0.43–1.02 |
| Chelyabinsk | 2017 | [92] | <1 mm | $HNO_3/HF/HClO_4$ | 0.85/0.56–1.20 |
| Seoul | 2004/05 | [59] | <0.075 mm | $HNO_3/HCl$ | 1.82/1.30–2.21 |
| Ottawa | 1993 | [86] | 0.1–0.25 | $HF/HNO_3/HClO_4$ | 4.77/1.22–5.81 |
| Buenos Aires | 2009 | [35] | <0.1 mm | $HNO_3$-HCl-HF | 3.09/1.56–3.87 |
| Luanda | 2002 | [100] | <0.1 mm | $HCl/HNO_3/H_2O$ | 0.47/0.28–0.88 |

**Table A22.** Iron in road dust.

| | sampling year | reference | grain size | Digestion | % Fe |
|---|---|---|---|---|---|
| Upper crust | | [136] | | | 5.20 |
| Continental crust | | [40] | | | 4.32 |
| Budapest city | 2004/2005 | [59] | <0.075 mm | $HNO_3$/HCl | 2.48/1.53–8.75 |
| Chelyabinsk | 2017 | [92] | <1 mm | $HNO_3$/HF/$HClO_4$ | 1.48/1.01–2.33 |
| Dhaka industrial + old | 2004 | [98] | <1 mm | XRF | 3.50 ± 2.61 |
| Dhaka commercial | 2004 | [98] | <1 mm | XRF | 2.82 ± 0.49 |
| Dhaka residential | 2004 | [98] | <1 mm | XRF | 2.41 ± 0.29 |
| Dhaka residential-low-traffic | 2004 | [98] | <1 mm | XRF | 2.22 ± 0.33 |
| Seoul | 2004/05 | [59] | <0.075 mm | $HNO_3$/HCl | 3.63/2.57–3.79 |
| Sydney | 2009 | [36] | <0.425 mm | $HNO_3$/HCl | 8.52/2.20–10.30 |
| Suva(Fiji) | 2015 | [88] | none | $HNO_3$/HCl | 4.10/2.61–10.48 |
| Ottawa | 1993 | [86] | 0.1–0.25 | HF/$HNO_3$/$HClO_4$ | 1.80/0.73–2.77 |
| Buenos Aires | 2009 | [35] | <0.1 mm | $HNO_3$-HCl-HF | 3.35/2.80–4.11 |
| Luanda | 2002 | [100] | <0.1 mm | HCl/$HNO_3$/$H_2O$ | 1.11/0.80–2.01 |

**Table A23.** Arsenic in road dust.

| | sampling year | reference | grain size | | As mg/kg |
|---|---|---|---|---|---|
| Upper crust | | [136] | | | 5.7 |
| Continental crust | | [40] | | | 1.7 |
| Budapest city | 2004–2005 | [59] | <0.075 mm | $HNO_3$/HCl | 11.6/8.2–15.8 |
| Chelyabinsk | 2017 | [92] | <1 mm | $HNO_3$/HF/$HClO_4$ | 3.8/1.6–7.9 |
| Dhaka industrial + old | 2004 | [98] | <1 mm | XRF | 7 ± 2.2 |
| Dhaka commercial | 2004 | [98] | <1 mm | XRF | 8 ± 3.1 |
| Dhaka residential | 2004 | [98] | <1 mm | XRF | 5 ± 0.9 |
| Dhaka residential-low traffic | 2004 | [98] | <1 mm | XRF | 4 ± 0.3 |
| Seoul | 2004–2005 | [98] | <0.075 mm | $HNO_3$/HCl | 24.9/15.2–31.2 |
| Ottawa | 1993 | [86] | 0.1–0.25 mm | HF/$HNO_3$/$HClO_4$ | 1.4/<–2.5 |
| Buenos Aires | 2009 | [35] | <0.1 mm | $HNO_3$-HCl-HF | 5.5/2.3–11.0 |
| Luanda | 2002 | [100] | <0.1 mm | HCl/$HNO_3$/$H_2O_2$ | 4.9/3.5–7.8 |

**Table A24.** Boron in road dust.

| | | reference | | | B mg/kg |
|---|---|---|---|---|---|
| Upper crust | | [136] | | | |
| Continental crust | | [40] | | | 11 |
| Luanda | 2002 | [100] | <0.1 mm | HCl/$HNO_3$/$H_2O_2$ | 8/3–16 |

**Table A25.** Barium in road dust.

| | sampling year | reference | grain size | | Ba mg/kg |
|---|---|---|---|---|---|
| Upper crust | | [136] | | | 628 |
| Continental crust | | [40] | | | 584 |
| Oslo | 1994 | [95] | <0.1 mm | $HNO_3$/$HClO_4$/HF | 526 ± 14 |
| Budapest city | 2004–2005 | [59] | <0.075 mm | $HNO_3$/HCl | 304/137–961 |
| Seoul | 2004–2005 | [59] | <0.075 mm | $HNO_3$/HCl | 570/217–716 |
| Ottawa | 1993 | [86] | 0.1–0.25 mm | HF/$HNO_3$/$HClO_4$ | 584/153–687 |
| Luanda | 2002 | [100] | <0.1 mm | HCl/$HNO_3$/$H_2O_2$ | 121/68–363 |

**Table A26.** Cadmium in road dust.

| | sampling year | reference | grain size | | Cd mg/kg |
|---|---|---|---|---|---|
| Upper crust | | [136] | | | 0.06 |
| Continental crust | | [40] | | | 0.10 |
| Oslo | 1994 | [95] | <0.1 mm | HNO$_3$/HClO$_4$/HF | 1.4 ± 0.2 |
| Budapest city | 2004–05 | [59] | <0.075 mm | HNO$_3$/HCl | 0.81/0.39–1.89 |
| Murcia | 2010 | [96] | <2 mm | HNO$_3$/HClO$_4$ | 0.67 ± 0.16 |
| Samsun residential-low traffic | 2007 | [97] | none | HNO$_3$/HCl | 0.30/0.19–0.40 |
| Samsun residential-high traffic | 2007 | [97] | none | HNO$_3$/HCl | 0.23/0.02–0.65 |
| Samsun industrial | 2007 | [97] | none | HNO$_3$/HCl | 1.47/0.04–6.16 |
| Chelyabinsk | 2017 | [92] | <1 mm | HNO$_3$/HF/HClO$_4$ | 0.4/0.06–2.00 |
| Delhi | 2009 | [90] | <0.075 mm | unknown | 2.65/1.9–3.8 |
| Seoul before rainy season | 1996 | [76] | <0.18 mm | HNO$_3$/HClO$_4$ | 1.5/0.4–3.1 |
| Seoul urban soils | 2004–05 | [59] | <0.075 mm | HNO$_3$/HCl | 3.45/1.40–6.65 |
| Shanghai | 2006 | [33] | <0.125 mm | HF/HNO$_3$/HClO$_4$ | 1.23/0.36–4.72 |
| Hong Kong | 2000 | [82] | <2 mm | HNO$_3$ | 3.77 ± 2.25 |
| Sydney | 2009 | [36] | <0.425 mm | HNO$_3$/HCl | 0.73/0.24–1.72 |
| Suva(Fiji) | 2015 | [88] | none | HNO$_3$/HCl | 3.7/2.4–12.2 |
| Ottawa | 1993 | [86] | 0.1–0.25 mm | HF/HNO$_3$/HClO$_4$ | 0.30/0.08–0.79 |
| Buenos Aires | 2009 | [35] | <0.1 mm | HNO$_3$-HCl-HF | 2.1/0.5–3.9 |
| Luanda | 2002 | [100] | <0.1 mm | HCl/HNO$_3$/H$_2$O$_2$ | 1.1/0.7–4.0 |

**Table A27.** Cobalt in road dust.

| | sampling year | reference | grain size | | Co mg/kg |
|---|---|---|---|---|---|
| Upper crust | | [136] | | | 15 |
| Continental crust | | [40] | | | 24 |
| Oslo | 1994 | [95] | <0.1 mm | HNO$_3$/HClO$_4$/HF | 19 ± 0.5 |
| Budapest city | 2004–05 | [59] | <0.075 mm | HNO$_3$/HCl | 7.9/4.8–12.6 |
| Madrid | 1990 | [95] | <0.1 mm | HNO$_3$/HClO$_4$/HF | 3 ± 0.6 |
| Samsun residential-low traffic | 2007 | [97] | none | HNO$_3$/HCl | 7.3/5.1–9.7 |
| Samsun residential-high traffic | 2007 | [97] | none | HNO$_3$/HCl | 8.1/4.9–10.2 |
| Samsun industrial | 2007 | [97] | none | HNO$_3$/HCl | 8.9/4.5–41.7 |
| Chelyabinsk | 2017 | [92] | <1 mm | HNO$_3$/HF/HClO$_4$ | 6.3/4.5–8.3 |
| Seoul before rainy season | 1996 | [76] | <0.18 mm | HNO$_3$/HClO$_4$ | 11.5/3.6–61 |
| Seoul | 2004–05 | [59] | <0.075 mm | HNO$_3$/HCl | 17.9/15.4–87.6 |
| Ottawa | 1993 | [86] | 0.1–0.25 mm | HF/HNO$_3$/HClO$_4$ | 8.65/2.31–11.15 |
| Suva(Fiji) | 2015 | [88] | none | HNO$_3$/HCl | 35/27–58 |
| Luanda | 2002 | [100] | <0.1 mm | HCl/HNO$_3$/H$_2$O$_2$ | 2.7/1.9–7.0 |

**Table A28.** Chromium in road dust.

| | sampling year | reference | grain size | | Cr mg/kg |
|---|---|---|---|---|---|
| Upper crust | | [136] | | | 73 |
| Continental crust | | [40] | | | 126 |
| Budapest city | 2004–2005 | [59] | <0.075 mm | HNO$_3$/HCl | 65.5/37.4–121 |
| Madrid | 1990 | [95] | <0.1 mm | HNO$_3$/HClO$_4$/HF | 61 ± 7 |
| Chelyabinsk | 2017 | [92] | <1 mm | HNO$_3$/HF/HClO$_4$ | 49/23–95 |
| Delhi | 2009 | [90] | <0.075 mm | unknown | 149/56–500 |
| Dhaka industrial + old | 2004 | [98] | <1 mm | XRF | 136 ± 35 |
| Dhaka commercial | 2004 | [98] | <1 mm | XRF | 105 ± 17 |
| Dhaka residential | 2004 | [98] | <1 mm | XRF | 99 ± 17 |
| Dhaka residential-low traffic | 2004 | [98] | <1 mm | XRF | 77 ± 14 |
| Seoul before rainy season | 1996 | [76] | <0.18 mm | HNO$_3$/HClO$_4$ | 36/10–420 |
| Seoul | 2004–2005 | [59] | <0.075 mm | HNO$_3$/HCl | 130/104–195 |
| Shanghai | 2006 | [33] | <0.125 mm | HF/HNO$_3$/HClO$_4$ | 159/18–1325 |
| Sydney | 2009 | [36] | <0.425 mm | HNO$_3$/HCl | 152/49–486 |
| Suva(Fiji) | 2015 | [88] | none | HNO$_3$/HCl | 40/21–82 |
| Ottawa | 1993 | [86] | 0.1–0.25 mm | HF/HNO$_3$/HClO$_4$ | 42/14.7–63.9 |
| Luanda | 2002 | [100] | <0.1 mm | HCl/HNO$_3$/H$_2$O$_2$ | 26/17–37 |

**Table A29.** Copper in road dust.

| | sampling year | reference | grain size | | Cu mg/kg |
|---|---|---|---|---|---|
| Upper crust | | [136] | | | 27 |
| Continental crust | | [40] | | | 25 |
| Oslo | 1994 | [95] | <0.1 mm | $HNO_3/HClO_4/HF$ | 123 ± 13 |
| Budapest city | 2004–05 | [59] | <0.075 mm | $HNO_3/HCl$ | 236/144–352 |
| Madrid | 1990 | [95] | <0.1 mm | $HNO_3/HClO_4/HF$ | 188 ± 24 |
| Murcia | 2010 | [96] | <2 mm | $HNO_3/HClO_4$ | 130± 39 |
| Samsun residential-low traffic | 2007 | [97] | none | $HNO_3/HCl$ | 46/12–69 |
| Samsun residential-high traffic | 2007 | [97] | none | $HNO_3/HCl$ | 101/33–203 |
| Samsun industrial | 2007 | [97] | none | $HNO_3/HCl$ | 158/20–352 |
| Chelyabinsk | 2017 | [92] | <1 mm | $HNO_3/HF/HClO_4$ | 56/15–218 |
| Delhi | 2009 | [90] | <0.075 mm | unknown | 192/87–499 |
| Dhaka industrial + old | 2004 | [98] | <1 mm | XRF | 105 ± 110 |
| Dhaka commercial | 2004 | [98] | <1 mm | XRF | 46 ± 19 |
| Dhaka residential | 2004 | [98] | <1 mm | XRF | 22 ± 9 |
| Dhaka residential-low traffic | 2004 | [98] | <1 mm | XRF | 14 ± 6.6 |
| Seoul before rainy season | 1996 | [76] | <0.18 mm | $HNO_3/HClO_4$ | 73/12–1860 |
| Seoul after rainy season | 1996 | [76] | <0.18 mm | $HNO_3/HClO_4$ | 63/12–225 |
| Seoul | 2004–05 | [59] | <0.075 | $HNO_3/HCl$ | 351/302–478 |
| Shanghai | 2006 | [33] | <0.125 mm | $HF/HNO_3/HClO_4$ | 197/17–1175 |
| Hong Kong | 2000 | [82] | <2 mm | $HNO_3$ | 173 ± 190 |
| Sydney | 2009 | [36] | <0.425 mm | $HNO_3/HCl$ | 544/314–730 |
| Suva(Fiji) | 2015 | [88] | none | $HNO_3/HCl$ | 172/59–328 |
| Ottawa | 1993 | [86] | 0.1–0.25 | $HF/HNO_3/HClO_4$ | 30/4.8–236 |
| Buenos Aires | 2009 | [35] | <0.1 mm | $HNO_3$-HCl-HF | 273/124–602 |
| Luanda | 2002 | [100] | <0.1 mm | $HCl/HNO_3/H_2O_2$ | 39/18–118 |

**Table A30.** Mercury in road dust.

| | sampling year | reference | grain size | | Hg mg/kg |
|---|---|---|---|---|---|
| Upper crust | | [136] | | | 0.05 |
| Continental crust | | [40] | | | 0.04 |
| Budapest city | 2004–2005 | [59] | <0.075 mm | $HNO_3/HCl$ | 0.13/0.07–0.22 |
| Chelyabinsk | 2017 | [92] | <1 mm | $HNO_3/HF/HClO_4$ | 2.0/<–4.8 |
| Seoul | 2004–2005 | [59] | <0.075 mm | $HNO_3/HCl$ | 0.45/0.27–0.58 |
| Ottawa | 1993 | [86] | 0.1–0.25 mm | $HF/HNO_3/HClO_4$ | 0.018/0.004–0.096 |
| Luanda | 2002 | [100] | <0.1 mm | $HCl/HNO_3/H_2O_2$ | 0.11/0.03–0.57 |

**Table A31.** Manganese in road dust.

| | sampling year | reference | grain size | | Mn mg/kg |
|---|---|---|---|---|---|
| Upper crust | | [136] | | | 774 |
| Continental crust | | [40] | | | 716 |
| Oslo | 1994 | [95] | <0.1 mm | $HNO_3/HClO_4/HF$ | 833 ± 16 |
| Budapest city | 2004–05 | [59] | <0.075 mm | $HNO_3/HCl$ | 417/345–1011 |
| Madrid | 1990 | [95] | <0.1 mm | $HNO_3/HClO_4/HF$ | 362 ± 13 |
| Samsun residential-low traffic | 2007 | [97] | none | $HNO_3/HCl$ | 147/134–179 |
| Samsun residential-high traffic | 2007 | [97] | none | $HNO_3/HCl$ | 161/140–166 |
| Samsun industrial | 2007 | [97] | none | $HNO_3/HCl$ | 162/133–171 |
| Chelyabinsk | 2017 | [92] | <1 mm | $HNO_3/HF/HClO_4$ | 421/226–739 |
| Seoul | 2004–05 | [59] | <0.075 mm | $HNO_3/HCl$ | 639/541–681 |
| Sydney | 2009 | [36] | <0.425 mm | $HNO_3/HCl$ | 1276/489–3966 |
| Ottawa | 1993 | [86] | 0.1–0.25 mm | $HF/HNO_3/HClO_4$ | 426/145–582 |
| Buenos Aires | 2009 | [35] | <0.1 mm | $HNO_3/HCl/HF$ | 622/464–906 |
| Luanda | 2002 | [100] | <0.1 mm | $HCl/HNO_3/H_2O_2$ | 238/157–728 |

**Table A32.** Molybdenum in road dust.

| | sampling year | reference | grain size | | Mo mg/kg |
|---|---|---|---|---|---|
| Upper crust | | [136] | | | 0.6 |
| Continental crust | | [40] | | | 1.1 |
| Oslo | 1994 | [95] | <0.1 mm | HNO$_3$/HClO$_4$/HF | 4 ± 0.3 |
| Budapest city | 2004–2005 | [59] | <0.075 mm | HNO$_3$/HCl | 5.00/1.25–8.76 |
| Seoul | 2004–2005 | [59] | <0.075 mm | HNO$_3$/HCl | 13.7/6.7–18.6 |
| Ottawa | 1993 | [86] | 0.1–0.25 mm | HF/HNO$_3$/HClO$_4$ | 1.38/0.38–2.16 |
| Buenos Aires | 2009 | [35] | <0.1 mm | HNO$_3$-HCl-HF | 3.6/2.4–7.2 |
| Luanda | 2002 | [100] | <0.1 mm | HCl/HNO$_3$/H$_2$O$_2$ | 1.9/1.2–6.3 |

**Table A33.** Nickel in road dust.

| | sampling year | reference | grain size | | Ni mg/kg |
|---|---|---|---|---|---|
| Upper crust | | [136] | | | 34 |
| Continental crust | | [40] | | | 56 |
| Oslo | 1994 | [95] | <0.1 mm | HNO$_3$/HClO$_4$/HF | 41 ± 1 |
| Budapest city | 2004–05 | [59] | <0.075 mm | HNO$_3$/HCl | 27.5/19.2–50 |
| Madrid | 1990 | [95] | <0.1 mm | HNO$_3$/HClO$_4$/HF | 44 ± 5 |
| Samsun residential-low traffic | 2007 | [97] | none | HNO$_3$/HCl | 69/33–323 |
| Samsun residential-high traffic | 2007 | [97] | none | HNO$_3$/HCl | 51/25–121 |
| Samsun industrial | 2007 | [97] | none | HNO$_3$/HCl | 39/19–97 |
| Chelyabinsk | 2017 | [92] | <1 mm | HNO$_3$/HF/HClO$_4$ | 21.9/13.2–52 |
| Delhi | 2009 | [90] | <0.075 mm | unknown | 36/27–62 |
| Dhaka industrial + old | 2004 | [98] | <1 mm | XRF | 35 ± 14 |
| Dhaka commercial | 2004 | [98] | <1 mm | XRF | 26 ± 4.7 |
| Dhaka residential | 2004 | [98] | <1 mm | XRF | 23 ± 4.2 |
| Dhaka residential-low traffic | 2004 | [98] | <1 mm | XRF | 24 ± 2.5 |
| Seoul before rainy season | 1996 | [76] | <0.18 mm | HNO$_3$/HClO$_4$ | 30/10–742 |
| Seoul | 2004–05 | [59] | <0.075 mm | HNO$_3$/HCl | 62/42–109 |
| Shanghai | 2006 | [33] | <0.125 mm | HF/HNO$_3$/HClO$_4$ | 84/8–1251 |
| Sydney | 2009 | [36] | <0.425 mm | HNO$_3$/HCl | 69/20–208 |
| Suva(Fiji) | 2015 | [88] | none | HNO$_3$/HCl | 54/32–110 |
| Ottawa | 1993 | [86] | 0.1–0.25 mm | HF/HNO$_3$/HClO$_4$ | 14.6/4.7–19.4 |
| Buenos Aires | 2009 | [35] | <0.1 mm | HNO$_3$-HCl-HF | 26.2/10.3–35.4 |
| Luanda | 2002 | [100] | <0.1 mm | HCl/HNO$_3$/H$_2$O$_2$ | 9.7/6.2–32 |

**Table A34.** Phosphorus in road dust.

| | sampling year | reference | grain size | | P mg/kg |
|---|---|---|---|---|---|
| Upper crust | | [136] | | | 654 |
| Continental crust | | [40] | | | 757 |
| Oslo | 1994 | [95] | <0.1 mm | HNO$_3$/HClO$_4$/HF | 1086 ± 50 |
| Budapest city | 2004–2005 | [59] | <0.075 mm | HNO$_3$/HCl | 927/550–1469 |
| Dhaka industrial + old | 2004 | [98] | <1 mm | XRF | 1220 ± 437 |
| Dhaka commercial | 2004 | [98] | <1 mm | XRF | 873 ± 131 |
| Dhaka residential | 2004 | [98] | <1 mm | XRF | 873 ± 262 |
| Dhaka residential-low traffic | 2004 | [98] | <1 mm | XRF | 742 ± 87 |
| Ottawa | 1993 | [86] | 0.1–0.25 mm | HF/HNO$_3$/HClO$_4$ | 606/344–875 |
| Seoul | 2004–2005 | [59] | <0.075 mm | HNO$_3$/HCl | 1088/748–1481 |
| Luanda | 2002 | [100] | <0.1 mm | HCl/HNO$_3$/H$_2$O$_2$ | 1180/580–2210 |

**Table A35.** Lead in road dust.

| | sampling year | reference | grain size | | Pb mg/kg |
|---|---|---|---|---|---|
| Upper crust | | [136] | | | 17 |
| Continental crust | | [40] | | | 14.8 |
| Oslo | 1994 | [95] | <0.1 mm | HNO$_3$/HClO$_4$/HF | 180 ±14 |
| Budapest city | 2004–05 | [59] | <0.075 mm | HNO$_3$/HCl | 408/49–1891 |
| Madrid | 1990 | [95] | <0.1 mm | HNO$_3$/HClO$_4$/HF | 1927 ± 508 |
| Murcia | 2010 | [96] | <2 mm | HNO$_3$/HClO$_4$ | 123 ± 21 |
| Chelyabinsk | 2017 | [92] | <1 mm | HNO$_3$/HF/HClO$_4$ | 21.9/13.2–52.2 |
| Samsun residential-low traffic | 2007 | [97] | none | HNO$_3$/HCl | 12.7/5.5–25.7 |
| Samsun residential-high traffic | 2007 | [97] | none | HNO$_3$/HCl | 36/12–94 |
| Samsun industrial | 2007 | [97] | none | HNO$_3$/HCl | 48/13–224 |

**Table A35.** *Cont.*

| | | | | | |
|---|---|---|---|---|---|
| Delhi | 2009 | [90] | <0.075 mm | unknown | 121/69–316 |
| Dhaka | 2004 | [98] | <1 mm | XRF | 54 ± 19 |
| Dhaka industrial + old | 2004 | [98] | <1 mm | XRF | 74 ±36 |
| Dhaka commercial | 2004 | [98] | <1 mm | XRF | 35 ± 8 |
| Dhaka residential | 2004 | [98] | <1 mm | XRF | 25 ± 5 |
| Seoul before rainy season | 1996 | [58] | <0.18 mm | $HNO_3/HClO_4$ | 80/22–27000 |
| Seoul after rainy season | 1996 | [58] | <0.18 mm | $HNO_3/HClO_4$ | 58/20–199 |
| Seoul | 2004–05 | [59] | <0.075 mm | $HNO_3/HCl$ | 214/130–284 |
| Shanghai | 2006 | [33] | <0.125 mm | $HF/HNO_3/HClO_4$ | 295/28–4443 |
| Hong Kong | 2000 | [82] | <2 mm | $HNO_3$ | 181 ± 93 |
| Sydney | 2009 | [36] | <0.425 mm | $HNO_3/HCl$ | 119/36–379 |
| Suva(Fiji) | 2015 | [88] | none | $HNO_3/HCl$ | 54/32–110 |
| Ottawa | 1993 | [86] | 0.1–0.25 mm | $HF/HNO_3/HClO_4$ | 33/12.6–85 |
| Buenos Aires | 2009 | [35] | <0.1 mm | $HNO_3$-HCl-HF | 296/168–405 |
| Luanda | 2002 | [100] | <0.1 mm | $HCl/HNO_3/H_2O_2$ | 306/74–1856 |

**Table A36.** Antimony in road dust.

| | sampling year | reference | grain size | | Sb mg/kg |
|---|---|---|---|---|---|
| Upper crust | | [136] | | | 0.4 |
| Continental crust | | [40] | | | 0.30 |
| Oslo | 1994 | [95] | <0.1 mm | $HNO_3/HClO_4/HF$ | 6 ± 0.5 |
| Budapest city | 2004–2005 | [59] | <0.075 mm | $HNO_3/HCl$ | 10.3/2.8–20.4 |
| Chelyabinsk | 2017 | [92] | <1 mm | $HNO_3/HF/HClO_4$ | 1.33/<–2.2 |
| Seoul | 2004–2005 | [59] | <0.075 mm | $HNO_3/HCl$ | 44.3/10.0–60.1 |
| Ottawa | 1993 | [86] | 0.1–0.25 mm | $HF/HNO_3/HClO_4$ | 0.42/0.09–1.62 |
| Buenos Aires | 2009 | [35] | <0.1 mm | $HNO_3$-HCl-HF | 4.5/2.0–13.7 |
| Luanda | 2002 | [100] | <0.1 mm | $HCl/HNO_3/H_2O$ | 2.8/1.1–3.7 |

**Table A37.** Tin in road dust.

| | Sampling year | reference | grain size | | Sn mg/kg |
|---|---|---|---|---|---|
| Upper crust | | [136] | | | 2.2 |
| Continental crust | | [40] | | | 2.3 |
| Budapest city | 2004–2005 | [59] | <0.075 mm | $HNO_3/HCl$ | 20.9/6.0–31.0 |
| Seoul | 2004–2005 | [59] | <0.075 mm | $HNO_3/HCl$ | 39.7/21.1–50.4 |
| Ottawa | 1993 | [86] | 0.1–0.25 mm | $HF/HNO_3/HClO_4$ | 1.19/0.30–10.3 |
| Buenos Aires | 2009 | [35] | <0.1 mm | $HNO_3$-HCl-HF | 11.6/3.3–23 |

**Table A38.** Vanadium in road dust.

| | sampling year | reference | grain size | | V |
|---|---|---|---|---|---|
| Upper crust | | [136] | | | 106 |
| Continental crust | | [40] | | | 98 |
| Budapest city | 2004–2005 | [59] | <0.075 mm | $HNO_3/HCl$ | 25.1/14.7–29.6 |
| Madrid | 1990 | [95] | <0.1 mm | $HNO_3/HClO_4/HF$ | 17 ± 2 |
| Dhaka industrial + old | 2004 | [98] | <1 mm | XRF | 72 ± 23 |
| Dhaka commercial | 2004 | [98] | <1 mm | XRF | 68 ± 13 |
| Dhaka residential | 2004 | [98] | <1 mm | XRF | 64 ± 11 |
| Dhaka residential-low traffic | 2004 | [98] | <1 mm | XRF | 64 ± 6 |
| Seoul | 2004–2005 | [59] | <0.075 mm | $HNO_3/HCl$ | 35/23–43 |
| Ottawa | 1993 | [86] | 0.1–0.25 mm | $HF/HNO_3/HClO_4$ | 34/14–51 |
| Luanda | 2002 | [100] | <0.1 mm | $HCl/HNO_3/H_2O_2$ | 20/13–30 |

**Table A39.** Zinc in road dust.

| | sampling year | reference | grain size | | Zn mg/kg |
|---|---|---|---|---|---|
| Upper crust | | [136] | | | 75 |
| Continental crust | | [40] | | | 65 |
| Oslo | 1994 | [95] | <0.1 mm | HNO$_3$/HClO$_4$/HF | 412 ± 61 |
| Budapest city | 2004–05 | [59] | <0.075 mm | HNO$_3$/HCl | 891/317–2110 |
| Madrid | 1990 | [95] | <0.1 mm | HNO$_3$/HClO$_4$/HF | 476 ± 30 |
| Murcia | 2010 | [96] | <2 mm | HNO$_3$/HClO$_4$ | 377 ± 32 |
| Samsun residential-low traffic | 2007 | [97] | none | HNO$_3$/HCl | 78.8/43.0–89.0 |
| Samsun residential-high traffic | 2007 | [97] | none | HNO$_3$/HCl | 106/72–142 |
| Samsun industrial | 2007 | [97] | none | HNO$_3$/HCl | 117/53–173 |
| Chelyabinsk | 2017 | [92] | <1 mm | HNO$_3$/HF/HClO$_4$ | 154/66–616 |
| Delhi | 2009 | [90] | <0.075 mm | unknown | 285/188–524 |
| Dhaka industrial + old | 2004 | [98] | <1 mm | XRF | 169 ± 71 |
| Dhaka commercial | 2004 | [98] | <1 mm | XRF | 154 ± 42 |
| Dhaka residential | 2004 | [98] | <1 mm | XRF | 97 ± 29 |
| Dhaka residential-low traffic | 2004 | [98] | <1 mm | XRF | 65 ± 15 |
| Seoul before rainy season | 1996 | [76] | <0.18 mm | HNO$_3$/HClO$_4$ | 247/61–1200 |
| Seoul after rainy season | 1996 | [76] | <0.18 mm | HNO$_3$/HClO$_4$ | 197/49–1160 |
| Seoul | 2004–05 | [59] | <0.075 mm | HNO$_3$/HCl | 1476/1075–2065 |
| Shanghai | 2006 | [38] | <0.125 mm | HF/HNO$_3$/HClO$_4$ | 734/82–2136 |
| Hong Kong | 2000 | [82] | <2 mm | HNO$_3$ | 1450 ± 869 |
| Sydney | 2009 | [36] | <0.425 mm | HNO$_3$/HCl | 1109/557–2117 |
| Suva(Fiji) | 2015 | [88] | none | HNO$_3$/HCl | 685/146–3263 |
| Ottawa | 1993 | [86] | 0.1–0.25 mm | HF/HNO$_3$/HClO$_4$ | 99/29–194 |
| Buenos Aires | 2009 | [35] | <0.1 mm | HNO$_3$-HCl-HF | 766/370–1228 |
| Luanda | 2002 | [100] | <0.1 mm | HCl/HNO$_3$/H$_2$O$_2$ | 271/142–1412 |

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
