# Peer review of "Urban Soils and Road Dust—Civilization Effects and Metal Pollution—A Review"

_environments, doi:10.3390/environments7110098_

Round 1
Reviewer 1 Report
This manuscript (MS) is an interesting issue related to urban soils. The topic fits the aims and scope of the Environments. In my point of view, the manuscript needs in depth transformation.
Here is a list of corrections to be made to the text.
Title
I think that title "Urban Soils" is too short, this is a very broad title. It needs to give details what aspects of soils are considered. In addition, the MS discusses road dusts, it is not shown in the title.
Abstract
Line 6 Please change "Urban soil have been changed" to "Urban soil has been changed" or "Urban soils have been changed".
Lines 7-8 I believe that the mostly MS deals with metals in soils and urban dust but not with " changes from compaction, mixing, water retention, nutrient inputs..."
The author gives many examples (line 11), but a large number of reference books exist that provide information on the local concentration of metals. The aim of any review is analysis, criticism and finding patterns. The author shows two revealed patterns:
(1) "In the monsoonal regions, pollutants are swept away to the watershed, leaving the soils less polluted than in Europe" (lines 13-14), but term "monsoonal regions" uses for different areas. The regional monsoons are the North American monsoon , North African
monsoon , Indian monsoon , East Asian monsoon , Western North Pacific monsoon , South American monsoon, South African monsoon, and the Australian monsoon... Author describes Beijing (Line 659), Shanghai (Line 718), Bangkok (Line 774) as a monsoon-influenced areas but only for Dhaka shows (line 887) that the ranges of average contents of Pb, As, Zn,Cu were much lower than in many European road dust samples, which may be due
to monsoons. But it is not author' conclusion, it was shown in paper 75.
(2) Lines 14-16. Author writes that metals concentrations in urban soils is higher in 3rd world countries, and which is not treated within this paper in details. But if it is not treated within this paper, that information should not be in abstract. There is no that conclusion in the MS text except abstract.
Line 22. Last part of MS?
Body of MS
introduction should give the aim and organizational structure of MS. This would be useful since the manuscript is poorly structured. MS has long p. 1, there is no p.2 and there are short p.3 and p.4
P.1 was divided to many sub titles which make it difficult to follow for readers. Please combine some related sub titles.
P.4 deals with roadside flora. But it is very short and unclear. Perhaps it needs to be removed? What is its purpose?
The MS contains conflicting statements. For example, it remains unclear whether the fine dust fraction contains more or less metals than the coarse one. Lines 151-152. "In soil analysis, dry sieving down to 2 mm grain size is usual, but some papers refer to lower grain sizes (details given)" Where are given? What papers? Line 191. "Sieving to finer fractions than 2 mm, seems to yield higher data for Cr".
There are no conclusions.
Author Response
Urban soils and road dust - civilization effects and metal pollution - a review" Because there was some confusion about the numbering of the chapters, I have added an index between keywords and introduction.
As an update, I have added some 20 references, preferably from the last 2 years The tables in the appendix are part of the paper, and might be numbered, if the editors like, and brought to equal spaces.
Reviewer 2 Report
Dear author,
Detailed review in the attached file.
Best regards.

Author Response
Indeed, a general conclusion on the aspects of soil pollution was lacking, which has been added in this revised version.
Recent data from the US (New York, Florida) have been found and added, as well as from Siberia, India, Fiji etc.
A further rather new cumulative index for characterization of metal contaminations has been added.
Please apologize typing errors. Automatic corrections are for German spelling on this computer. The format of the references will be changed, when the text has been approved.
Reviewer 3 Report
-
The overall content of the manuscript does not tally with the main title. Author has provided more information related to urban metal soil pollution, and therefore suggest to change the title and narrow down the focus of the manuscript to only metal pollution. In addition, there are grammatical and structural issues that should be addressed prior to possible publication. A major concern is that this review is not critically summarised the information given and a lot of unnecessary information were discussed, making the manuscript too long and not eye-catching. Section numbering is inappropriate and could not find Section 2.
-
I have noticed that most of the references used in the manuscript have been published before10 years. A review should be at least focused on the references related to last 10 years and author has failed to review most of the recent publications related to metal pollution in urban soils. To the best of my knowledge, a number of research groups around the world (ex: Australia, China and Germany) has done a significant work in this area and published several papers.
-
Most of the statements and statistics were given without citing any references even though it has been mentioned in the manuscript.
Example: Some papers use a sampling grid including presumably non-contaminated suburban areas, whereas others sample preferably at hotspots.
Enrichment of antimony in dusts has been observed in many countries, even in Arctic ice.
-
Line 216- Metal corrosion increase Zn, Cd, Cd, Ni, Cu and Mo [26]. Reference #26 discusses the anthropogenic Pt, Pd and Rh levels in soils from major avenues of Sao Paulo City, Brazil. There is no relationship with the given sentence and the reference. This has been observed in many places throughout the manuscript.
-
Section 1.2.3 - There are several pollution indices available for characterization of metal contamination. However, author has summarized few indices which had been published more than two decades ago. Some pollution indices have been modified/proposed in recent years.
-
Section 1.2.4 - Author has provided many details regarding the metal contamination in different cities and regions around the world. The sources of details are lacking and I doubt that how author has given these details without citing references. Tables are not attractive and difficult to follow the discussion, tallying data. Under the discussion related to different regions, only one publication was discussed even though there are a plenty of studies available. As suggested earlier, details should be critically summarised (i.e. comparisons using tables and figures, pros and cons of methods and data) in a review rather than just mentioning different studies. Rather than giving this much of details, it would be good to present a comparison by selecting major urbanised cities.
-
Section 3. Platinum metals – Author has given a lot of details related to Pt metals discussing the distribution among soil, road dust, flora and different regions. Same comment here as above.
-
What are the future research directions of urban soils and related studies? The contribution of this review and it's major findings to scientific world was not discussed, which is another drawback of this manuscript.
Author Response
The title has been changed.
References already exist that provide already information about urban dusts and soils. However, sampling and disgestion methods are often neglected, because it seems that the writer has not done the data by himself.
This review also aims about systematic errors from sampling and digestion, as a start for further investigations. The term "monsoonal regions" is in deed incorrect, and I am now referring to mean annual precipitations to trace washouts. The abstract has been corrected.
The original numbering of chapters has been given between keywords and introduction.
Roadside flora: there is not much known about; text has been revised.
research directions: a global standardization of sampling, sieving and digestion is needed to ensure compatibility of results, which has to be based upon a profound investigation of the respective parameters from samples at the same sites. A mere compilation of means is useless.
Round 2
Reviewer 1 Report
Paper has been improved. It can be accepted in present form.
Reviewer 2 Report
Dear authors!
All comments of the reviewer are taken into your paper and necessary corrections are made. But:
In section 4. "The Roadside flora" it is necessary to correct the Latin names of the plants in the following lines: 2071, 2082, 2120, 2137-2139, 2146.
Best regards